*Resource*

# A comprehensive molecular atlas of the mesenchymal cell types in the mouse liver

Riikka Pietilä[1], Guillem Genové [2], Giuseppe Mocci[2], Yuyang Miao[2], Jianping Liu [2,3], Stefanos Leptidis [2], Francesca Del Gaudio [2,4], Martin Uhrbom [2], Elisa Vázquez-Liébanas [1], Sonja Gustafsson[2], Byambajav Buyandelger[2], Elisabeth Raschperger[2], Johan L M Björkegren[2], Emil M Hansson[2,5], Konstantin Gaengel [1], Maarja Andaloussi Mäe[1], Marie Jeansson [2], Michael Vanlandewijck [1,2], Liqun He [1], Carina Strell[1,6], Xiao-Rong Peng [7], Urban Lendahl [4], Christer Betsholtz [1,2✉] & Lars Muhl [2,6✉]

## Abstract

**The liver plays crucial roles in many essential physiological processes, and its impaired function due to liver fibrosis from various causes is an increasingly significant health issue. The liver's functionality relies on the precise arrangement of its cellular structures, yet the molecular architecture of these units remains only partially understood. We created a comprehensive molecular atlas detailing the major cell types present in the adult mouse liver through deep single-cell RNA sequencing. Our analysis offers new insights into hepatic endothelial and mesenchymal cells, specifically highlighting the diversity of cells in the periportal microvasculature, the sinusoids, and the portal vein, the latter exhibiting a mixed arteriovenous phenotype. We identified distinct subpopulations of hepatic stellate cells, fibroblasts, and vascular mural cells located in different anatomical liver regions. Comparisons with transcriptomic data from disease models indicate that a previously unrecognized capsular population of hepatic stellate cells expands in response to fibrotic disease. Our findings reveal that various fibroblast subpopulations respond differently to pathological insults. This data resource will be relevant for the advancement of therapies targeting hepatic diseases.**

**Keywords** Hepatic Stellate Cells (HSC); Liver Cell Transcriptomes; Liver Mesenchymal Cell Heterogeneity; Peribiliary Vasculature; Single-cell RNA-sequencing (scRNA-seq)
**Subject Category** Methods & Resources

## Introduction

The liver performs many critical physiological functions, including metabolic regulation, immune surveillance, and the detoxification of harmful metabolites and toxins (Shetty et al, 2018). Structurally, the liver is comprised of lobules, each featuring a specific arrangement of cells that distribute and function along an axis extending from the portal area (portal triad) to the central vein (Fig. 1A). Hepatocytes, the primary parenchymal cells of the liver, are aligned along this portal-central axis and carry out various metabolic and synthetic functions, including the production of plasma proteins and bile (Halpern et al, 2017).

The vascular system of the liver displays unique characteristics, notably a dual blood supply comprising (i) nutrient-rich, oxygen-poor blood from the portal vein, which drains the gut, and (ii) oxygen-rich blood from the hepatic artery, a branch from the celiac trunk of the aorta. These two blood supplies converge in the major hepatic capillary system, known as the sinusoids, which drain into the central vein. The sinusoids are highly specialized and distinguish in multiple ways from capillaries in most other organs. Their distinct features include discontinuous endothelial lining, which allows macromolecules to exchange freely between the blood and parenchyma (Sorensen et al, 2015), as well as the lack of a distinct basement membrane and associated pericytes. A second, smaller, capillary bed of the liver, the peribiliary capillaries, supply the bile duct system and support cholangiocyte function; this vascular bed arises directly from branches of the hepatic artery (Gaudio et al, 1996; Haratake et al, 1990). An elaborate paracrine signaling between cholangiocytes and the cells of the peribiliary vasculature (PBV) regulates angiogenesis and vascular morphology in this region (Morell et al, 2013). However, the transcriptomic

[1]Department of Immunology, Genetics, and Pathology, Uppsala University, Uppsala, Sweden. [2]Department of Medicine (Huddinge), Karolinska Institutet, Huddinge, Sweden. [3]Jiangxi Provincial Key Laboratory of Digestive Diseases, Department of Gastroenterology, The First Affiliated Hospital, Jiangxi Medical College, Nanchang University, 330006 Nanchang, Jiangxi, China. [4]Department of Cell and Molecular Biology, Karolinska Institutet, Stockholm, Sweden. [5]Department of Laboratory Medicine, Karolinska Institutet, Stockholm, Sweden. [6]Department of Clinical Medicine, Centre for Cancer Biomarkers (CCBIO), University of Bergen, Bergen, Norway. [7]Bioscience Metabolism, Research and Early Development Cardiovascular, Renal and Metabolism, BioPharmaceuticals R&D, AstraZeneca, Gothenburg, Sweden. ✉E-mail: Christer.Betsholtz@igp.uu.se; Christer.Betsholtz@ki.se; Lars.Muhl@ki.se; Lars.Muhl@uib.no

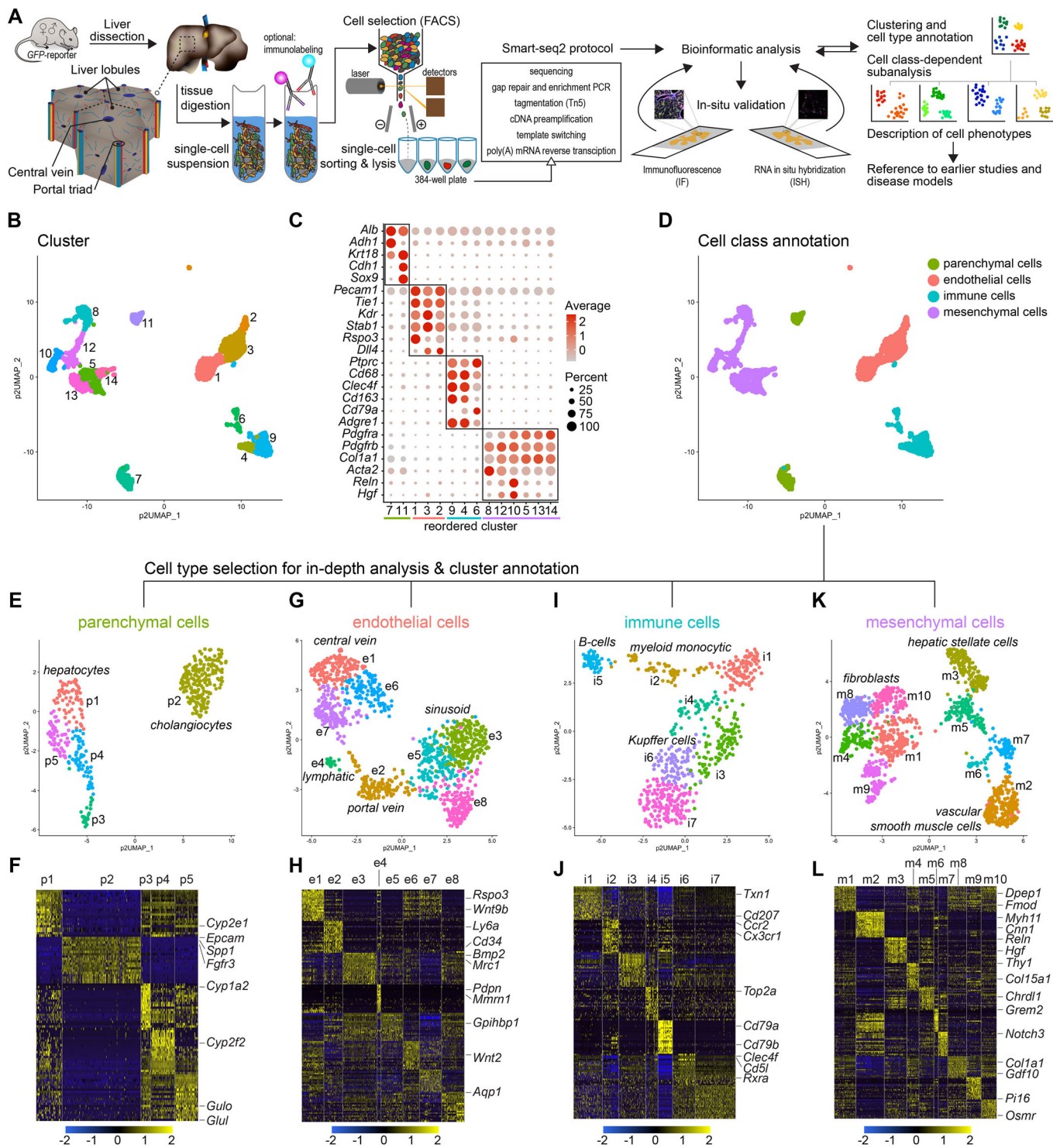

**Figure 1. Liver cells characterized by scRNA-sequencing.**

(A) Schematic overview of the experimental layout. (B) UMAP visualization of the clustering result using pagoda2 multilevel setting of the complete adult mouse liver scRNA-seq dataset (3491 single-cell transcriptomes), annotated and color coded for the different clusters. (C) Dot plot showing the expression of canonical marker genes representative to the annotated cell type classes. (D) UMAP visualization of the complete dataset color coded for cell type classes: parenchymal cells, endothelial cells, immune cells, and mesenchymal cells. (E–L) UMAP visualization of the clustering results and heat maps showing the top20 cluster-enriched genes in the separate analysis of the parenchymal cell (E, F), endothelial cell (G, H), immune cell (I, J), and mesenchymal cell (K, L) datasets. Exemplary genes used for the annotation of the datasets are indicated. Heat maps in this figure and in all other figures from which not all genes are readable can be accessed with full annotation at https://doi.org/10.5281/zenodo.16875427.

identities of the vascular cells in the peribiliary capillary network still have remained unidentified.

The space between the sinusoidal endothelial wall and the hepatocytes harbors two liver-specific cell types, the hepatic stellate cells (HSC), which are mesenchymal cells with properties similar to both pericytes and fibroblasts (Kamm and McCommis, 2022), and the Kupffer cells, which are a type of resident macrophage (Guilliams and Scott, 2022). Additionally, the liver contains the cholangiocyte-lined bile duct system (Tabibian et al, 2013), which runs parallel to the portal veins, venules, and hepatic arteries. Fibroblasts in the connective tissue surrounding the portal triad, together with mesothelial cells at the liver capsule, contribute to the formation of hepatic fibrous tissue sheaths also known as *the tunics of Glisson* (Helling and McCleary, 2016; Wells, 2014a).

While the anatomy and histology of the liver is well-established, in-depth information about how gene expression patterns differ among the various specialized hepatic cell types under physiological conditions and in disease has only recently been unraveled, primarily due to the advent of single-cell RNA-sequencing (scRNA-seq) and spatial transcriptomic techniques (Dobie et al, 2019b; Halpern et al, 2018; Halpern et al, 2017; Hildebrandt et al, 2021; MacParland et al, 2018; The Tabula Muris Consortium, 2018; Watson et al, 2025). These studies have provided rich transcriptomic information about some of the major hepatic cell types at homeostasis, including hepatocytes, endothelial cells, and Kupffer cells, but limited or no information about other hepatic cell types, including vascular mural cells, fibroblasts, and cholangiocytes. For example, it is largely unexplored how HSC compare to their closest relatives in other organs; the vascular mural cells. Moreover, a parallel analysis of the different mesenchymal cell populations of the liver has not been conducted. Such information would be crucial to better understand the liver's cellular architecture and how hepatic cells respond to injury and disease. A more complete inventory of hepatic cell types would also illuminate potential hepatic transcriptional zonation (gene expression differences in the same cell type along an anatomical axis) for cell types other than hepatocytes and sinusoidal endothelial cells, for which zonation has been described along the portal-central axis, as well as for HSC, for which zonation has been suggested (Dobie et al, 2019b; Duan et al, 2022; Guilliams and Scott, 2022; Halpern et al, 2018; Halpern et al, 2017; Inverso et al, 2021; Krenkel et al, 2019; Paris and Henderson, 2022; Rosenthal et al, 2021). The gradual or punctuated transcriptomic/phenotypic variation of other mesenchymal hepatic cell types along spatial axes remains less explored.

To address these questions, we have generated a comprehensive molecular atlas of the various parenchymal and mesenchymal cell types of the adult mouse liver, utilizing a deep scRNA-seq approach. This, supported with immunofluorescence (IF) and in situ RNA hybridization (ISH), helped us to establish genome-wide transcriptomic profiles for the major hepatic cell types. It also identified several hitherto poorly characterized cell type subpopulations in the liver, elucidating the zonation principles for HSC, vascular smooth muscle (mural) cells, and fibroblasts. With this transcriptomic and anatomical information at hand, we reevaluate data from several previously published hepatic scRNA-seq studies, encompassing normal and diseased livers from both mice and humans, offering new hypothesis about disease-regulated gene expression patterns in distinct mesenchymal cell subpopulations.

# Results

## Transcriptional profiling of the principal cell types of the adult mouse liver

To obtain single-cell transcriptomic information from the major cell types in the adult mouse liver, we employed a combination of transgenic reporter mice and antibody-based cell enrichment strategies, as schematically depicted in Fig. 1A. Cholangiocytes were isolated using antibodies against EPCAM (epithelial cell adhesion molecule, CD326), while hepatocytes could be captured in sufficient numbers without enrichment, due to their abundance in liver tissue. Endothelial cells were isolated by immunopanning with antibodies against PECAM1 (platelet/endothelial cell adhesion molecule 1, CD31) and/or VE-cadherin (cadherin 5, CDH5/CD144). Kupffer cells were collected using an antibody against CD68. Transgenic reporter mice expressing GFP from the promoter constructs from *Pdgfrb* (*Pdgfrb$^{GFP}$*), *Pdgfra* (*Pdgfra$^{H2bGFP}$*), or *Acta2* (*Acta2$^{GFP}$*) were utilized to enrich for different mesenchymal cell populations, as previously described (Muhl et al, 2020; Muhl et al, 2022b). In total, we analyzed the transcriptomes of 3491 cells derived from livers of adult C57Bl6 mice of both sexes (Appendix Fig. S1A). The Smart-seq2 protocol was employed to achieve the deepest possible mRNA sequence capture from each individual cell (Picelli et al, 2014) (Fig. 1A).

A first round of analysis of all captured cells using the Seurat analysis pipeline (Satija et al, 2015; Stuart et al, 2019), combined with cluster definitions based on the pagoda2 algorithm (Fan et al, 2016), revealed 14 distinct clusters (Fig. 1B). By assessing the expression of widely accepted canonical marker genes for specific cell types and classes (for example, for parenchymal/epithelial cells: *Krt18*, *Alb*, *Cdh1*; endothelial cells: *Pecam1*, immune cells: *Ptprc* [CD45], mesenchymal cells: *Pdgfra*, *Pdgfrb*) (Figs. 1C and EV1A), we grouped these 14 clusters into four distinct cell classes encompassing parenchymal cells, endothelial cells, immune cells, and mesenchymal cells (Fig. 1D). To obtain a more granular view of each cell class, we separately analyzed and further annotated them (Fig. 1E–L) as outlined in detail in the following chapters. The parenchymal cells split into five distinguishable clusters comprising one cluster of cholangiocytes and four clusters of hepatocytes (Fig. 1E,F). Endothelial cells separated into eight clusters, comprising subtypes originating from the portal vein, capillary sinusoids, central vein and lymphatics (Fig. 1G,H). Immune cells distributed into seven clusters, representing B-cells, myeloid monocytic cells, and Kupffer cells/macrophages (Fig. 1I,J). Mesenchymal cells formed ten clusters, encompassing different types of fibroblasts, HSC, and vascular smooth muscle cells (SMC) (Fig. 1K,L).

We constructed a searchable database including UMAP visualization of the complete dataset, as well as the four separately analyzed datasets, along with the barplot-visualization of all cells in the dataset ordered by cell class and clusters (Appendix Fig. S1B–D). This database can be explored cell-by-cell and gene-by-gene at https://muhldatahub.org/Publications/LiverScRNAseq/database.html.

## Parenchymal cells

Hepatocytes are organized as stacks of repetitive cellular layers within the liver lobules. Within each layer, hepatocyte zonation along the portal-central axis was reported based on morphological

observations (Asada-Kubota and Kanamura, 1981; Rappaport and Potvin, 1963) and subsequently confirmed by scRNA-seq and in situ RNA and protein analysis (Aizarani et al, 2019; Halpern et al, 2017). Our data identified four hepatocyte clusters (cluster #p1, p3, p4, p5) (Fig. EV1B), corroborating earlier reports of a transcriptomic hepatocyte zonation along the portal-central axis and further revealed a distinct hepatocyte population that exhibits distinct expression of, for example, *Aqp9* which was recently reported as important for liver regeneration (Zhang et al, 2022) (Fig. EV1B–D; Appendix Fig. S2A).

Cholangiocytes constitute the second major parenchymal cell type in the liver (Banales et al, 2019). Somewhat unexpectedly, we identified only one cholangiocyte cluster (cluster #p2) (Fig. EV1B), despite the previously reported heterogeneity in cholangiocyte cell size (Fukushima and Ueno, 2006; Li et al, 2023; Tabibian et al, 2013; Tulasi et al, 2021; Ueno et al, 2003). Cholangiocyte-enriched transcripts encoded e.g. multiple claudins, cytokeratins, solute carrier transporters, and metabolic enzymes (Fig. EV1E,F; Appendix Fig. S2B).

Our present analysis largely confirms the previously described heterogeneity and zonation of the hepatic parenchyma (Halpern et al, 2017). We therefore focused our analysis on hepatic stromal cells, as detailed in the following chapters.

## Endothelial cells

Like hepatocytes, sinusoidal endothelial cells have been reported to exhibit molecular zonation along the lobular portal-central axis (Duan et al, 2022; Halpern et al, 2018; Inverso et al, 2021). Our analysis of the 1106 hepatic endothelial cell transcriptomes identified eight distinguishable clusters confirming the previously reported zonation. Annotation based on commonly accepted markers, such as *Ly6a*, *Cd34* for portal vein endothelial cells, *Bmp2*, *Mrc1*, *Gpihbp1*, *Aqp1* for sinusoidal endothelial cells, and *Rspo3*, *Wnt9b*, *Wnt2* for central vein endothelial cells (Gomez-Salinero et al, 2022; Halpern et al, 2018) (Fig. 1H), we assigned one cluster (cluster #e2) as portal vein endothelial cells, four clusters (clusters #e3, e5, e6 and e8) as sinusoidal endothelial cells, and two clusters (clusters #e1 and e7) as central vein endothelial cells (Fig. 2A,B). The cluster of portal vein endothelial cells could be further split into two clusters (#e2c and e5c) using the pagoda2 *community* setting (Figs. 2A,C and EV2A, see 'Methods' for details). Endothelial cell cluster #e4/e4c represents lymphatic endothelial cells, as shown by the expression of canonical markers (Ulvmar and Makinen, 2016), including *Prox1*, *Mmrn1*, *Pdpn*, *Thy1*, and *Ccl21a* (Fig. EV2B).

Differential expressed gene (DEG) analysis revealed further zone-specific gene expression patterns. Transcripts for e.g., *Adgrg6*, *Vegfc*, *Cyp1a1* (encoding a cytochrome P450 family member monooxygenase, also known as aryl hydrocarbon hydroxylase, AHH), *Gja5*, and *Jag1* were enriched in portal vein endothelial cells. *Stab2*, *Bmp2*, *Gatm*, and *Plxnc1*, among others, were enriched in sinusoidal endothelial cells, and *Bmp4*, *Cyp1b1*, and *Selp* (encoding P-selectin) were enriched in central vein endothelial cells (Figs. 2D and EV2C). ISH for *Adgrg6* (encoding the adhesion G protein-coupled receptor G6) and *Cyp1a1* (suggested to protect against metabolic hepatic diseases) (Uno et al, 2018), confirmed that cells in cluster #e2c originate from the portal vein, rather than the hepatic artery (Figs. 2E,F and EV2D). We next analyzed the

localization of *Gja5* (encoding the gap junction protein connexin 40), a known marker for arterial endothelial cells (Bastide et al, 1993) (Fig. 2G). Analysis of the *Gja5*$^{GFP}$ reporter mice (Miquerol et al, 2004) revealed a strong GFP signal in both hepatic artery and portal vein endothelial cells, but not in the sinusoids or central veins (Fig. 2H). ISH confirmed the presence of *Gja5* mRNA in portal vein endothelial cells (Figs. 2I and EV2E). Because *Gja5* is an established marker of arterial endothelial cells in many organs, its expression along with *Vegfc* (another arterial endothelial marker) in the portal vein suggests that these cells have a mixed phenotype, including expression of both venous and arterial marker genes. ISH for the arterial endothelial cell marker *Sema3g* (Kutschera et al, 2011) showed high expression in endothelial cells lining hepatic arteries and peribiliary arterioles, but not portal vein endothelial cells (Figs. 2I and EV2E). The transcript for *Sema3g* was not found in our scRNA-seq database, suggesting that hepatic artery endothelial cells may not have been captured for scRNA-seq in our experiments. ISH confirmed the specific expression of *Rspo3* by endothelial cells in and around the central vein, but not at the portal tract (Fig. 2J,K). IF analysis for KIT and LYVE1 revealed expression in sinusoidal endothelial cells, with KIT being localized pericentrally and LYVE1 exhibiting mid-sinusoidal expression (Figs. 2L,M and EV2F). In addition to the zonal endothelial cell gene expression pattern along the portal-central axis, a small cluster, #e5c, identified by the community setting of pagoda2, mapped between the portal and central vein but distant from the sinusoidal endothelial cells in the UMAP display (Fig. EV2A). Cells in cluster #e5c are characterized by expression of *Cd200*, a suggested regulator of immune and inflammatory processes (Choe and Choi, 2023), *Aplnr*, *Rgcc* (encoding the regulator of cell cycle), *Cxcl12*, and *Car4* (encoding the carbonic anhydrase 4) previously shown with specific expression in a subset of lung capillary endothelial cells (Gillich et al, 2020) (Figs. 2N and EV2G). IF for CD200 and ISH for *Aplnr* showed expression in capillaries and arterioles (CD200) of the peribiliary vascular plexus (Fig. 2O,P), suggesting that the cells in the cluster #e5c likely originate from the PBV. Transcripts for *Lrg1*, *Ephx1*, and the two elastic fiber constituents *Fbln5* (encoding fibulin 5), and *Eln* (encoding elastin) were enriched in both portal and central vein endothelium, in contrast to *Cxcl10* and *Cyp4b1*, which were enriched in sinusoidal endothelium (Fig. EV2G).

GO analysis of endothelial zone-enriched genes identified terms such as 'blood-vessel morphogenesis', 'regulation of blood pressure', and 'response to toxic substance' associated with portal vein endothelial cells, and terms including 'cell-cell adhesion' and 'neutrophil homeostasis' with sinusoidal endothelial cells. Terms related to 'cell migration', 'connective tissue development', and 'angiogenesis' were associated with central vein endothelial cells (Appendix Fig. S3A).

To further confirm and validate our observations, particularly the hybrid phenotype of portal vein endothelial cells and identity of PBV endothelial cells, we expanded the search for molecular characteristics of endothelial cell subpopulations by analyzing previously published scRNA-seq datasets from human liver samples (Fig. EV2H,I; Appendix Fig. S3B,C) (Buonomo et al, 2022b; Ramachandran et al, 2019b). This corroborated the hybrid gene expression in portal vein endothelial cells (cluster #12 in Ramachandran et al and cluster #4 in Buonomo et al) and could show that these cells are different from arterial endothelial cells

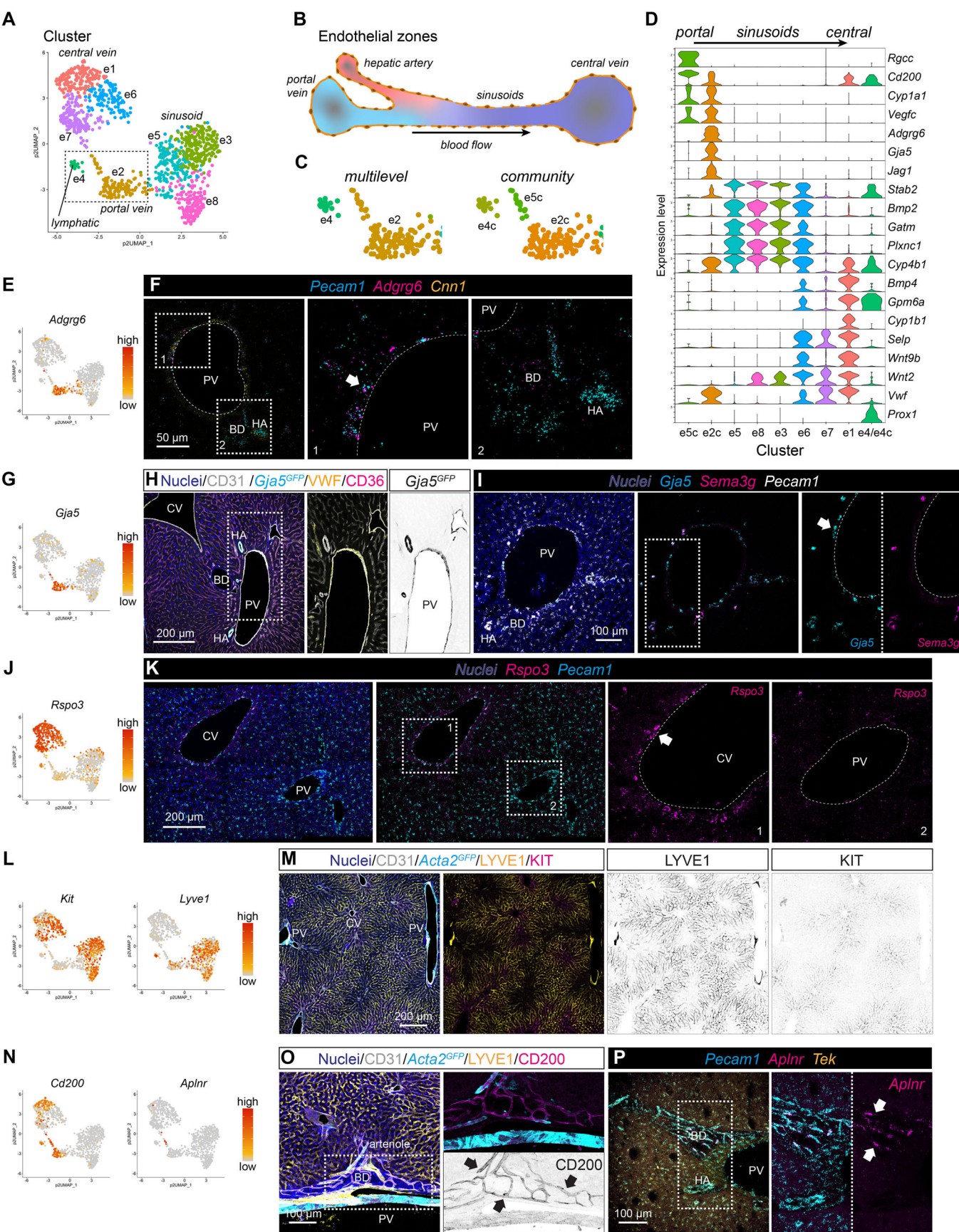

**Figure 2. Endothelial cell subset analysis.**

(A) UMAP visualization of pagoda2 clustering result of the endothelial cell dataset. (B) Schematic depiction of the intra-hepatic vasculature. (C) Magnified section of the UMAP landscape (indicated in A) showing the pagoda2 clustering result using the multilevel (left panel) or community (right panel) setting, respectively. (D) Violin plot showing exemplary genes with differential expression in endothelial cells along the portal-central axis (*n* refers to cells: e5c = 14, e2c = 114, e5 = 151, e8 = 155, e3 = 236, e6 = 112, e7 = 150, e1 = 152, e4/e4c = 22). (E) UMAP visualization of the expression levels of *Adgrg6*. (F) ISH for *Pecam1*, *Adgrg6*, and *Cnn1* on a liver tissue section. Arrow highlights *Pecam1 Adgrg6* double-positive portal vein endothelial cells. (G) UMAP visualization of the expression level of *Gja5*. (H) IF for CD31, VWF, and CD36 on a liver tissue section from a *Gja5^GFP* reporter mouse. (I) ISH for *Gja5*, *Sema3g*, and *Pecam1* on a liver tissue section. Arrow highlights *Gja5* positive portal vein endothelial cells. (J) UMAP visualization of the expression levels of *Rspo3*. (K) ISH for *Rspo3* and *Pecam1* on a liver tissue section. Arrow highlights *Rspo3* positive central vein endothelial cells. (L) UMAP visualization of the expression levels of *Kit* and *Lyve1*. (M) IF for CD31, LYVE1, and KIT on a liver tissue section from an *Acta2^GFP* reporter mouse. (N) UMAP visualization of the expression levels of *Cd200* and *Aplnr*. (O) IF for CD200, CD31, and LYVE1 on a liver tissue section from an *Acta2^GFP* reporter mouse. Arrows highlight CD200 positive endothelial cells of the peribiliary vasculature. (P) ISH for *Pecam1*, *Aplnr*, and *Tek* (Tie2) on a liver tissue section. Arrows highlight *Aplnr* positive endothelial cells of the peribiliary vasculature. PV portal vein, CV central vein, HA hepatic artery, BD bile duct. Scale bars are indicated in the respective image panels. Source data are available online for this figure.

marked by *SEMA3G* expression (cluster #7 and cluster #3, respectively) (Fig. EV2J). Further, analysis of the same human liver datasets also confirmed the presence of the *CD200*, *RGCC*, *CA4* (human homologue to Car4), *APLNR* positive PBV endothelial cell subpopulation (within cluster #0 in both datasets) (Fig. EV2K).

## Immune cells

The liver contains many types of immune cells and at least two major types of macrophages: tissue-resident macrophages (Kupffer cells), and monocyte-derived macrophages (MoMFs) (Guillot and Tacke, 2019; Heymann and Tacke, 2016; Liu et al, 2024; Su et al, 2021). Kupffer cells constitute the largest population of tissue-resident macrophages in the human body and localize across the liver lobule, patrolling/interfacing both with the sinusoidal lumen and the space of Disse (Kubes and Jenne, 2018). Kupffer cells are established in the early embryo and are maintained in adulthood independently of bone-marrow derived monocytes (Yona et al, 2013). In contrast, MoMFs, as their name suggests, derive from monocytes. They are found in smaller numbers compared to Kupffer cells and primarily localize near the portal tract (English et al, 2022; Guilliams and Scott, 2022).

Analysis of 651 transcriptomes from immune cells revealed that most of them (528 cells) represented Kupffer cells (clusters #i1, i3, i4, i6, i7), as determined by the expression of *Clec4f*, *Folr2*, and *Vsig4* (Guilliams et al, 2022). Additionally, we identified *Cd79a*-positive B-cells (cluster #i5), as well as a cluster (#i2) of myeloid monocytic cells (Fig. 3A–C; Appendix Fig. S4A). Notably, some of the immune cells may originate from blood, since no organ perfusion was performed before tissue harvest for scRNA-seq analysis. Using the pagoda2 community setting, we further subdivided cluster #i2 into three subclusters #i2c, i4c, i7c (Fig. 3B) characterized by the expression of *Siglech* and *Ccr9* markers for dendritic cells (cluster #i7c), *Cd9* and *Cx3cr1* markers for MoMF (cluster #i2c), and *Ddx60* for a not further defined subpopulation of Kupffer cells (cluster #i4c) (Fig. 3C). Next, we used IF analysis to distinguish Kupffer cells from MoMFs (Fig. 3D). Cells double-positive for CD68 and CLEC4F (Kupffer cells) were found along the sinusoidal capillaries, whereas CD68-positive and CLEC4F-negative cells (MoMFs) resided near the portal tract (Fig. 3D), supporting the previously proposed anatomical distribution of these cell types (English et al, 2022). The separation of the Kupffer cells into five subclusters did not reveal qualitative marker differences between the putative Kupffer cell subtypes, except for a signature associated with cell cycle progression (*Mki67*, *Top2a*,

*Ccna2*, *Ccnb1*) in cluster #i4/i5c, and the contamination by endothelial cell transcriptomes (e.g. *Tie1*, *Kdr*, *Ptprb*) in subcluster #i9c (Appendix Fig. S4A,B). Although earlier studies identified two distinct Kupffer cell populations (designated KC1 and KC2) based on differential expression levels of *Mrc1* (encoding the C-type 1 mannose receptor CD206), *Esam*, and *Cdh5* (the two latter also known as endothelial cell markers) (Bleriot et al, 2021; De Simone et al, 2021), all Kupffer cells in our dataset exhibited high expression of *Mrc1* and *Cdh5* while being negative for *Esam* (Appendix Fig. S4B). Because our data did not confirm the previously described KC1/KC2 subtypes and further suggested endothelial cell contamination as a putative driver of Kupffer cell sub-clustering (Appendix Fig. S4B) (Guilliams and Scott, 2022; Hume et al, 2022), we removed the cells corresponding to cluster #i5c (proliferating Kupffer cells) and i9c (Kupffer cells contaminated by endothelial cells) from the consensus Kupffer cell population (containing clusters #i1c, i3c, i8c, i10c, i11c) displayed in the searchable barplot database (Appendix Fig. S1B–D).

Notwithstanding the suspected endothelial cell contamination of a small population of the Kupffer cells, our data demonstrate that Kupffer cells express *Cdh5* (encoding cadherin 5, CDH5, also known as vascular endothelial (VE)-cadherin) at levels comparable to those in hepatic endothelial cells (Fig. 3E). CDH5 mediates homotypic cell-cell binding, thus suggesting a potential junctional interaction through CDH5 between Kupffer cells and sinusoidal endothelial cells (Ito et al, 1980). High-resolution confocal imaging revealed a complex pattern for CDH5 protein localization in sinusoidal endothelial cells depending on their vicinity to Kupffer cells (Fig. 3F; Appendix Fig. S4C).

GO analysis of immune cell population-enriched genes confirmed the identities of B-cells and dendritic cells. Additionally, the GO analysis revealed that the terms 'inflammatory response' and 'monocyte chemotaxis' were associated with both MoMF and Kupffer cell-enriched genes, suggesting the expression of distinct inflammatory gene expression programs in the two cell types (Appendix Fig. S4D).

## Mesenchymal cells

The liver is known to contain multiple mesenchymal cell populations at distinct anatomical locations. HSC, which partially resemble pericytes found in other organs, populate the perisinu-soidal space. Vascular SMC and other vascular mural cells reside in the vessel wall of hepatic arteries, arterioles, portal veins, and central veins. Fibroblasts reside in the portal tract and at the liver

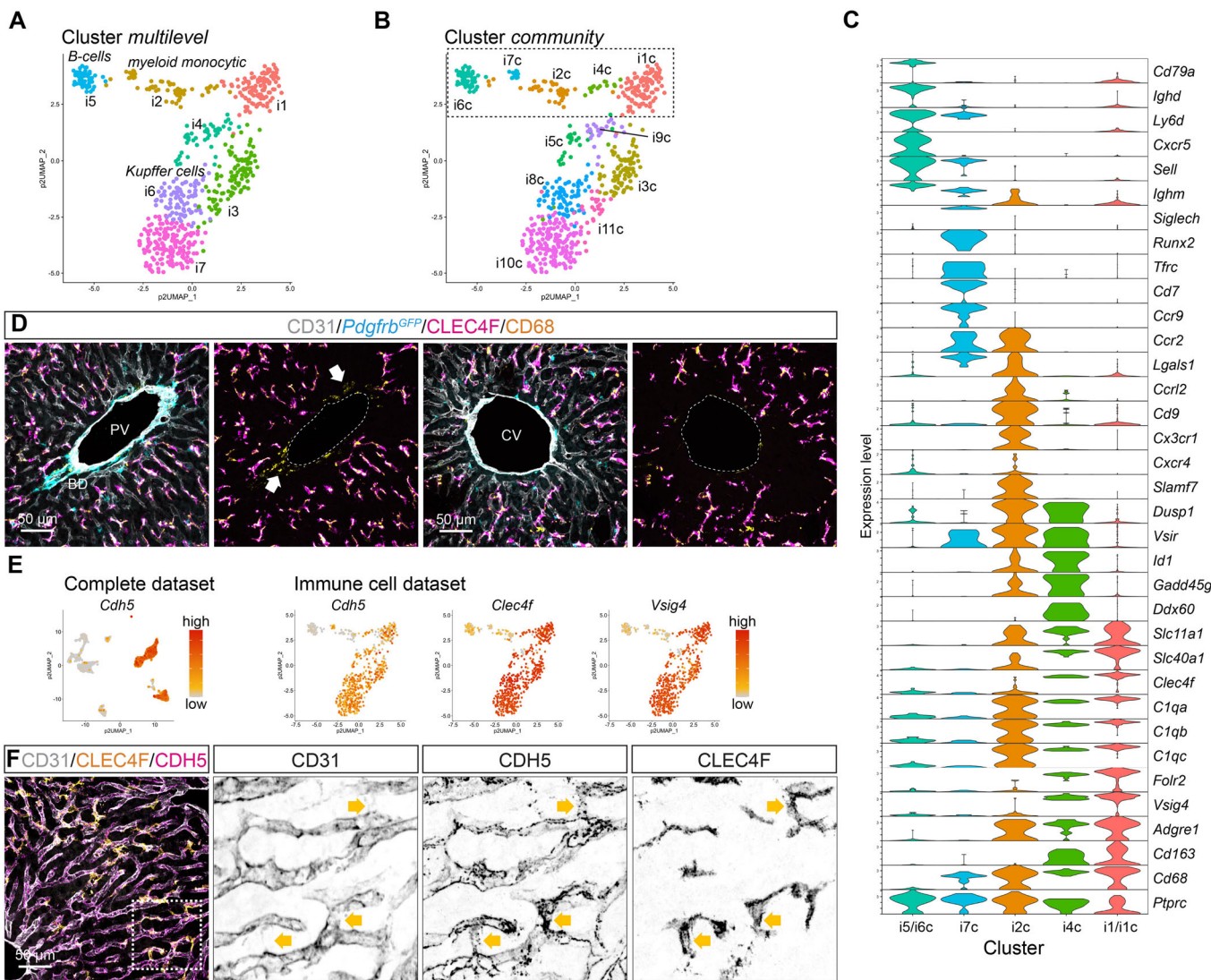

**Figure 3. Immune cell subset analysis.**

(A, B) UMAP visualization of pagoda2 clustering result of the immune cell dataset using the multilevel (A) or community (B) setting. (C) Violin plot showing the expression of enriched exemplary genes in the clusters obtained with the community setting (highlighted in B, n refers to cells: i5/i6c = 24, i7c = 16, i2c = 41, i4c = 18, i1/i1c = 111). (D) IF for CD31, CLEC4F, and CD68 on a liver tissue section from a *Pdgfra^H2bGFP* reporter mouse. The arrows highlight CLEC4F-negative, CD68-positive macrophages. (E) UMAP visualization of the expression level of *Cdh5* in the complete dataset (left panel), or *Cdh5, Clec4f*, and *Vsig4* in the immune cell dataset (right panel). (F) IF for CD31, CDH5, and CLEC4F on a liver tissue section. The boxed area is shown magnified to the right for each antibody stain. Arrows highlight CDH5 signals colocalizing with CLEC4F. PV portal vein, BD bile duct, CV central vein. Scale bars are indicated in the respective image panels. Source data are available online for this figure.

capsule (Bhunchet and Wake, 1992; Wells, 2014a). In our mesenchymal cell dataset, transcriptomes from 1337 cells from adult mice were classified into ten clusters using the pagoda2 multilevel algorithm (Figs. 4A and EV3A). HSC, vascular mural cells, and fibroblasts, identified by canonical markers (Dobie et al, 2019b; Muhl et al, 2020; Muhl et al, 2022b) (Fig. 1L), were allocated to separate cell clusters within the UMAP display. The distribution suggested heterogeneity not only between, but also within, these cell classes, particularly among the fibroblasts and mural cells (Fig. 4A). In the following sections, we detail the mesenchymal cell diversity in relation to their anatomical locations.

**Cells at the liver capsule**

The liver capsule is known to be covered by an outer monolayer of mesothelial cells (Li et al, 2013). In-depth analysis of the mesenchymal cell dataset revealed a small group of mesothelial cells, separated by pagoda2 community clustering (cluster #m16c), and characterized by the expression of *Msln*, *Gpm6a* and *Krt7* (Figs. 4B and EV3A,B). These cells also expressed the mesothelial cell marker Wilms Tumor 1 transcription factor (encoded by *Wt1*). Notably, *Wt1* was also expressed by subpopulations of fibroblasts and HSC (Figs. 4C,D and EV3B). Tissue analysis revealed *Wt1*/WT1-expressing cells located at the capsule and sub-capsular

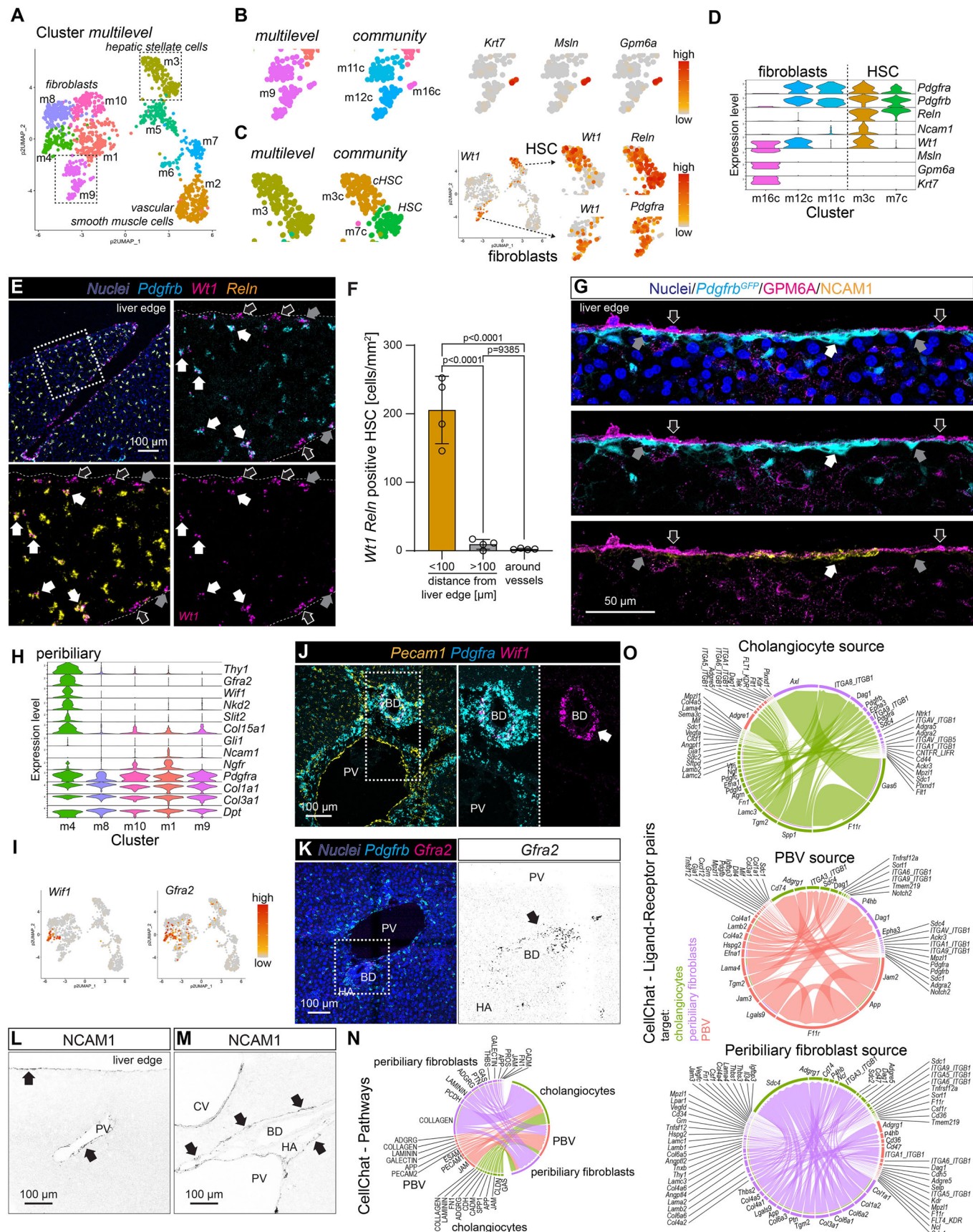

**Figure 4.  Analysis of mesenchymal cell populations at the liver capsule and portal tract.**

(A) UMAP visualization of the clustering result for the mesenchymal cell subset. (B) Magnified section of UMAP landscape containing cluster #m9 (multilevel), or clusters #m11c, m12c, and m16c (community), respectively. To the right, UMAP visualization of expression levels of *Krt7*, *Msln*, and *Gpm6a*. (C) Magnified section of UMAP visualization containing cluster #m3 (multilevel), or clusters #m3c, m7c (community), respectively. To the right, UMAP visualization of the expression level of *Wt1*, *Reln*, and *Pdgfra*. (D) Violin plot showing the expression of exemplary genes in selected fibroblast and HSC clusters (n refers to cells: m16c = 6, m12c = 60, m11c = 53, m3c = 113, m7c = 72). (E) ISH for *Pdgfrb*, *Wt1*, and *Reln* on a liver tissue section, focusing on the edge of the tissue. White arrows indicate triple-positive cells, grey arrows indicate *Wt1 Pdgfrb* double-positive cells, and open arrows indicate *Wt1* single-positive cells. (F) Quantification of *Wt1 Reln* double-positive cells at different anatomic locations within the liver (biological replicates n = 4, P values were calculated using one-way ANOVA and are given in the graph), error bars show s.d. (G) 3D-redering (angled to achieve 90° view of outer cell layers) of IF for GPM6A and NCAM1 on a liver tissue section from a *Pdgfrb*ᴳᶠᴾ reporter mouse. The arrows indicate the mesothelial cell layer marked by GPM6A and the underlying fibroblast and cHSC layer indicated by *Pdgfrb*ᴳᶠᴾ and NCAM1 (compare to E). (H) Violin plot showing the expression of genes with enriched expression in the cells from cluster #m4 (*Thy1*, *Gfra2*, *Wif1*, *Nkd2*), or cluster #m1 (*Ncam1*, *Ngfr*) of the mesenchymal cell subset, together with general fibroblast markers (n refers to cells: m4 = 98, m8 = 156, m10 = 127, m1 = 181, m9 = 119). (I) UMAP visualization of the expression levels of *Wif1* and *Gfra2*. (J) ISH for *Epcam*, *Pdgfra*, and *Wif1* on a liver tissue section. The arrow highlights *Wif1*-positive peribiliary fibroblasts. (K) ISH for *Pdgfrb* and *Gfra2* on a liver tissue section. The arrow indicates *Gfra2* expression at the bile duct. (L, M) IF for NCAM1 on a liver tissue section, focusing on the edge of the liver (L) or the portal tract (M). Arrows indicate NCAM1-positive structures. (N) Chord diagram showing interacting pathways from CellChat cell-cell communication analysis between cells located in the biliary niche: cholangiocytes, peribiliary fibroblasts, and PBV endothelial cells. (O) Chord diagrams showing the interacting ligand-receptors pairs from CellChat cell-cell communication analysis with (top to bottom) cholangiocytes, PBV endothelial cells, or peribiliary fibroblasts as source. PV portal vein, BD bile duct, HA hepatic artery, CV central vein, PBV peribiliary vasculature. Scale bars are indicated in the respective image panels. Source data are available online for this figure.

region, partially overlapping with expression of *Pdgfrb*, *Pdgfra*ᴴ²ᵇᴳᶠᴾ, and *Reln* (Figs. 4E–G and EV3C). This suggests the presence of heterogenous fibroblast and HSC subpopulations contributing to the formation of the capsule and sub-capsular region. However, we could also show occasional cells double-positive for WT1 and *Pdgfra*ᴴ²ᵇᴳᶠᴾ in the wall of large central veins (Fig. EV3D).

Capsular *Wt1+* fibroblasts (clusters #m11c, m12c) expressed *Pdgfra* and *Pdgfrb* but not *Reln*, while capsular HSC (cHSC, cluster #m3c) expressed *Reln* and *Ncam1* (encoding the neural cell adhesion molecule 1) in addition to *Wt1* (Fig. 4D,G). To further explore the molecular phenotype of *Wt1+* fibroblasts and cHSC, we performed DEG analysis comparing the capsular cells to other liver fibroblast or HSC populations, respectively. *Wt1+* fibroblasts displayed distinct expression of genes, such as *Scara5*, *Fbln2*, *Adgrd1*, and *Osr1* (encoding the odd-skipped related transcription factor 1) that has been linked to liver fibrosis (Nian et al, 2024), and mesenchymal collagen production (Murugapoopathy et al, 2021) (Fig. EV3E). In contrast, cHSC expressed many conventional fibroblast markers (Buechler et al, 2021; Muhl et al, 2020), including *Gpx3*, *Mfap4*, *Fmod*, and *Dpt*, but also genes not normally associated with fibroblasts, such as *Alcam* (encoding the activated leukocyte cell adhesion molecule) (Fig. EV4F), which is also expressed by cholangiocytes.

GO analysis of overrepresented genes in *Wt1+* fibroblasts or cHSC revealed terms related to 'extracellular matrix' and 'encapsulating' for both populations, as well as 'chemotaxis' specifically for cHSC (Appendix Fig. S5A). This suggests their active contribution to the formation of the liver capsule and potential roles as progenitors of HSC.

### Cells at the portal tract and central vein

The portal tract contains the most complex composition of mesenchymal cell types found in the liver. These mesenchymal cells are associated with hepatic arteries, arterioles, portal veins, and bile ducts. We identified a fibroblast population residing in the peribiliary niche (cluster #m4) (Fig. 4A), characterized by a distinct molecular profile that includes the expression of *Thy1* (encoding CD90), a commonly used fibroblast marker, as well as *Col15a1*, *Slit2*, *Gli1* (Gupta et al, 2020; Lei et al, 2022), *Gfra2*, *Nkd2* (encoding the naked cuticle 2), and *Wif1* (encoding the Wnt

inhibitory factor 1), the latter two reported as Wnt-pathway antagonists (Gammons et al, 2020; Poggi et al, 2018) (Fig. 4H,I). The expression of *Wif1*, *Gfra2*, and *Pi16* at bile ducts with multilayered mesenchymal stromal cells was confirmed by ISH (Figs. 4J,K and EV3G), in agreement with earlier reports (Gupta et al, 2020; Lei et al, 2022). *Wif1* signal was also observed around central veins, although not colocalizing with *Pdgfra*, and therefore likely originating from perivenous hepatocytes (Fig. EV3H). GO analysis of genes overrepresented in peribiliary fibroblasts (Fig. EV3I) revealed terms related to 'morphogenesis' and 'extracellular matrix' (Appendix Fig. S5A). Of note, IF for NCAM1 marked cHSC, but also a subset of portal fibroblasts (cluster #m1) that forms a capsule-like structure lining the portal tract (Fig. 4L,M). This suggests the presence of heterogenous fibroblast subpopulations contributing to the formation of the portal tract connective tissue and capsule (Glisson's capsule).

Intrigued by the identification of several specific cell types that occupy the peri-biliary niche; cholangiocytes, PBV, and peribiliary fibroblasts, we performed a cell-cell communication analysis using CellChat (Jin et al, 2025). Probing for communication pathways between the three cell types of the bile duct-niche showed extracellular matrix related and cell-cell contact pathways (Fig. 4N). Detailed analysis of the ligand-receptor pairs for paracrine interaction corroborated the presence of extracellular matrix-related interactions (Fig. 4O), suggesting that all three cell types contribute to the microenvironment of this specific niche in homeostasis.

Mural cells were identified by canonical marker gene expression, such as *Acta2*, *Tagln*, *Notch3*, and *Des*, as previously reported (Muhl et al, 2022b) (Figs. 5A–C and EV4A,B). Notably, scRNA-seq analysis identified *Ncam1* expressing cells amongst the fibroblasts likely involved in the formation of the portal capsule (see above), and in mural cell clusters (#m5c and m6/m15c), additionally exhibiting expression of *Cdh3*, encoding the cell-cell adhesion protein cadherin 3 (also known as placental (P)-cadherin), and cluster m6/m15c additionally showed expression of *Npnt* (encoding nephronectin) (Fig. 5D,E). Large and medium caliber muscularized portal veins with a strong and continuous *Acta2*/αSMA signal recurrently showed IF signals for NPNT (Fig. 5F,G). NPNT-positive cells were also observed around large (>500 μm

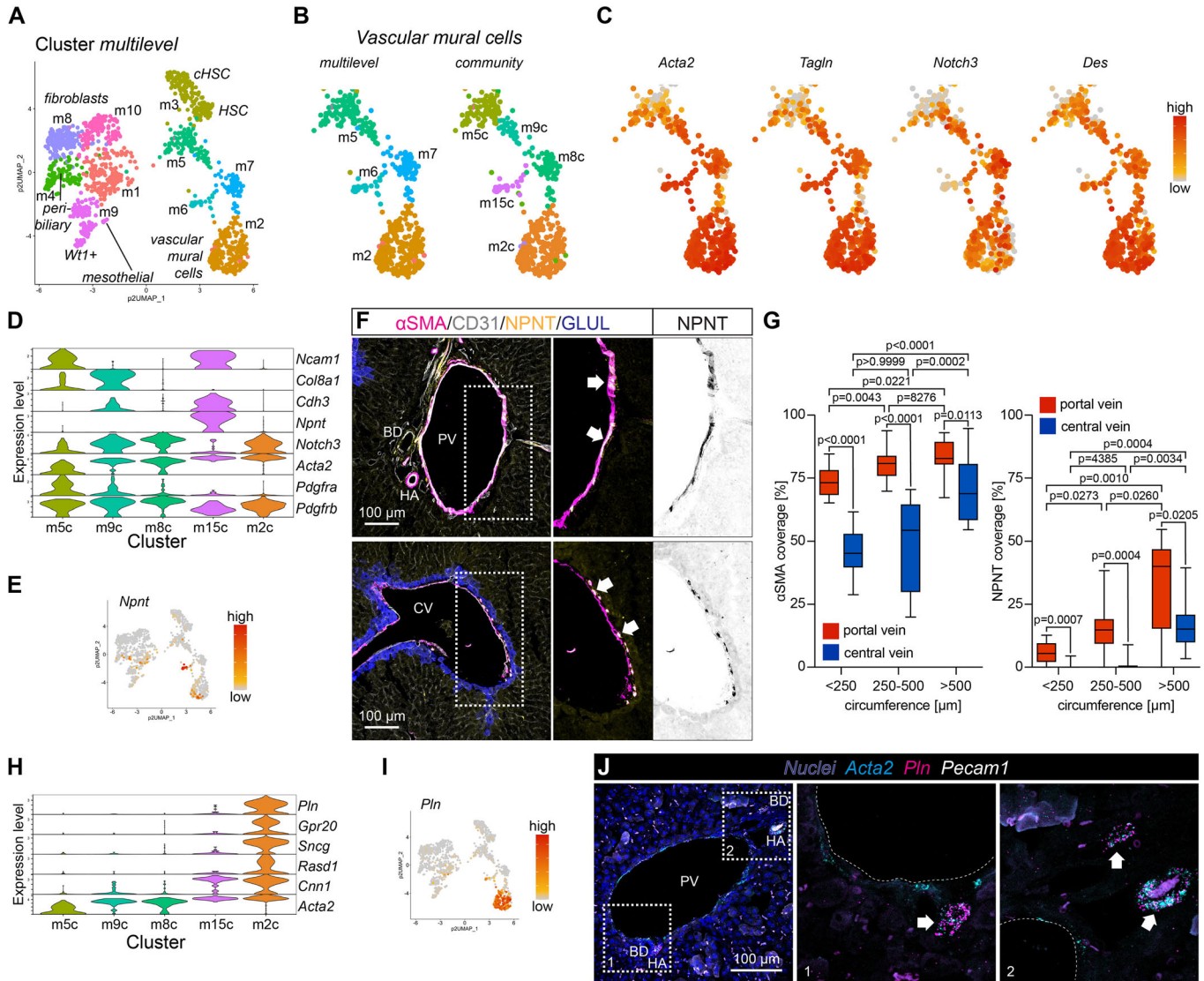

**Figure 5.  Identification of SMC of the elastic and arterial subtype.**

(A) UMAP visualization of the clustering result for the mesenchymal cell subset. (B) Magnified section of UMAP landscape containing vascular mural cell clusters (multilevel #m2, m5, m6, m7; community: #m2c, m5c, m8c, m9c, m15c). (C) UMAP visualization of the expression levels of vascular SMC marker genes (Acta2, Tagln, Notch3, and Des) in the magnified section of the UMAP landscape containing vascular mural cell clusters. (D) Violin plot showing the expression level of exemplary genes differentially expressed between vascular mural cell subpopulations (community clusters, n refers to cells: m5c = 87, m9c = 39, m8c = 72, m15c = 33, m2c = 236). (E) UMAP visualization of the expression level of Npnt. (F) IF for CD31, αSMA, GLUL, and NPNT on a liver tissue section, showing an exemplary portal tract (upper panel) and central vein (lower panel) region. Arrows highlight NPNT-positive SMC. (G) Quantification of portal vein and central vein coverage with αSMA (right panel) and NPNT (right panel) staining (biological replicates n = 11–12, P values were calculated using two-way ANOVA, mixed-effects analysis, with Sídák's multiple comparisons test and are given in the graph), boxes show lower and upper quartile, whiskers show min-to-max range, and horizontal lines indicate median. (H) Violin plot showing the expression level of genes with enriched expression in arterial SMC (cluster #m2/m2c, n refers to cells: m5c = 87, m9c = 39, m8c = 72, m15c = 33, m2c = 236). (I) UMAP visualization of the expression level of Pln. (J) ISH for Acta2, Pln, and Pecam1 on a liver tissue section. The arrows indicate Pln positive arterial SMC. PV portal vein, BD bile duct, HA hepatic artery, CV central vein. Scale bars are indicated in the respective image panels. Source data are available online for this figure.

circumference) central veins, although with lower frequency (Fig. 5F,G), suggesting that cells in cluster #m6/m15c represent a distinct subpopulation of vascular SMC preferentially found in large-diameter veins.

The largest group of vascular SMC (cluster #m2/m2c) was characterized by the expression of Pln (encoding phospholamban), Gpr20, Sncg, Rasd1 and high levels of Cnn1 (encoding calponin 1) (Fig. 5H,I), genes previously reported as arterial SMC markers

(Muhl et al, 2022b). ISH analysis for Pln demonstrated restricted expression in cells of the hepatic artery and larger arterioles around the bile duct (Fig. 5J), confirming that the cells in cluster #m2/m2c are SMC from the hepatic arteries and arterioles of the PBV.

Cluster #m7/m8c was characterized by the expression of Ephx3 (encoding the epoxide hydrolase 3), as well as low levels of Cnn1 (Fig. 6A,B). ISH for Ephx3 revealed specific expression in mural cells of the portal vein and the peribiliary vascular plexus, but not

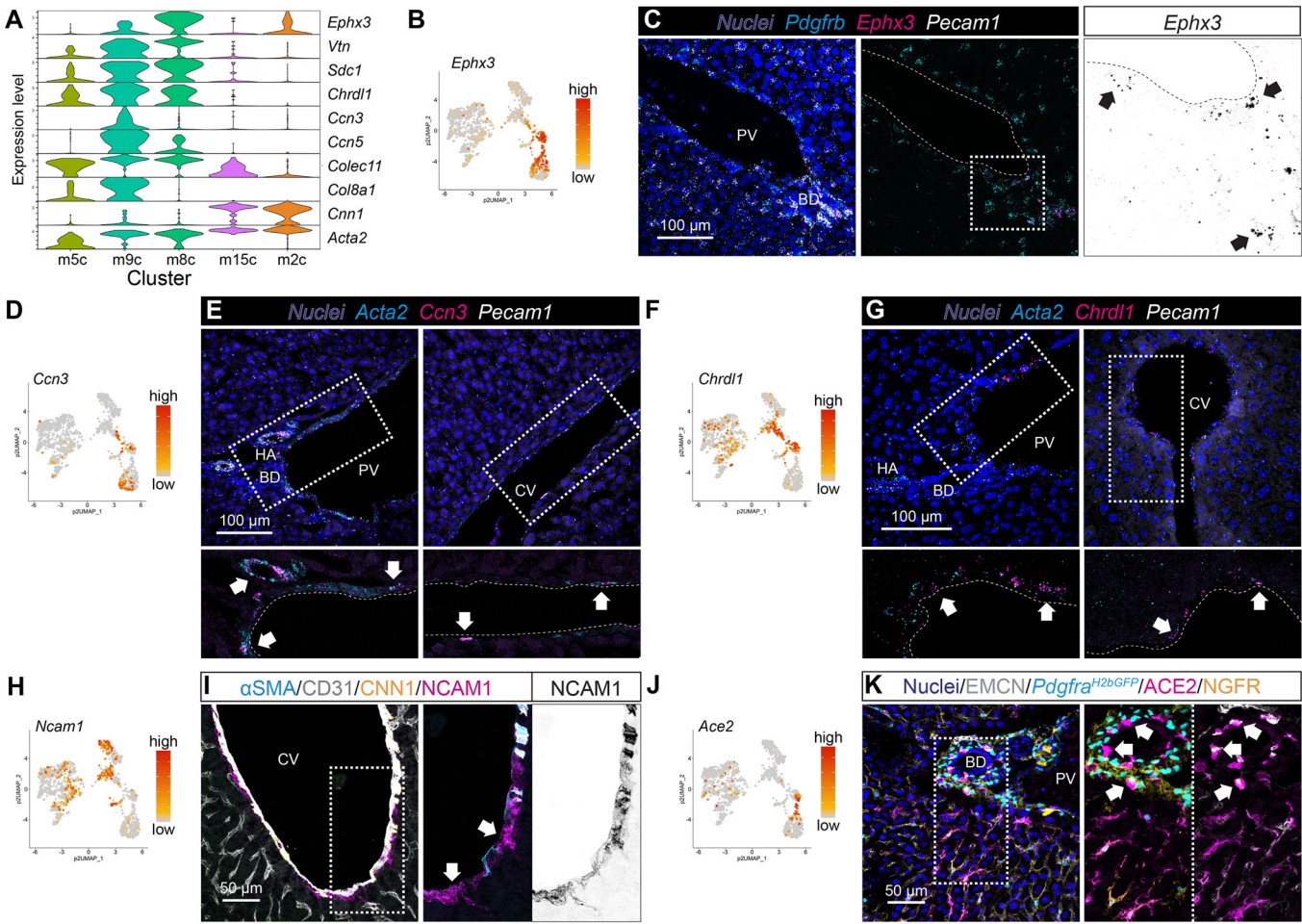

**Figure 6. Identification of venous SMC subtypes and pericytes.**

(A) Violin plot showing the expression level of genes with enriched expression in venous SMC (clusters #m8c, m9c, n refers to cells: m5c = 87, m9c = 39, m8c = 72, m15c = 33, m2c = 236). (B) UMAP visualization of the expression level of *Ephx3*. (C) ISH for *Pdgfrb*, *Ephx3*, and *Pecam1* on a liver tissue section focusing on the portal tract. The arrows highlight expression of *Ephx3* in the portal tract. (D) UMAP visualization of the expression level of *Ccn3*. (E) ISH for *Acta2*, *Ccn3*, and *Pecam1* on a liver tissue section. The arrows indicate *Ccn3*-positive SMC. (F) UMAP visualization of the expression level of *Chrdl1*. (G) ISH for *Acta2*, *Chrdl1*, and *Pecam1* on a liver tissue section. The arrows highlight *Chrdl1*-positive venous SMC. (H) UMAP visualization of the expression level of *Ncam1*. (I) IF for αSMA, CD31, CNN1, and NCAM1 on a liver tissue section. The arrows highlight NCAM1-positive αSMA-negative cells in the wall of a central vein. (J) UMAP visualization of the expression level of *Ace2*. (K) IF for EMCN, ACE2, and NGFR on a liver tissue section from a *Pdgfra^H2bGFP* reporter mouse. The arrows indicate ACE2-positive pericytes located at the peribiliary vasculature. PV portal vein, BD bile duct, HA hepatic artery, CV central vein. Scale bars are indicated in the respective image panels. Source data are available online for this figure.

around the central vein (Figs. 6C and EV4C). We therefore designated the cells in cluster #m7/m8c as portal vein SMC. Cells allocated to cluster #m9c expressed genes previously described as venous SMC markers (Muhl et al, 2022b), including *Ccn3* (encoding the cellular communication network factor 3) (Fig. 6D,E) and *Chrdl1* (encoding chordin-like 1), a BMP-signaling regulator (Fig. 6F,G). ISH confirmed the expression of both transcripts at portal and central veins although at lower levels in the central veins (Fig. 6E,G), possibly due to the overall lover coverage of αSMA-positive cells at central veins (Fig. 5G). This suggests that cells in cluster #m9c represent a subpopulation of venous SMC distributed throughout the hepatic venous system partially sharing the anatomical niche with vascular SMC of cluster #m7/m8c and m6/m15c.

So-called second layer cells (SLC) are morphologically and positionally distinct from HSC and known to reside in the walls of

central veins, occupying the space between the central vein SMC and the nearest sinusoids. The number of SLC declines in the walls of smaller central veins, which have instead been suggested to harbor HSC (Bhunchet and Wake, 1992). Cells in cluster #m5c, placed between venous SMC and HSC in the UMAP landscape, expressed *Ncam1* (Figs. 5D and 6H), and IF for NCAM1 showed staining around central veins (Figs. 6I and EV4D), and as described above, also in large muscularized portal veins, the liver capsule, and the portal tract. The NCAM1 staining pattern at central veins, however, aligns with the expected position of SLC, particularly where there is no overlap with αSMA staining (Figs. 6I and EV4D). We therefore assume that cluster #m5c represents SLC; yet the same staining pattern is also observed around portal veins (Fig. EV4D), perhaps indicating that SLC may occupy both central and portal areas in the mouse liver. By applying a gene signature that identifies fibroblast-like and mural cell-like expression

characteristics (Muhl et al, 2020), we found that the SLC cluster #m5c showed greater similarity to fibroblasts than to mural cells (Fig. EV4E). GO analysis of the different hepatic mural cell populations revealed terms related to 'muscle tissue development', 'serotonin transport' and 'neurotransmitter uptake' associated with large-caliber vein SMC, and terms related to 'muscle cell development' and 'Notch signaling' overrepresented in arterial SMC. Additionally, terms related to 'angiogenesis', 'artery morphogenesis' and 'hepatic stellate cell activation' were identified for portal vein SMC, while terms associated with 'extracellular matrix organization', 'angiogenesis' and 'wound healing' were noted in venous SMC. For SLC, terms related to 'extracellular structure organization', 'positive regulation of locomotion' and 'hormone metabolic process' were associated (Appendix Fig. S5B).

Pericytes did not form their own distinct cluster in our dataset, presumably because of their low number. However, cells placed between arterial SMC (cluster #m2/m2c) and portal vein SMC (cluster #m7/m8c) by UMAP likely represent pericytes of the peribiliary vascular plexus due to their expression of *Ephx3*, *Ace2*, *Cspg4*, and *Higd1b*, while being negative for *Pdgfra* (Figs. 6B,J,K and EV4F,G). Peribiliary pericytes also morphologically resemble pericytes found in other tissues (Armulik et al, 2011; Muhl et al, 2020; Vanlandewijck et al, 2018) (Fig. EV4G).

### Cells at the sinusoids

HSC reside in the space of Disse between sinusoidal endothelial cells and hepatocytes. HSC have previously been characterized by scRNA-seq, and specific marker genes have been proposed for their identification (Dobie et al, 2019b; Filliol et al, 2022; Krenkel et al, 2019; Mederacke et al, 2022b; Yang et al, 2021). With this information as a guide, we initially used the expression of *Reln*, *Lrat*, and *Hgf* to tentatively define HSC clusters (Fig. 1L). To achieve a more comprehensive basis for HSC identification, we analyzed earlier Smart-seq2 data from the GSE137720 dataset (Dobie et al, 2019b) and created a HSC signature comprising 180 genes with enriched expression in HSC (Appendix Fig. S5C,D; Dataset EV1). Application of this HSC signature score to our mesenchymal dataset revealed cluster #m7c as having the highest score, thus likely representing HSC (Figs. 7A–C and EV5A–C). *Ngfr* and *Adamstl2* have been suggested as markers for periportal and pericentral HSC, respectively (Fig. EV5C) (Dobie et al, 2019b); however, our transcriptomic data did not identify HSC diversity, possibly due to the low number of HSC in the dataset (Fig. EV5D). Nevertheless, our IF analysis confirmed the expected distribution of NGFR-positive HSC near the portal tract and mid-lobular region (Fig. EV5D).

Through a series of DEG analysis, comparing the identified HSC to different groups of mesenchymal cells, together with the above generated 180 gene HSC signature, we generated a core HSC profile consisting of 59 genes (Fig. EV5E,F; Dataset EV1). In addition to previously established markers, we found distinct HSC expression of *Plvap* (encoding the plasmalemma vesicle associated protein, PLVAP, a commonly used endothelial cell marker), *Tlcd4* (previously annotated as *Tmem56*), *Fcna* (encoding ficolin a), *Hhip* (encoding the hedgehog-interacting protein), *Pth1r* (encoding the parathyroid hormone 1 receptor), and *Bco1* (encoding the beta-carotene oxygenase 1) (Fig. 7D). IF for PLVAP and RELN confirmed their co-expression in *Pdgfra*$^{H2bGFP}$ positive HSC (Figs. 7E,F and EV5G). This finding was further corroborated by

ISH for *Plvap* and *Pdgfrb* (Fig. EV5H,I). A recent study reported a central role of PLVAP expressed in HSC for hepatic lipid handling and insulin signaling (Hansen et al, 2025). HSC localizing to the pericentral region co-expressed *Rspo3* (a marker for central vein EC) and *Hhip* (Figs. 7G,H and EV5J). GO analysis of the core HSC gene set revealed terms related to 'angiogenesis', 'epithelium morphogenesis', and 'olefinic compound metabolism' associated with HSC (Appendix Fig. S5E).

Notably, we observed that HSC near the portal tract expressed ACE2 (encoding angiotensin converting enzyme 2, also known as a receptor for SARS-CoV-2) as did pericytes of the peribiliary vascular plexus (Fig. 7I–K). Our previous work revealed *Ace2*/ACE2 expression in several but not all populations of pericytes in other organs (Muhl et al, 2022a). The co-expression of *Ace2*/ACE2 with the HSC marker *Reln* (Fig. 7K), alongside the expression of *Pdgfra*$^{H2bGFP}$ (Fig. 7J), and the absence of αSMA expression (Fig. EV5K), suggests that ACE2 marks a subpopulation of periportal HSC, distinct from *Ace2*-positive pericytes of the PBV. Conversely, a HSC subpopulation localized in the pericentral niche was identified by CDH3 staining. Like the periportal HSC subpopulation, the pericentral HSC co-expressed *Pdgfra*$^{H2bGFP}$ (Fig. 7L,M). However, the portal HSC zonation marker NGFR rarely colocalized with CDH3-positive HSC (Fig. EV5L). Although we could identify these two HSC subpopulations through IF/ISH analysis, we concluded that these HSC subpopulations were likely not captured by our scRNA-seq experiments. This conclusion was supported by the GSE137720 dataset (Dobie et al, 2019b), which demonstrated co-expression of *Ace2* with *Ngfr* and *Myh11* in the HSC cluster. However, *Cdh3* was not expressed amongst the HSC cluster (Appendix Fig. S5F), suggesting that this rare pericentral HSC subpopulation was also not captured in the GSE137720 dataset. Recently, a periportal HSC subpopulation positive for *Myh11* (encoding smooth muscle myosin heavy chain, SMMHC) was identified and suggested to participate in capillarization during hepatic fibrosis (Kan et al, 2021), which may reflect the ACE2-positive subpopulation discussed above. Further work is needed to identify the molecular phenotype of the periportal and pericentral HSC subpopulations.

In summary, we present a comprehensive overview of the complex landscape of mesenchymal cells that populate the various anatomical niches of the adult murine liver. We provide molecular characteristics for capsular, peribiliary, and portal fibroblasts, as well as for arterial and different venous SMC populations. Additionally, we highlight the presence of SLC (which may also be present in the portal vein), canonical HSC, capsular HSC (cHSC), and two rare subpopulations of HSC residing near the portal vein or central vein, respectively.

## Relation to published work and relevance for hepatic diseases

It is increasingly appreciated that mesenchymal cells of the liver have a role in hepatic disease progression (Kamm and McCommis, 2022; Wells, 2014a). To gain further insight into the molecular transitions from homeostasis to disease, we incorporated the results from our scRNA-seq analysis with datasets from other sources. To this end, scRNA-seq data from two independent studies of mouse disease models (Dobie et al, 2019b; Mederacke et al, 2022b) were reprocessed and analyzed.

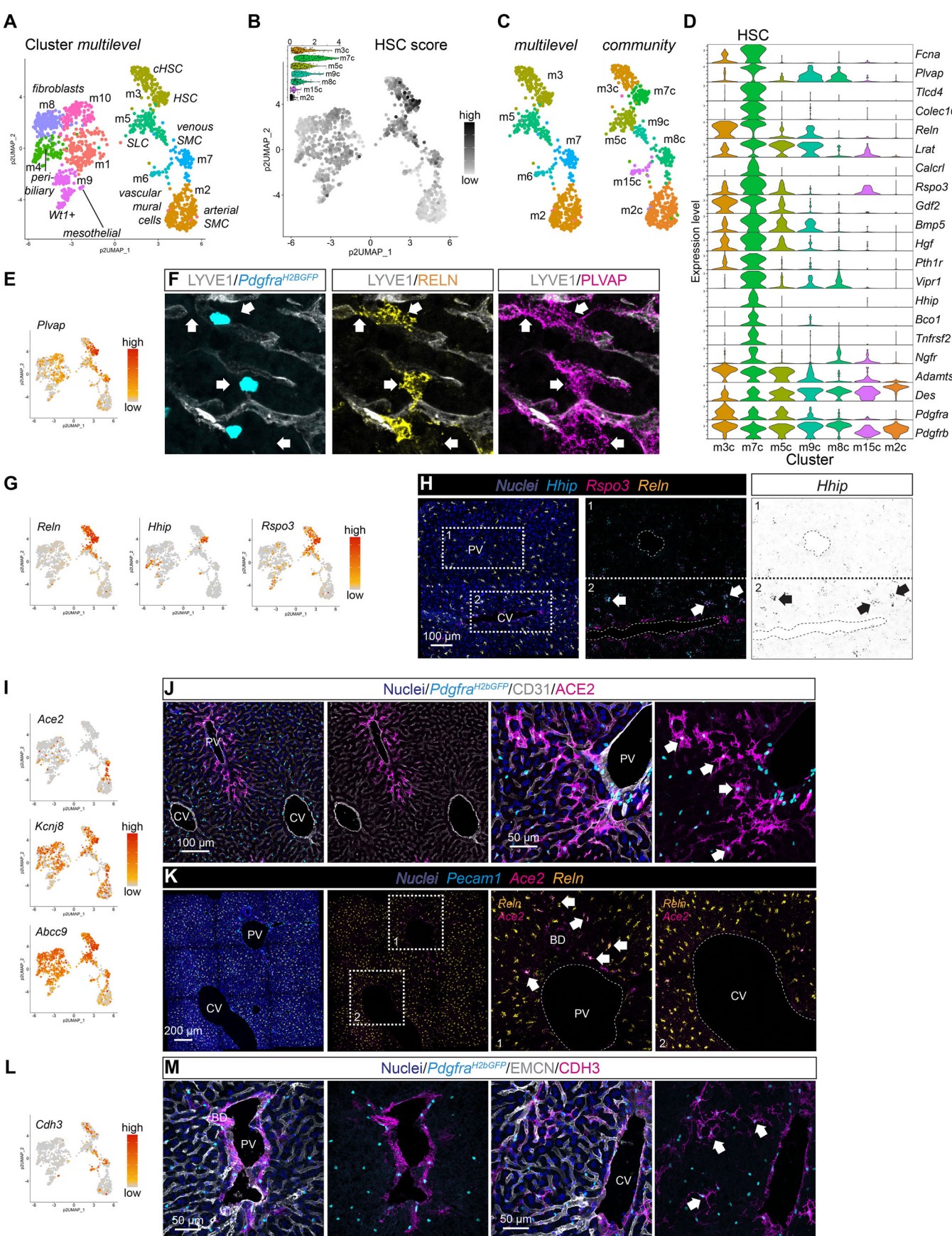

**Figure 7. Analysis of mesenchymal cells located at the sinusoids.**

(A) UMAP visualization of the clustering result for the mesenchymal cell subset. (B) UMAP visualization of the mesenchymal cell subset showing the cumulative expression level of HSC-specific genes (HSC score). The inlayed violin plot shows the individual scores (arbitrary units) of clusters containing HSC and vascular mural cells. (C) UMAP visualization of the magnified section of the UMAP landscape containing HSC and vascular mural cell clusters (multilevel: #m2, m3, m5, m6, m7; community: #m2c, m3c, m5c, m7c, m8c, m9c, m15c). (D) Violin plot showing the expression level of HSC enriched genes in the HSC and vascular mural cell clusters (community, n refers to cells: m3c = 113, m7c = 72, m5c = 87, m9c = 39, m8c = 72, m15c = 33, m2c = 236). (E) UMAP visualization of the expression level of *Plvap*. (F) IF for LYVE1, RELN, and PLVAP on a liver tissue section from a *Pdgfra^{H2bGFP}* reporter mouse. Arrows indicate PLVAP RELN double-positive HSC. Compare to Fig. EV5G. (G) UMAP visualization of the expression level of *Reln*, *Hhip*, and *Rspo3*. (H) ISH for *Hhip*, *Rspo3*, and *Reln* on a liver tissue section. Arrows indicate *Hhip Rspo3 Reln* triple-positive HSC. (I) UMAP visualization of the expression level of pericyte marker genes (*Ace2*, *Kcnj8*, and *Abcc9*). (J) IF for CD31 and ACE2 on a liver tissue section from a *Pdgfra^{H2bGFP}* reporter mouse. Arrows indicate GFP ACE2 double-positive HSC. (K) ISH for *Pecam1*, *Ace2*, and *Reln* on a liver tissue section. Arrows indicate *Ace2 Reln* double-positive HSC. (L) UMAP visualization of the expression level of *Cdh3*. (M) IF for EMCN and CDH3 on a liver tissue section from a *Pdgfra^{H2bGFP}* reporter mouse focusing on the portal tract (left pair) or central vein (right pair). Arrows indicate GFP CDH3 double-positive HSC. PV portal vein, BD bile duct, HA hepatic artery, CV central vein. Scale bars are indicated in the respective image panels. Source data are available online for this figure.

We wanted to investigate the potential contribution of the herein identified HSC subpopulations in hepatic fibrotic disease. Therefore, we analyzed HSC populations after $CCl_4$ treatment (Dobie et al, 2019b; Mederacke et al, 2022b). Reprocessing and analysis revealed a comparable cell distribution pattern between the two studies, with respect to control/untreated samples and different levels of $CCl_4$ exposure (Fig. 8A–C; Appendix Fig. S6A–D). To identify cHSC within the disease datasets we generated gene signatures for cHSC and HSC from our data using stringent parameters (Fig. 8D, compare to Fig. EV3F). Visualization of the gene signature values in the disease datasets showed that the cHSC signature was increased in cells from the $CCl_4$-treated animals and that this population was increasing over time, whereas the HSC gene signature declined with $CCl_4$ exposure (Fig. 8E,F). Expression of the prototypic marker for cHSC, *Wt1*, localized in cells from $CCl_4$ treatment groups (Fig. 8B,G). Collectively, these data suggest that the cHSC population expands in hepatic fibrotic disease. A related study (Filliol et al, 2022) categorized two distinct HSC subtypes under physiological and pathological conditions; myHSC (myofibroblastic HSC) and cyHSC (cytokine- and growth factor-expressing HSC). We collected marker genes for myHSC and cyHSC and analyzed their differential expression between cHSC and HSC (Appendix Fig. S6E,F; Dataset EV2). Similarly, this data emphasizes that cHSC–myHSC, and HSC–cyHSC, respectively, exhibit related phenotypes and possibly share lineage relationships. These results suggest that an improved determination of disease-relevant gene expression profiles can be generated considering the herein defined subtypes of HSC and their gene signatures (Appendix Fig. S6G).

Next, we addressed the reaction of fibroblast to either bile duct ligation (BDL) and $CCl_4$ treatment. When reanalyzing the GSE137720 Smart-seq2 dataset (Dobie et al, 2019b), we first mapped fibroblast subtypes in the uninjured dataset based on the identified marker genes from our analysis (Appendix Fig. S7A,B). All three major subtypes: portal, peribiliary, and *Wt1+* fibroblast were identified (Appendix Fig. S7B). We then separately analyzed fibroblasts from BDL and $CCl_4$ samples to identify the lineage relationship of homeostatic to reactive fibroblast populations, using monocle3 trajectory analysis (Cao et al, 2019; Trapnell et al, 2014) (Appendix Fig. S7C–F). For the BDL samples, we identified two trajectories of disease-specific fibroblast subtypes with origin from peribiliary/portal and *Wt1+* fibroblasts (Fig. 8H,I). For the $CCl_4$ samples one trajectory was identified with *Wt1+* fibroblasts as the nearest origin (Fig. 8J,K). DEG analysis highlighted distinct and

potentially disease-relevant gene expression patterns in the respective fibroblast subpopulations (Appendix Fig. S7G,H; Dataset EV3). GO analysis highlighted terms related to 'extracellular matrix' for BDL-induced fibroblast subtypes and terms related to 'inflammation' and 'chemotaxis' for $CCl_4$-induced fibroblast subtypes (Appendix Fig. S7I). These disease-relevant gene expression patterns should be further explored and validated.

In summary, the re-analysis and comparison of previously published scRNA-seq datasets with the data provided herein illustrate the importance of a detailed knowledge of the heterogeneity of HSC and fibroblasts, as identified in our present study, to understand the cell types involved in physiological and pathological processes in the liver.

## Discussion

In the present paper, we provide a comprehensive analysis of hepatic cell types in the adult mouse with particular emphasis on vascular and peri-vascular mesenchymal cells, summarized in Appendix Fig. S8. Our study includes molecular signatures for hepatocytes, cholangiocytes, endothelial cells from the portal vein, sinusoids, central vein, peribiliary vasculature (PBV), and lymphatics, as well as Kupffer cells, and signatures that differentiate between subtypes of hepatic stellate cells (HSC), vascular mural cells, and subpopulations of fibroblasts. Specifically, we outline molecular features to demarcate up to four subtypes of HSC, several different fibroblast populations, and substantial heterogeneity among vascular mural cells, including vascular SMC that form the walls of portal and central veins. Our data can be explored in an appended database, providing online access to gene-by-gene expression patterns at: https://muhldatahub.org/Publications/LiverScRNAseq/database.html.

Additionally, we provide examples of how our refined liver cell type annotation can be used together with previously published scRNA-seq data from human and mouse samples in order to verify the presence of PBV endothelial cells and hybrid nature of portal vein endothelial cells in the human liver, and illustrate the importance of well-defined cell type classification at the homeostatic state when inferring phenotypic changes in distinct cell types, such as HSC or fibroblasts, in response to pathological challenges.

Mesenchymal cells in the liver, in particular HSC, become activated during liver fibrosis. This leads to the production and deposition of high amounts of extracellular matrix contributing to

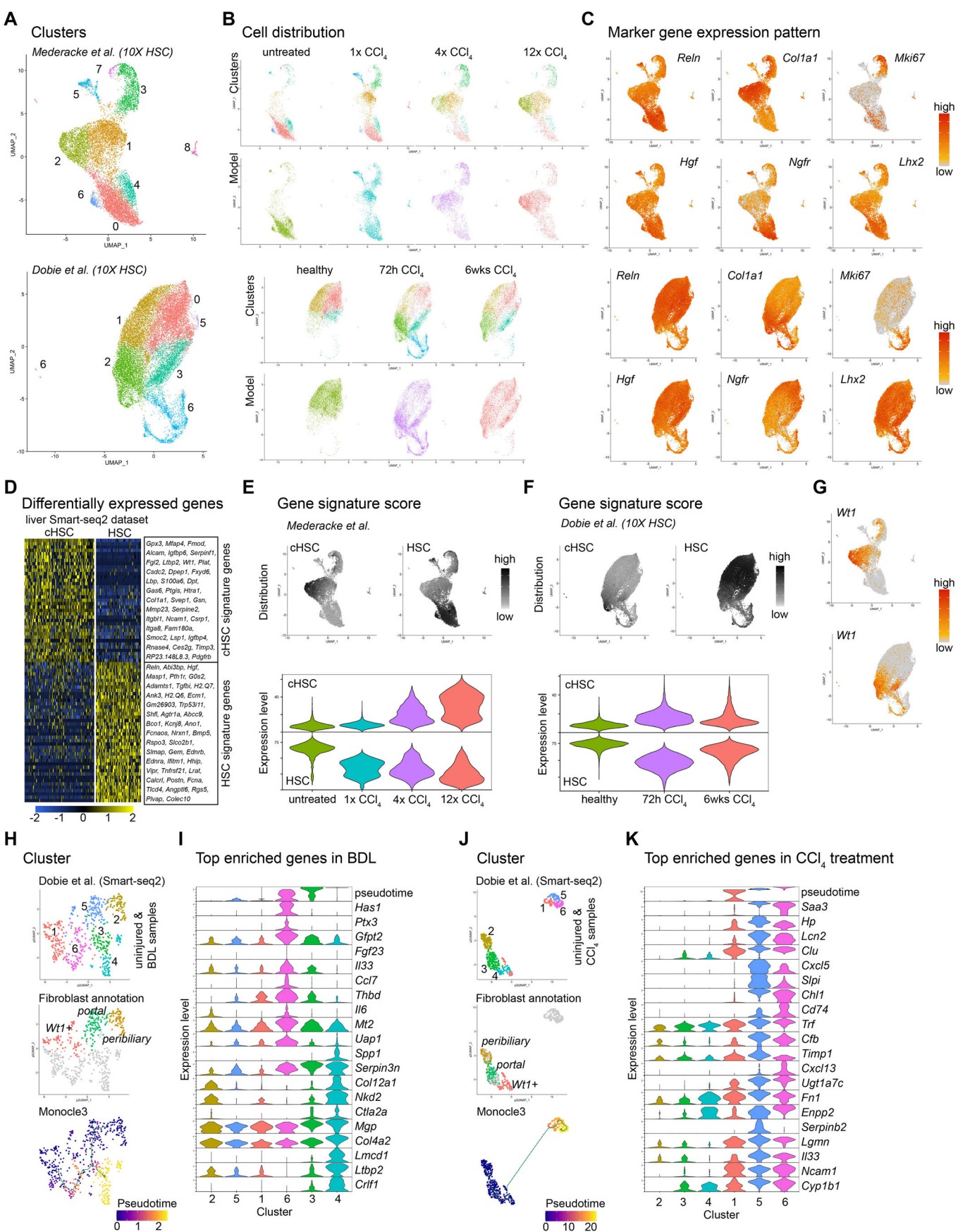

**Figure 8. Analysis of mesenchymal subpopulations in liver disease models.**

(A) UMAP visualization of the clustering result of the 10X datasets GSE172492 (upper panel) and GSE137720 (lower panel) using samples with isolated HSC from uninjured/healthy and CCl$_4$-treated animals, color coded for the Seurat clustering result. (B) UMAP visualization of the cell distribution dependent on sample treatment for both datasets (GSE172492 upper panels and GSE137720 lower panels). (C) UMAP visualization of the expression of exemplary HSC marker genes (*Reln*, *Ngfr*, *Hgf*, *Lhx2*), a fibroblast/myofibroblast maker (*Col1a1*), and a marker for cycling cells (*Mki67*), in the two datasets (GSE172492 upper panels, GSE137720 lower panels). (D) Heat map showing the genes identified as cHSC or HSC signature genes from DEG analysis using the thresholds, adjusted *P* value ≤ 0.05, log$_2$ fold-change ≥1 and expressed in ≥50% of cells. (E, F) cHSC or HSC signature gene score distribution and expression in the GSE172492 dataset (E), and GSE137720 dataset (F) visualized in the UMAP landscape (upper panels) and as violin plot highlighting the different treatment groups (lower panels, *n* refers to cells: untreated=3036, 1× CCl$_4$ = 3509, 4× CCl$_4$ = 2778, 12× CCl$_4$ = 3199/healthy=6839, 72 h CCl$_4$ = 7260, 6wks CCl$_4$ = 5453). (G) UMAP visualization of the expression level of *Wt1* in the GSE172492 (upper panel) and GSE137720 (lower panel) datasets. (H) UMAP visualizations showing the clustering result of the selected fibroblast populations (uninjured, BDL) from the GSE137720 (Smart-seq2) dataset (upper panel), the distribution of identified fibroblast subtypes from the uninjured samples (middle panel), and the result of monocle3 trajectory analysis color coded for pseudotime (lower panel). (I) Violin plot showing the expression level of exemplary genes enriched in clusters with high pseudotime values (clusters #3,4 and #6, *n* refers to cells: 2 = 80, 5 = 117, 1 = 128, 6 = 99, 3 = 73, 4 = 77). (J) UMAP visualizations showing the clustering results of the selected fibroblast populations (uninjured, CCl$_4$) from the GSE137720 (Smart-seq2) dataset (upper panel), the distribution of identified fibroblast subtypes from the uninjured samples (middle panel), and the result of the monocle3 trajectory analysis color coded for pseudotime (lower panel). (K) Violin plot showing the expression level of exemplary genes enriched in clusters with high pseudotime values (clusters #1,5,6, *n* refers to cells: 2 = 112, 3 = 159, 4 = 64, 1 = 75, 5 = 54, 6 = 57).

fibrous scarring that impedes liver function (Koyama and Brenner, 2017; Mederacke et al, 2013). The origin of these reactive HSC during liver fibrosis remains enigmatic, however. We provide molecular definition to a population of *Wt1* expressing HSC (cHSC) located below the liver capsule that has previously been suggested as the source of HSC populating the entire liver (Asahina et al, 2011; Ijpenberg et al, 2007). Analysis of published scRNA-seq studies indicates that this *Wt1*-positive population expands in models of liver injury. While mesothelial-mesenchymal transition has been suggested to account for the emergence of *Wt1*-positive HSC in disease (Li et al, 2013), we demonstrate that a *Wt1*-positive HSC population (cHSC) exists already during homeostasis and is a likely source of fibrogenic cells in liver disease. Based on the revised cell type definitions presented herein, lineage tracing studies (past and future) should be reevaluated to clarify the origin of myofibroblasts (assumed to be activated HSC), i.e. whether they differentiate from *Wt1*-positive cHSC, arise from the de novo differentiation of mesothelial cells, or derive from activated HSC from periportal, mid-lobular, or pericentral location. Additionally, the possible contribution of *Wt1*+ fibroblasts from the capsular niche requires clarification through further studies.

Heterogenic fibroblast populations have been described surrounding the extra- and intrahepatic bile ducts (Lei et al, 2022). Rare *Wt1 Pdgfra* double-positive cells were also found proximal to the mesothelial cell layer in the extrahepatic bile duct (Singh et al, 2024). Their relationship to the *Wt1*+ fibroblasts at the liver capsule is intriguing given the possible continuous anatomic layer of the capsule. Our analysis characterizes different fibroblast subpopulations residing within the portal tract forming distinct anatomic structures, such as the (Glisson's) capsule and peribiliary niche. The importance of hedgehog and Wnt/BMP signaling for portal tract mesenchymal cells has been reported (Gupta et al, 2020; Wells, 2014b), but the fine-tuned roles of portal fibroblast subpopulations in homeostasis and disease are yet to be disentangled.

The heterogeneous nature of portal vein endothelial cells—exhibiting arterial and venous features—has not, to the best of our knowledge, been described in vascular beds of other organs. This likely reflects the unique physiological function of portal veins that conduct blood between two capillary beds—the intestinal and the hepatic—in contrast to other veins that direct blood from capillaries directly back to the heart. Notably, a similar arterial/

venous hybrid phenotype was not noticed in portal vein mural cells, which exhibit molecular phenotypes resembling those in venous mural cells of other tissues (Muhl et al, 2022b). However, we identified several subtypes of venous SMC in the walls of hepatic veins. Portal veins exhibited a higher number of SMC and greater variety of venous SMC subpopulations compared to central veins, possibly accommodating the higher-pressure profile for the portal venous system. The underlying cause and function of portal vein endothelial hybridism requires further investigation to disentangle developmental cues, the physiological, and potential pathological roles of this specialized endothelial cell subpopulation.

The presence of KIT-positive sinusoidal endothelial cells in the terminal sinusoids but not in the central vein endothelial cell pool suggests a distinct function of the sinusoids at this location, which however remains unclear. Notch-signaling has been shown to regulate the expression of *Kit* in sinusoidal endothelium (Duan et al, 2022), and in a loss-of-function model for *Dll4-Notch1* signaling, it was demonstrated that pericentral sinusoidal endothelial cells were the most reactive endothelial population (Fernandez-Chacon et al, 2023), despite a low endogenous expression of *Dll4*.

Our analysis of the capillary endothelial cells of the PBV indicate that these vessels have distinct molecular phenotypes compared to hepatic sinusoids, and that they are supported by mural cells that resemble pericytes of other peripheral organs, such as the heart, in contrast to the sinusoids, which lack regular pericytes but instead harbor HSC. However, similar to the sinusoidal endothelium, the PBV capillaries express genes indicative of fenestrations, and ultrastructural analysis of human liver indeed identified fenestrations in PBV capillaries (Kono and Nakanuma, 1992), suggesting a potential exchange of macromolecules between the blood and biliary cells (Morell et al, 2013). The role of vascular and perivascular cells of the bile duct system in portal and biliary fibrotic processes (Gupta et al, 2020; Lei et al, 2022; Singh et al, 2024; Wells, 2014b) warrants further analysis. Such approaches, including techniques like Cre-recombinase mediated cellular labeling and targeting, can now be undertaken using cell type specific markers described herein. For example, the lack of cholangiocyte-specific promoters has hampered the progress in the understanding of cholangiocyte physiology and pathology (Banales et al, 2019), and our transcriptomic characterization of these and other hepatic cells should aid the development of new genetic tools.

Kupffer cells are the resident macrophage population of the liver, patrolling between sinusoidal lumen and space of Disse (Guilliams and Scott, 2022). A unique feature of Kupffer cells among macrophages is the expression of CDH5 (VE-cadherin) (Scott et al, 2018), a commonly considered endothelial cell-specific protein, which has recently been identified also in fibroblast populations of the mouse meninges (Mapunda et al, 2023; Pietila et al, 2023). The functional consequence of CDH5 expression by Kupffer cells remains unknown, but one can speculate that homotypic interaction of CDH5 aids cell-cell contacts between Kupffer cells and sinusoidal EC.

In conclusion, this comprehensive atlas of murine liver cell types will be useful—and hopefully inspire new research—to further explore the various roles of distinct hepatic cell types during health and disease.

## Limitations of the study

Although we provide transcriptomic and histological analysis to demarcate more than 20 different cellular subpopulations of the adult mouse liver in homeostasis, we cannot exclude the existence of other resident cell types that may have been lost during our isolation protocol. For example, two populations of cholangiocytes (large and small) have been described (Tabibian et al, 2013), whereas our cholangiocyte transcriptomes appeared homogenous. Another example is the absence of hepatic artery endothelial cells in our dataset, which is surprising considering the efficient capture of arterial SMC. Additionally, we have utilized transgenic reporter animals for targeted sorting of some cell types and cannot exclude the possibility that expression of the fluorescent proteins may have inflicted alterations to the normal transcriptome of these cells. The observed—previously unexpected—heterogeneity of mural cell, HSC, and fibroblast populations led to comparably small cell numbers for certain subpopulations (also for endothelial cells of the PBV). However, the Smart-seq2 platform provides deep sequencing reads, which allows for reliable cell type annotations also from small numbers of cells. Finally, using IF and ISH analysis, we identified two distinct HSC populations that reside close to the portal tract, or central vein, respectively. These HSC subpopulations were not captured by single-cell transcriptome profiling, but the provided information herein should aid the design of dedicated approaches for the characterization of rare HSC populations.

All studies like the present one have inherent limitations that include the process of cell type annotation and the criteria for inclusion into or exclusion from pre-existing cell classes. The liver mesenchyme represents a good example for this problem—exhibiting substantial overlapping expression patterns for many commonly applied marker genes with assumed specificities (e.g., *Ngfr*, *Reln*, *Lrat*, *Pdgfra*, *Pdgfrb*, *Acta2*, *Kcnj8*, *Abcc9*). Consequentially arising questions include: How should one separate and demarcate HSC from SLC or fibroblasts? When should a mural cell be called SMC, SMC subtype, pericyte, or SLC? The simple use of the cluster assignment as an unbiased classification is tempting, however it inherits the problems that the results are dependent on the set parameters of the algorithms (e.g. the *multilevel* or *community* setting in pagoda2) and the overall heterogeneity of the input data. Presumably, there is no golden rule that fits all single-cell datasets and types of analysis. Researchers are required to test for and apply the best compromise for each dataset in question to obtain the most meaningful output.

For validation and to link the annotation of transcriptomes to anatomic locations, we used image analysis of IF, ISH, and transgenic reporters. Although this is essential, these procedures are biased and dependent on the quality and availability of the reagents. For example, to pinpoint the anatomic location of capsular *Wt1+* fibroblasts we used the most relevant markers obtained from the scRNA-seq analysis (*Scara5*, *Osr1*), however, we did not find a clear staining pattern for these markers at the liver capsule, or elsewhere in the tissue, and relied on fewer markers to annotate *Wt1+* fibroblasts (*Wt1 Pdgfra* double-positive, *Reln*-negative). Hence, we cannot exclude the possibility that cells here annotated as capsular *Wt1+* fibroblasts may also be found at other anatomical locations, for example, the extrahepatic bile duct (Singh et al, 2024).

New iterations of genetic lineage-tracing and reporter models, possibly created on the information from this and/or other contemporary studies may generate unambiguous results and interpretations that allow for further revised cell type identification. Although we have analyzed cells and tissues from female and male mice, we have not conducted a formal sex-specific analysis due to an imbalance in the obtained cell numbers from the sexes.

## Methods

### Reagents and tools table

| Reagent/resource | Reference or source | Identifier or catalog number |
|---|---|---|
| **Experimental models** | | |
| C57BL6/J | The Jackson Laboratory | Were maintained as breeding colony at the local animal facility |
| Pdgfrb^GFP | Gensat.org | Tg(Pdgfrb-eGFP) JN169Gensat/Mmucd |
| Pdgfra^H2bGFP | Gift from P. Soriano | Pdgfra^tm11(EGFP)Sor |
| Cspg4^dsRED | The Jackson Laboratory | Tg(Cspg4-DsRed.T1) Akik/J |
| Acta2^GFP | The Jackson Laboratory | Tg(Acta2-GFP)1Pfk |
| Cldn5^GFP | | Tg(Cldn5-GFP)Cbet/U |
| Gja5^GFP | provided by L. Miquerol and R. Kelly (IBDM, Aix-Marseille University, France) | Gja5^tm1Lumi |
| **Antibodies** | | |
| ACE2 (goat) | R&D Systems | AF3437 |
| Alpha-SMA-Cy3 (mouse) | Sigma | C6198 |
| Alpha-SMA-FITC (mouse) | Sigma | F3777 |
| CD200 (goat) | R&D Systems | AF2724 |
| CD31 (rat) | BD Bioscience | 550274 |
| CD31 (goat) | R&D Systems | AF3628 |
| CD31-APC (rat) | BD Bioscience | 561814 |

| Reagent/ resource | Reference or source | Identifier or catalog number |
|---|---|---|
| CD31-FITC (rat) | BD Bioscience | 561813 |
| CD36 (goat) | R&D Systems | AF2519 |
| CD68 (rabbit) | Cell Signaling Technology | 97778T |
| CD68-APC (rat) | BioLegend | 137008 |
| CLEC4F (goat) | R&D Systems | AF2784 |
| CLEC4F-AlexaFluor647 (mouse) | BioLegend | 156803 |
| CNN1 (rabbit) | Abcam | ab216651 |
| EMCN (rat) | eBiosciences | 14-5851-82 |
| EPCAM-APC (rat) | eBiosciences | 17-5791-82 |
| GPMA6A (rat) | LS-Bio | LS-C179305 |
| KIT (goat) | R&D Systems | AF1356 |
| LYVE1 (rabbit) | Cell Signaling Technology | 67538S |
| NCAM1 (goat) | R&D Systems | AF2408 |
| NGFR (rabbit) | Abcam | ab52987 |
| NPNT (goat) | R&D Systems | AF4298 |
| P-Cadherin (goat) | R&D Systems | AF761 |
| PLVAP (rat) | BD Bioscience | 550563 |
| RELN (goat) | R&D Systems | AF3820 |
| VE-Cadherin (goat) | R&D Systems | AF1002 |
| VE-Cadherin-AlexaFluor488 (rat) | eBiosciences | 53-1441-82 |
| VE-Cadherin-PE (rat) | eBiosciences | 12-1441-82 |
| VE-Cadherin-eFluor450 (rat) | eBiosciences | 48-1441-80 |
| vWF (rabbit) | DAKO | A0082 |
| WT1 (rabbit) | Abcam | ab89901 |
| **Oligonucleotides and other sequence-based reagents** | | |
| RNAscope probes: | | |
| *Ace2* (mouse, Mm) | ACD Bio | 417081-C2 |
| *Acta2* (mouse, Mm) | ACD Bio | 319531-C3 |
| *Adgrg6* (mouse, Mm) | ACD Bio | 467351-C2 |
| *Aplnr*-O1 (mouse, Mm) | ACD Bio | 517691 |
| *Ccn3* (mouse, Mm) | ACD Bio | 415341 |
| *Chrdl1* (mouse, Mm) | ACD Bio | 442811 |
| *Cnn1* (mouse, Mm) | ACD Bio | 483791 |
| *Cyp1a1* (mouse, Mm) | ACD Bio | 464611 |
| *Epcam* (mouse, Mm) | ACD Bio | 418151-C2 |

| Reagent/ resource | Reference or source | Identifier or catalog number |
|---|---|---|
| *Ephx3* (mouse, Mm) | ACD Bio | 412511 |
| *Hhip* (mouse, Mm) | ACD Bio | 448441-C2 |
| *Gfra2* (mouse, Mm) | ACD Bio | 441481 |
| *Gja5* (mouse, Mm) | ACD Bio | 518041-C2 |
| *Pdgfra* (mouse, Mm) | ACD Bio | 480661-C2 |
| *Pdgfrb* (mouse, Mm) | ACD Bio | 411381-C2 |
| *Pecam1* (mouse, Mm) | ACD Bio | 316721, 316721-C3 |
| *Pecam1*-O1 (mouse, Mm) | ACD Bio | 471481-C2 |
| *Pi16* (mouse, Mm) | ACD Bio | 451311-C3 |
| *Pln* (mouse, Mm) | ACD Bio | 506241 |
| *Plvap* (mouse, Mm) | ACD Bio | 440221 |
| *Reln* (mouse, Mm) | ACD Bio | 405981-C3 |
| *Rspo3* (mouse, Mm) | ACD Bio | 402011 |
| *Sema3g* (mouse, Mm) | ACD Bio | 494691 |
| *Tek* (mouse, Mm) | ACD Bio | 414821-C2 |
| *Wif1* (mouse, Mm) | ACD Bio | 412361 |
| *Wt1* (mouse, Mm) | ACD Bio | 432711 |
| **Chemicals, enzymes and other reagents** | | |
| Skeletal Muscle Dissociation Kit, mouse and rat | Miltenyi Biotec | 130-098-305 |
| Collagenase IV type-S | Sigma-Aldrich | C1889 |
| RNAscope Fluorescence Multiplex assay | ACD | 320851 |
| RNAscope Fluorescence Multiplex assay V2 | ACD | 323110 |
| RNAscope ancillary kit | ACD | 323120 |
| Fluorescent dyes | Akoya Biosciences (Perkin Elmer) | NEL741E001KT NEL744E001KT NEL745E001KT FP1497991KT FP1495001KT |
| Protein Block, Serum-free | DAKO, Agilent Technologies | X090930-2 |
| Hoechst 33342 | ThermoFisher Scientific | H3570 |
| ProLong Gold antifade reagent | ThermoFisher Scientific | P36930 |

| Reagent/ resource | Reference or source | Identifier or catalog number |
|---|---|---|
| **Software** | | |
| R-software v4.1.1 | | |
| BackSPIN | https://github.com/ linnarsson-lab/BackSPIN | |
| LAS X v3.5.7.23225 | Leica Microsystems | |
| Fusion v2.4 | Andor Technologies Inc. | |
| ImageJ/FIJI v2.0.0-rc-69/ 1.52p | | |
| Prism v10.5.0 | GraphPad | |
| FACS Diva | BD Bioscience | |
| FACS Chorus | BD Bioscience | |
| **Other** | | |
| FACSAria III | BD Bioscience | |
| FACSMelody | BD Bioscience | |
| TapeStation 4200 | Agilent Biotechnologies | |
| Bioanalyzer 2100 | Agilent Biotechnologies | |
| HiSeq3000 | Illumina | |
| Leica SP8 confocal microscope | Leica Microsystems | |
| Nikon Eclipse Ti2 microscope | DragonFly 505, Andor Technologies Inc. | |

## Animals

All experimental procedures involving animals were carried out in accordance with the Swedish legislation and local regulations and guidelines for animal welfare. All mouse experiments were approved by local authorities: Linköpings Animal Research committee—approval ID 729, 3711-2020, and 11939-2023, and Uppsala Animal Research committee—approval ID 03029/20290, and 5.8.18-116497/2024. Mice were housed in standard, single ventilated cages with 12 h light, 12 h dark cycle, ad libitum access to water and chow and an ambient temperature of $20 \pm 2\,°C$ with a relative humidity of $50 \pm 5\%$.

The following mouse strains were used in the study: *C57BL6/J* (The Jackson Laboratory, *C57BL6/J*, were maintained as breeding colony at the local animal facility), *Pdgfrb*$^{GFP}$ (Gensat.org, Tg(*Pdgfrb*-eGFP)JN169Gensat/Mmucd), *Pdgfra*$^{H2bGFP}$ (*Pdgfra*$^{tm11(EGFP)Sor}$ (Hamilton et al, 2003), a gift from P. Soriano), *Cspg4*$^{dsRED}$ (The Jackson Laboratory, Tg(*Cspg4*-DsRed.T1)Akik/J), *Acta2*$^{GFP}$ (The Jackson Laboratory, Tg(*Acta2*-GFP)1Pfk) (Yokota et al, 2006), *Cldn5*$^{GFP}$ (Tg(*Cldn5*-GFP)Cbet/U), and combinations of these strains. All reporter transgenes were kept as heterozygous. For experiments, adult male and female mice with an age between 8 weeks and one year were used. Liver tissue samples from *Gja5*$^{GFP}$ mice (*Gja5*$^{tm1Lumi}$) (Miquerol et al, 2004), were generously provided by L Miquerol and R Kelly (IBDM, Aix-Marseille University, France).

## Preparation of single-cell suspension from mouse liver tissue

Mouse liver tissues were handled and processed using the same protocol as described before (Muhl et al, 2020; Muhl et al, 2022b): In brief, animals were euthanized by cervical dislocation and the liver was dissected out and placed into ambient-tempered phosphate-buffered saline (PBS) solution (DPBS, ThermoFisher Scientific), until further processing. The liver tissue was then cut into smaller pieces with scissors and scalpel and incubated in dissociation buffer (Skeletal muscle dissociation kit from Miltenyi Biotec, supplemented with 1 mg/ml Collagenase IV type-S from Sigma-Aldrich) at 37 °C with orbital shaking at 500–800 rpm. Mechanical disintegration by pipetting was applied to facilitate tissue disintegration in three to four cycles with 10 min incubation intervals. After dissociation, the remaining tissue debris were removed by sequential passing of the cell solution through a 70 µm and a 40 µm cell strainer. The 70 µm strainer was additionally washed with 5 ml of DMEM (ThermoFisher Scientific) to recover cells adherent to the surface. Thereafter, the cells were pelleted by centrifugation at $300 \times g$ for 5 min. The supernatant was removed, and the cell pellet resuspended in FACS buffer (DPBS supplemented with 0.5% bovine serum albumin, 2 mM EDTA, 25 mM HEPES). The cell suspension was then labeled with different combinations of fluorophore-conjugated antibodies (anti-CD31, anti-EPCAM, anti-CD68, anti-PDGFRα, anti-VE-Cadherin) to mark distinct cell populations for capture (Reagents and Tools Table) for 20–30 min at room temperature (RT). Thereafter, the cells were pelleted by centrifugation at $300 \times g$ for 5 min, the supernatant removed, and cells resuspended in ice-cold FACS buffer and placed on ice until further processing.

## Fluorescent activated cell sorting (FACS)

Single-cell analysis, selection, gating, and deposition of selected droplets (single cells) was done as described before (Muhl et al, 2020; Muhl et al, 2022b). The antibody stained single-cell suspensions were analyzed using a FACSAria III or FACSMelody (Becton Dickinson Biosciences) cell sorter, each equipped with a 100 µm nozzle. Single-cell events meeting the criteria as described below were collected by deposition into 384-well plates containing 2.3 µl lysis buffer (0.2% Triton X-100, 2 U/ml RNase inhibitor, 2 mM dNTP, 1 µM Smart-dt30VN, ERCC at 1:4 $\times10^4$ dilution) per well. Of note, the analysis of single-cell events at the cell sorter was not the basis for cell type identification, but for the enrichment of target cell populations dependent on the signal of expressed reporter genes or antibody labeling. For sorting into 384-well plates, first debris and red blood cells were excluded by setting a generous gate with forward scatter-area/side scatter-area (FCS-A/ SSC-A, linear scale). For doublet discrimination, a second gate using SSC-A/SSC-height and FSC-A/FSC-height was used. Thereafter, cells were analyzed for their fluorescent signals and separated dependent on their reporter gene expression and antibody labeling. Fluorescent signals were controlled applying the 'fluorescence minus one' method, using samples without antibody labeling and/ or from reporter gene negative animals.

Endothelial cells were collected based on their labeling with antibodies against CD31 and/or VE-Cadherin, or from *Cldn5*$^{GFP}$

mice dependent on the reporter gene signal. Of note, we noticed that sinusoidal endothelial cells exhibited a lower staining intensity for CD31 but higher staining for VE-Cadherin, compared to endothelial cells from the portal and central veins, which instead were CD31$^{high}$ VE-Cadherin$^{low}$. Epithelial cells (cholangiocytes) were selected dependent on their anti-EPCAM antibody labeling. Macrophages (e.g. resident Kupffer cells) were collected dependent in their anti-CD68 antibody labeling. Hepatic stellate cells (HSC), fibroblasts, pericytes, smooth muscle cells (SMC), and other vascular mural cells were sorted according to their distinct $Pdgfra^{H2bGFP}$, $Pdgfrb^{GFP}$, $Acta2^{GFP}$ reporter gene expression patterns, as well as with anti-PDGFRα antibodies. Hepatocytes were collected by unbiased sorting of a small amount of living cells. Before sorting, the 384-well plates containing lysis buffer were briefly centrifuged to ensure the proper dispersion of the lysis buffer in the bottom of the wells. Importantly, the correct deposition of the selected droplets (i.e., single cells) was controlled by test-spotting (aiming) of beads or cell populations from control samples onto the seal of the respective 384-well plate. If needed, the plate holder position was adjusted for a centered deposition of the droplets. This procedure was performed for each individual plate. The sample-stand and plate-holder were maintained at 4 °C during the analysis and sorting procedure. Completed sorted plates were immediately sealed and placed on dry-ice and then stored at −80 °C until further processing.

## ScRNA-seq library preparation and sequencing

Library preparation and sequencing was performed as described before (Muhl et al, 2020; Muhl et al, 2022b), and according to the previously established protocol for Smart-seq2 (Picelli et al, 2014). In brief, poly-adenylated mRNA was transcribed to cDNA using oligo(dT) primer and SuperScript II reverse transcriptase (Thermo-Fisher Scientific). Second strand cDNA synthesis was achieved using a template switching oligo, and the double stranded cDNA was then amplified with PCR. Amplified cDNA was purified, and quality was controlled (QC) by analyzing randomly selected wells (single cell samples) on a TapeStation 4200 or a 2100 Bioanalyzer using DNA high sensitivity chips (Agilent Biotechnologies). When samples (plates) passed the QC, the cDNA was enzymatically fragmented and tagged using Tn5 transposase. Each single well was uniquely indexed using the Illumina Nextera XT index kits (set A-D). Finally, the uniquely indexed cDNA libraries from one 384-well plate were pooled and sequenced together on one lane of a HiSeq3000 sequencer (Illumina), using the sequencing strategy of dual indexing and single 50 base-pair reads.

## Sequence data processing

The obtained sequences (as outlined above) were handled for demultiplexing, mapping and generation of per cell and gene raw-count expression matrices as described earlier (Muhl et al, 2020). ENSEMBLE identifiers were annotated using the org.Mm.eg.db package (version 3.14.0) in R-software (version 4.1.1, R Core Team, https://www.R-project.org), retaining ERCC counts in the expression matrix as technical control. Annotated raw counts were loaded into the Seurat package (version 4.3.0) (Satija et al, 2015). Low-quality cells ( ≤50,000 counts library size, ≤1500 expressed genes, ≥10% mitochondrial genes, ≥10% ERCC counts), as well as putative

doublets ( ≥12,000 expressed genes) were removed from the dataset. Low expressed genes: expressed in ≤3 cells with a detection limit = 20 counts per gene, and ≤300 total counts per gene, were also removed from the dataset before further processing. Additionally, cells that showed a transcriptome with clear signs of cross cell type contamination were removed from the dataset. After filtering, the dataset consisted of 3491 single-cell transcriptomes, collected from 18 individual mice.

Smart-seq2 data from this and earlier studies (see also below) were processed using the pagoda2 (version 1.0.10, https://github.com/hms-dbmi/pagoda2) R-software package (Fan et al, 2016). General attributes, such as PCA (nPCS = 50, n.odgenes = 3000) and nearest neighbor clustering were performed using default parameters of pagoda2 (k = 30, distance = "cosine"). Dimensional reduction analysis was done using the UMAP function (uniform manifold approximation and projection, n.neighbors = 30, metric = "cosine", dims = 1:50) (Becht et al, 2018). Pagoda2 clustering results using the *multilevel* as well as infomap.community (*community*) setting were considered for downstream analysis. Cluster calculation is based on the igraph R-software package (Csárdi and Nepusz, 2006) for both options. Clustering using community usually leads to more granular and a higher number of clusters compared to the multilevel setting, which uses Louvain based clustering. The results obtained from pagoda2 clustering were stored within the Seurat object and data visualizations were prepared using functions of the Seurat R-software package (dot plot: DotPlot(), UMAP: DimPlot() or FeaturePlot(), violin plot: VlnPlot(), heatmap: DoHeatmap), as well as the R-software package pheatmap (version 1.0.12) for heat maps, and clusterProfiler (version 4.2.2) for dot plots of gene ontology (see below) results. For the analysis of selected cell type datasets (parenchymal, endothelial, immune, mesenchymal) the same procedure as described above was applied. For the construction of the barplot graphs the pagoda2 clustering results were used as a basis. Dependent on the resolution, clusters defined by the multilevel setting, or community setting were used. The order of the clusters within the barplot graphs was manually defined. The distribution of cells within each cluster was calculated using the SPIN algorithm (/backspin -i input.cef -o output.cef -f 1000 -b both) (Tsafrir et al, 2005). For the barplot graphs, the library size for each cell was normalized to 500,000 counts.

For differential expressed gene (DEG) analysis the FindMarkers() function of the Seurat R-software package was applied, using the MAST test (MAST R-software package, version 1.20.0) (Finak et al, 2015) for adjusted $P$ value calculation. Thresholds used for gene-qualification from DEG analysis were an adjusted $P$ value ≤ 0.05, the expression in ≥30% or 50% of cells and a $\log_2$ fold-change ≥1 in the respective group (Dataset EV4).

## Cell-cell communication analysis

For cell-cell communication analysis, the CellChat (version 2.1.2) R-software package was used (Jin et al, 2025). In brief, after Seurat analysis the complete dataset was loaded into CellChat and further processed using interaction partners (ligand-receptor pairs) and standard functions provided with the CellChat R-software package. For probability calculations, the computeCommunProb() function (type = "truncatedMean" and trim = 10) was used. Cell-cell communication pathways of interest were selected using the subsetCommunication() function with slot = "netP", and ligand-receptor pairs of interest were selected using slot = "net" setting, focusing on cells identified as located within the peribiliary niche:

cholangiocytes, peribiliary fibroblasts, and peribiliary vasculature (PBV). A P value threshold of ≤0.01 and probability-score (prob) threshold of ≥0.1 were applied and paracrine ligand-receptor pairs (source ≠ target) were selected. Results were visualized as chord diagrams using the net_visual_chord_gene() function from the CellChat R-software package.

## Gene ontology (GO) analysis

For gene ontology (GO) analysis the clusterProfiler (version 4.2.2) R-software package was used (Wu et al, 2021). In brief, the enrichGO() function was used to identify enriched GO terms from the Biological Process subontology. The list for all genes in the respective dataset was used as reference (gene universe). Terms with an adjusted P value ≤ 0.05 (pvalueCutoff = 0.05, pAdjusted-Mehod = "BH" (BH = Benjamini–Hochberg)) were selected and after application of the simplify() function (cutoff = 0.7) to omit redundant terms, the top 10 terms (min-GSSize = 10, maxGSSize = 500) were displayed in the respective dot plots (Dataset EV5) created with the dotplot() function of the clusterProfiler R-software package.

## Published scRNA-seq datasets available in the public domain

Single-cell transcriptomes from earlier studies of human and mouse liver tissues were processed and analyzed in the study. We acquired the deposited raw data from the NCBI gene omnibus database with the following accession numbers: GSE172492 (Data ref: Mederacke et al, 2022a), GSE168933 (Data ref: Buonomo et al, 2022a), GSE136103 (Data ref: Ramachandran et al, 2019a), and GSE137720 (Data ref: Dobie et al, 2019a).

Processing and analysis of earlier Smart-seq2 datasets was done as described above. The HSC signature was calculated from the GSE137720 dataset, using only cells from uninjured samples. Differential gene expression was performed as described above and genes enriched in HSC clusters, compared to all other cells, were used to determine the signature score for our dataset.

For trajectory analysis of fibroblasts from the GSE137720 Smart-seq2 dataset (Data ref: Dobie et al, 2019a), we first annotated the cells within the fibroblast cluster of the uninjured dataset guided by marker gene expression from our defined fibroblast subpopulations. We then applied this information when analyzing all cells of the GSE137720 Smart-seq2 dataset. The fibroblast clusters were selected and then separately analyzed combining cells from the uninjured samples with either cells from the bile duct ligation (BDL) or $CCl_4$-treated animals. After analysis as described above, trajectories were calculated using the monocle3 (Cao et al, 2019; Trapnell et al, 2014) R-software package (version 1.4.23), retaining the UMAP coordinates and using standard functions and variable settings (cluster method = 'leiden', k = 10). For calculation of the trajectory path, the identified partitions were ignored. For pseudotime calculation, cells from the uninjured samples were set as starting point (root_cells). For the BDL samples, two possible trajectories were identified marked by clusters 3 and 4, or cluster 6, together with high pseudotime. For $CCl_4$ samples, one trajectory was identified marked by clusters 1, 5, and 6 and high pseudotime. Enriched genes with high pseudotime values (>1 for BDL samples,

>10 for $CCl_4$ samples) were identified using the FindMarkers() function of Seurat as described above, with $log_2$ fold-change ≥1, and adjusted P value ≤ 0.05 as threshold for gene qualification (Dataset EV4).

Processing and analysis of 10X datasets was done using the Seurat R-software package. In brief, raw count data was loaded into a Seurat object and low-quality cells were removed (≤500 detected genes, ≥10% counts from mitochondrial genes). The default Seurat pipeline was used to calculate general dataset features, such PCA, clusters, and UMAP distribution. When several samples were combined for analysis, but showed sample-specific clustering, technical batch effects were assumed, and data-integration functions of the Seurat R-software package (SelectIntegrationFeatures(), FindIntegrationAnchors(), IntegratData()) were applied before resuming the analysis of the dataset. Data visualization and differential gene expression analysis was performed using functions implemented in the Seurat R-software package as described above.

## Immunofluorescence staining

Standard procedures for immunofluorescence (IF) staining were applied. In brief, liver tissues were harvested from euthanized mice as described above and, if not otherwise stated, immersion fixed using 4% buffered formalin solution (Histolab) for 4–12 h at 4 °C. After fixation, the tissue samples were immersed in 20–30% sucrose/PBS solution for at least 24 h at 4 °C before further processing. For cryo-sections, tissues were, if necessary, carefully trimmed and dissected and then placed in cryo-molds and embedded using NEG50 cryo-medium and sectioned on a Cryostat NX70 (ThermoFisher Scientific) or Cryostat (Leica) into 14–30 μm thick sections, collected on SuperFrost Plus glass slides (Metzler Gläser). Sections were stored at −20 to −80 °C until further processing.

For staining, sections were placed at RT and allowed to dry for 15 min. Thereafter, the sections were briefly washed in PBS and then incubated with blocking buffer (Serum-free protein blocking solution, DAKO), supplemented with 0.2% Triton X-100 (Sigma-Aldrich). After the blocking, the sections were sequentially incubated with primary antibodies, diluted in blocking buffer supplemented with 0.2% Triton X-100, and corresponding fluorophore-conjugated secondary antibodies, diluted in blocking buffer, according to the manufacturers' recommendations (Reagents and Tools Table). For nuclear staining, Hoechst 33342 (trihydrochloride, trihydrate, ThermoFisher Scientific) at 10 μg/ml was added to the secondary antibody solution. Thereafter, sections were mounted using ProLong Gold® mounting medium (Thermo-Fisher Scientific) and sections stored at 4 °C. Micrographs were acquired using a Leica TCS SP8 confocal microscope with LAS X software (version 3.5.7.23225, Leica Microsystems), or a Nikon Eclipse Ti2 confocal microscope, equipped with iXon EMCCD and iZyla SCMOS cameras, with Fusion software (version 2.4, Dragonfly 505 high speed confocal platform, Andor Technologies, Inc). The acquired images were graphically processed and adjusted individually for brightness and contrast using ImageJ/FIJI software (Schindelin et al, 2012) or Adobe Photoshop software for optimal visualization. All images, if not otherwise stated, are presented as maximum intensity projections of the acquired z-stacks, covering the entire thickness of the respective sections.

## In situ hybridization staining (RNAscope®)

For fluorescent, multiplexed in situ RNA hybridization (ISH) the RNAscope® Fluorescent Multiplex Assay system (Advanced Cell Technologies), without or with TSA-amplification (V2 kit) was applied according to the manufacturers' recommendations. Cryosections from liver tissue were prepared as described above (Immunofluorescent staining), with or without (fresh frozen) immersion fixation before sectioning. After dehydration, the sections were prepared using Pretreat 4 solution for 15–30 min at RT. RNAscope® probes (Reagents and Tools Table) were applied according to the manufacturer's recommendations and incubated at 40 °C for 2 h. After completion of the protocol as described by the manufacturer, the sections were mounted using ProLong Gold® mounting medium and stored at 4 °C. Micrographs were acquired as described above (Immunofluorescent staining).

## Image quantification

The coverage of smooth muscle cells (SMC) around portal veins and central veins was quantified from IF-stained livers of 12 mice (5 females and 7 males, age 3–4 months, 3–7 images spread across 1–2 tissue sections per mouse). The circumference of portal veins and central veins (identified by surrounding GLUL-positive hepatocytes) of different sizes with αSMA signal was measured in FIJI, followed by measurement of the length of either the αSMA or NPNT positive signal in the wall of the respective hepatic vessel. Coverage by αSMA or NPNT was calculated as fraction of the vessel circumference. Portal veins and central veins were manually grouped into three size-groups: <250 μm, 250–500 μm, and >500 μm.

*Wt1* and *Reln* double-positive hepatic stellate cells (HSC) were quantified from ISH-stained livers of 4 mice (2 males and 2 females, age 2–4 months, 3–5 images per mouse). Double-positive cells were quantified at three different locations, (1) an area within 100 μm radius from the liver edge, (2) an area outside the 100 μm radius from the edge of the liver, and (3) from images taken of the parenchyma around large hepatic vessels (portal tracts and central veins).

## Statistics and reproducibility

Gene ontology (GO) analysis was performed using the clusterProfiler R-software package (Wu et al, 2021). Term enrichment was considered significantly when the adjusted *P* value, using the Benjamini–Hochberg procedure, was ≤0.05. Differential expressed gene (DEG) analysis was performed using the FindMarkers() function of the Seurat R-software package (Satija et al, 2015), with test.use = "MAST" option, using the MAST test (Finak et al, 2015) for adjusted *P* value calculation. An adjusted *P* value ≤ 0.05 was applied. All immunofluorescence (IF) and RNA in situ hybridization (ISH) experiments were performed at least twice using the same or varying antibody, or probe combinations, respectively. Tissue sections from at least two individual mice were analyzed for each respective IF or ISH staining experiment. For regional quantification of cHSC one-way ANOVA was used and for the quantification of αSMA and NPNT expression around portal and central veins the two-way ANOVA (mixed model) with Sídáks multiple comparisons test was used in GraphPad Prism (version 10.4.0). The respective *P* values are indicated in the figures. No blinding was done prior to statistical analysis.

## Data availability

All data to support the findings of this study are included in the paper and corresponding data-depositions. A combined barplot visualization for the display of searchable gene expression across the complete single-cell RNA-sequencing dataset and the corresponding UMAP visualization for the complete as well as sub-datasets can be accessed at https://muhldatahub.org/Publications/LiverScRNAseq/database.html. The generated sequencing raw data can be accessed at NCBI *GEO* under the accession number GSE297062. Additional heat map files for which the complete annotation is not readable in the figures can be accessed with full annotation at *Zenodo* (https://doi.org/10.5281/zenodo.16875427).

The source data of this paper are collected in the following database record: biostudies:S-SCDT-10_1038-S44319-025-00580-9.

## Peer review information

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

## Acknowledgements

We thank Professor Moustapha Hassan and the Pre-Clinical Laboratories (PKL)–Karolinska University Hospital Huddinge, as well as the Karolinska Institutet MedH fluorescent activated cell sorting facility, Cecilia Olsson, Pia Peterson, Veronica Sundell, Louise Larsson, Jana Chmielniakova, and Helene Leksell for technical help. We would also like to thank the Single-Cell Core Facility Flemingsberg campus, SICOF, which is supported by the infrastructure board of the Karolinska Institutet, for their single-cell sequencing services, and the Molecular Imaging Center (MIC), Department of Biomedicine, University of Bergen. We thank Dr. Lucile Miquerol for providing tissues from *Gja5*^GFP reporter mice. This study was supported by grants from Magn. Bergvalls Foundation (LM: 2020-03735, 2021-04275, 2022-158), the Norwegian Cancer Society (CS: Pioneer grant #255690, 2022), the Swedish Cancer Society (CB: 21 1714Pj), the Swedish Research Council (CB: 2015-00550, KG: 2021-04896, MAM: 2023-02655, UL: 2024-02414), Theme-based Research Scheme Hong Kong (UL: T12-712/21 R), the K and A Wallenberg Foundation (CB and KG: 2020.0057), and the Leduq Foundation (CB and MAM: 23CVD02; CB: 22CVD01).

## Author contributions

**Riikka Pietilä**: Conceptualization; Data curation; Formal analysis; Investigation; Visualization; Methodology; Writing—original draft; Writing—review and editing. **Guillem Genové**: Conceptualization; Data curation; Formal analysis; Investigation; Methodology; Writing—original draft. **Giuseppe Mocci**: Data curation; Methodology. **Yuyang Miao**: Investigation. **Jianping Liu**: Data curation; Methodology. **Stefanos Leptidis**: Data curation; Methodology. **Francesca Del Gaudio**: Investigation. **Martin Uhrbom**: Methodology. **Elisa Vázquez-Liébanas**: Resources; Data curation; Investigation. **Sonja Gustafsson**: Data curation. **Byambajav Buyandelger**: Data curation. **Elisabeth Raschperger**: Data curation. **Johan L M Björkegren**: Supervision. **Emil M Hansson**: Supervision. **Konstantin Gaengel**: Resources; Funding acquisition. **Maarja Andaloussi Mäe**: Resources; Supervision; Funding acquisition. **Marie Jeansson**: Resources. **Michael Vanlandewijck**: Data curation; Supervision; Methodology. **Liqun He**: Software; Visualization; Methodology. **Carina Strell**: Supervision; Funding acquisition; Project administration. **Xiao-Rong Peng**: Conceptualization; Supervision; Project administration. **Urban Lendahl**: Conceptualization; Formal analysis; Supervision; Funding acquisition; Writing—original draft; Project administration; Writing—review and editing. **Christer Betsholtz**: Conceptualization; Resources; Formal analysis; Supervision; Funding acquisition; Writing—original draft; Project administration; Writing—review and editing. **Lars Muhl**: Conceptualization; Data curation; Software; Formal analysis; Funding acquisition; Investigation; Visualization; Methodology; Writing—original draft; Project administration; Writing—review and editing.

Source data underlying figure panels in this paper may have individual authorship assigned. Where available, figure panel/source data authorship is listed in the following database record: biostudies:S-SCDT-10_1038-S44319-025-00580-9.

## Funding

## Disclosure and competing interests statement

UL is a member of the scientific advisory board of the company Satellos. SL is an employee of Causaly. The remaining authors declare no competing interests.

# Expanded View Figures

**Figure EV1.  Cell type class identification and analysis of parenchymal cell dataset.**

(**A**) UMAP visualization of the expression level of selected canonical marker genes representative for the cell type class identification. (**B**) UMAP visualization of the clustering result for the parenchymal cell (hepatocytes and cholangiocytes) datasets. (**C, D**) UMAP visualization of the expression level of exemplary genes identified as differentially expressed in the hepatocytes along the portal-central axis (**C**), or in a distinct hepatocyte subpopulation (**D**). (**E**) Heat map showing genes with enriched expression in cholangiocytes compared to hepatocytes. (**F**) Violin plots showing the expression levels of genes belonging to the GO term cell-cell junction organization (left panel) or steroid metabolic process (right panel), which are enriched in cholangiocytes when compared to the complete dataset (compare to Appendix Fig. S2B, n refers to cells: 7 = 205, 11 = 192, 1 = 415, 3 = 561, 9 = 394, 4 = 138, 6 = 119, 8 = 276, 12 = 199, 10 = 184, 5 = 279, 13 = 283, 14 = 116).

▶

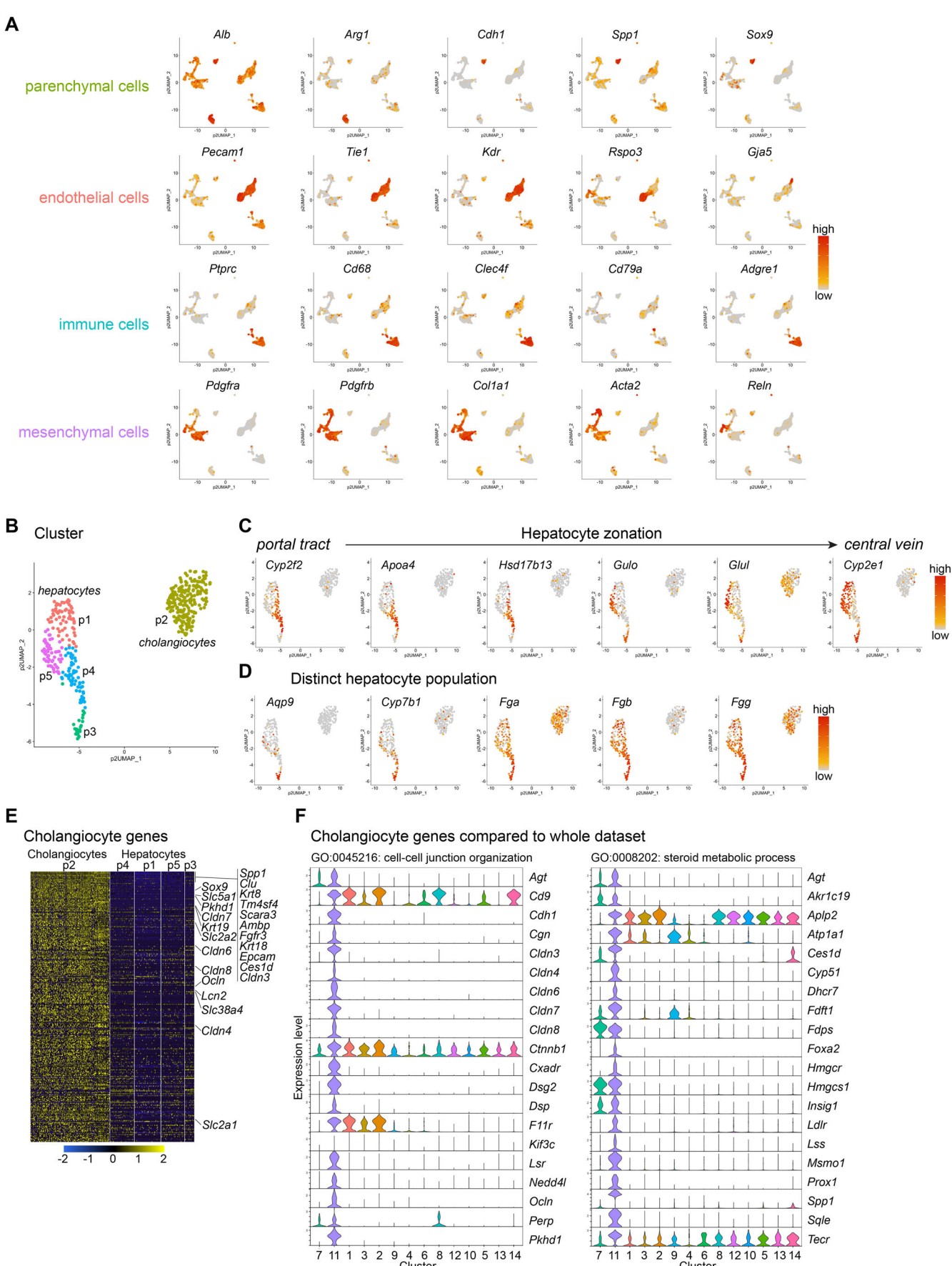

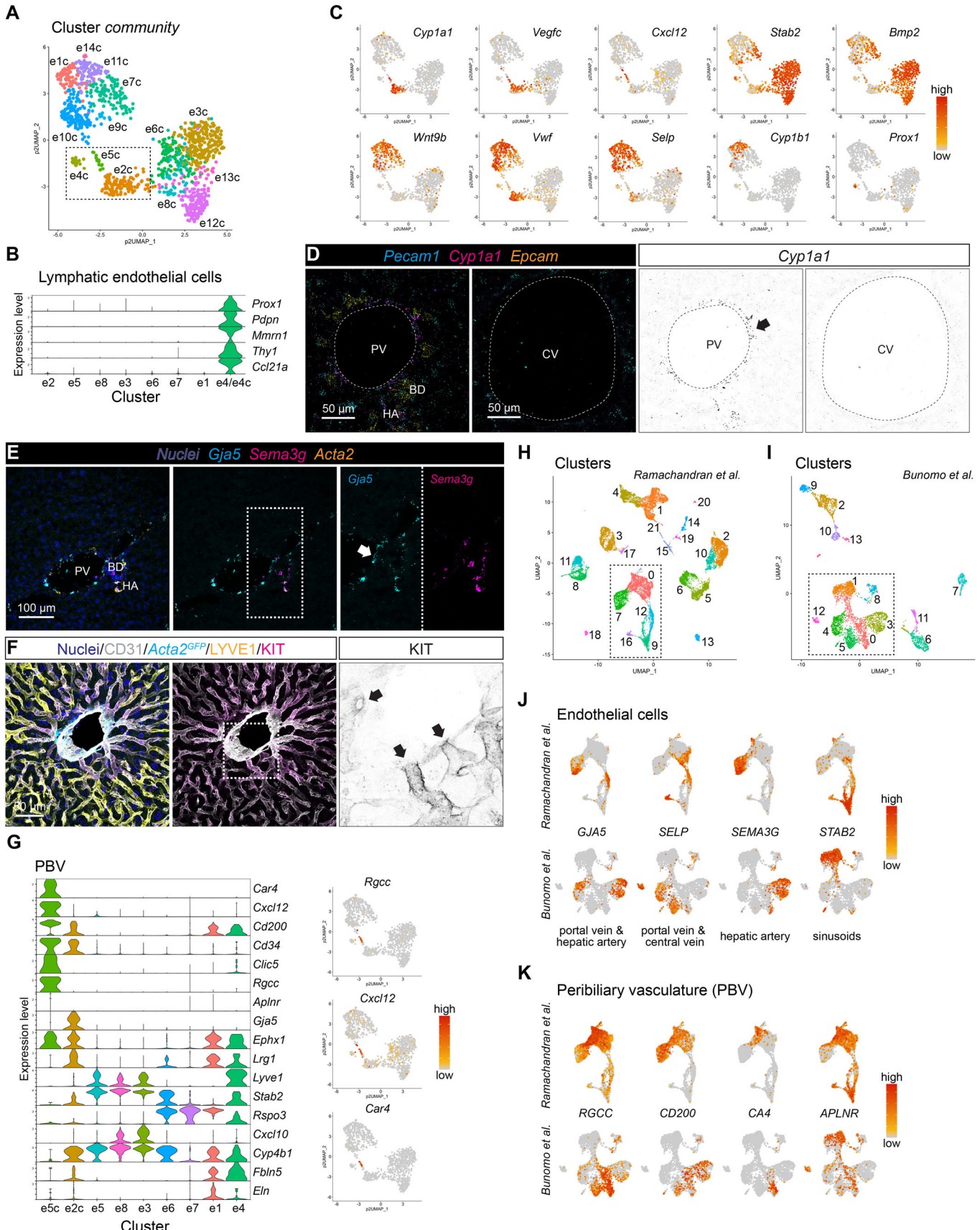

**Figure EV2.   Analysis of endothelial cell subclusters and their relation to human liver single-cell data.**

(A) UMAP visualization of the clustering results of the endothelial cell dataset using the pagoda2 community setting for high resolution clustering. (B) Violin plot showing genes expressed in lymphatic endothelial cells (cluster #e4/e4c, n refers to cells: e2 = 128, e5 = 151, e8 = 155, e3 = 236, e6 = 112, e7 = 150, e4/e4c = 22). (C) UMAP visualization of the expression levels of exemplary genes with zonated expression along the portal-central axis in hepatic endothelial cells, and *Prox1* as lymphatic endothelial cell marker. (D) ISH for *Pecam1*, *Cyp1a1*, and *Epcam* on a liver tissue section. Arrow indicates *Cyp1a1*-positive portal vein endothelial cells. (E) ISH for *Gja5*, *Sema3g*, and *Acta2* on a liver tissue section. Arrow highlights *Gja5* positive, *Sema3g* negative portal vein endothelial cells. (F) IF for CD31, LYVE1, and KIT on a liver tissue section from an *Acta2*^GFP reporter mouse. Arrows highlight KIT-positive last sinusoidal endothelial cells surrounding the central vein. (G) Violin plot showing the expression level of exemplary genes expressed by endothelial cells of the peribiliary vasculature (PBV) compared to other endothelial cell clusters (left panel, n refers to cells: e5c = 14, e2c = 114, e5 = 151, e8 = 155, e3 = 236, e6 = 112, e7 = 150, e1 = 152, e4 = 22). UMAP visualization of the expression level of peribiliary vasculature marker genes, *Rgcc*, *Cxcl12*, and *Car4* (right panel). (H, I) UMAP visualization of clustering results from human liver single-cell datasets GSE136103 (H) and GSE168933 (I). Cell clouds representing endothelial cells are indicated by the boxed area. (J, K) UMAP visualization of the magnified part of the UMAP landscapes from the human datasets showing the expression level of endothelial cell subtype markers: (J) *GJA5* for portal vein and hepatic artery, *SELP* for portal vein and central vein, *SEMA3G* for hepatic artery, and *STAB2* for sinusoids, or (K) *RGCC*, *CD200*, *CA4*, and *APLNR* for the peribiliary vasculature (PBV). PV portal vein, CV central vein, HA hepatic artery, BD bile duct. Scale bars are indicated in the respective image panels.

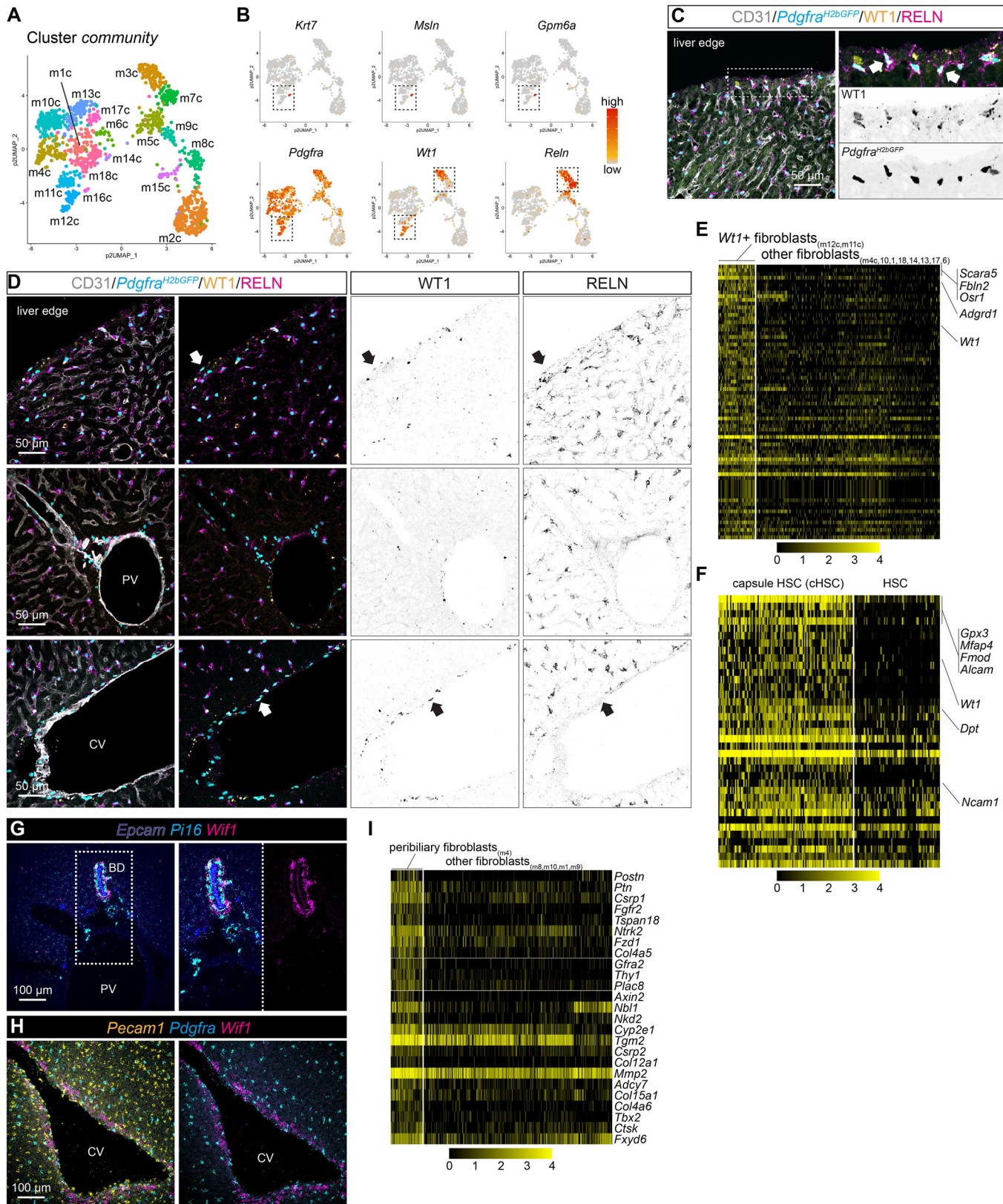

◀ **Figure EV3. Marker genes and tissue validation for mesenchymal cell subpopulations; cHSC and peribiliary fibroblasts.**

(A) UMAP visualization of the pagoda2 clustering result of the mesenchymal cell subset using the community setting for high resolution clustering. (B) UMAP visualization of the expression level of selected genes within the indicated boxed areas of clusters #m3 (m3c, m7c) and #m9 (m11c, m12c, m16c). (C) IF for CD31, WT1, and RELN on liver tissue section from a *Pdgfra*$^{H2bGFP}$ reporter mouse. The arrows indicate WT1, RELN, GFP triple-positive cells. (D) IF for CD31, WT1, and RELN on a liver tissue sections from a *Pdgfra*$^{H2bGFP}$ reporter mouse focusing on the liver edge (upper panel), the portal tract (middle panel), or the central vein (lower panel). Arrows highlight WT1 *Pdgfra*$^{H2bGFP}$ double-positive cells. (E) Heat map showing the expression of genes that exhibit enriched expression in *Wt1*+ fibroblasts (clusters #m11c, m12c), compared to all other fibroblast populations. (F) Heat map showing the expression of genes that exhibit enriched expression in cHSC (cluster #m3c), compared to HSC (cluster #m7c). (G) ISH for *Epcam*, *Pi16*, and *Wif1* on a liver tissue section. (H) ISH for *Pecam1*, *Pdgfra*, and *Wif1* on a liver tissue section. (I) Heat map showing the genes that exhibit enriched expression in peribiliary fibroblasts (cluster #m4), compared to all other fibroblast populations. PV portal vein, BD bile duct, HA hepatic artery, CV central vein. Scale bars are indicated in the respective image panels.

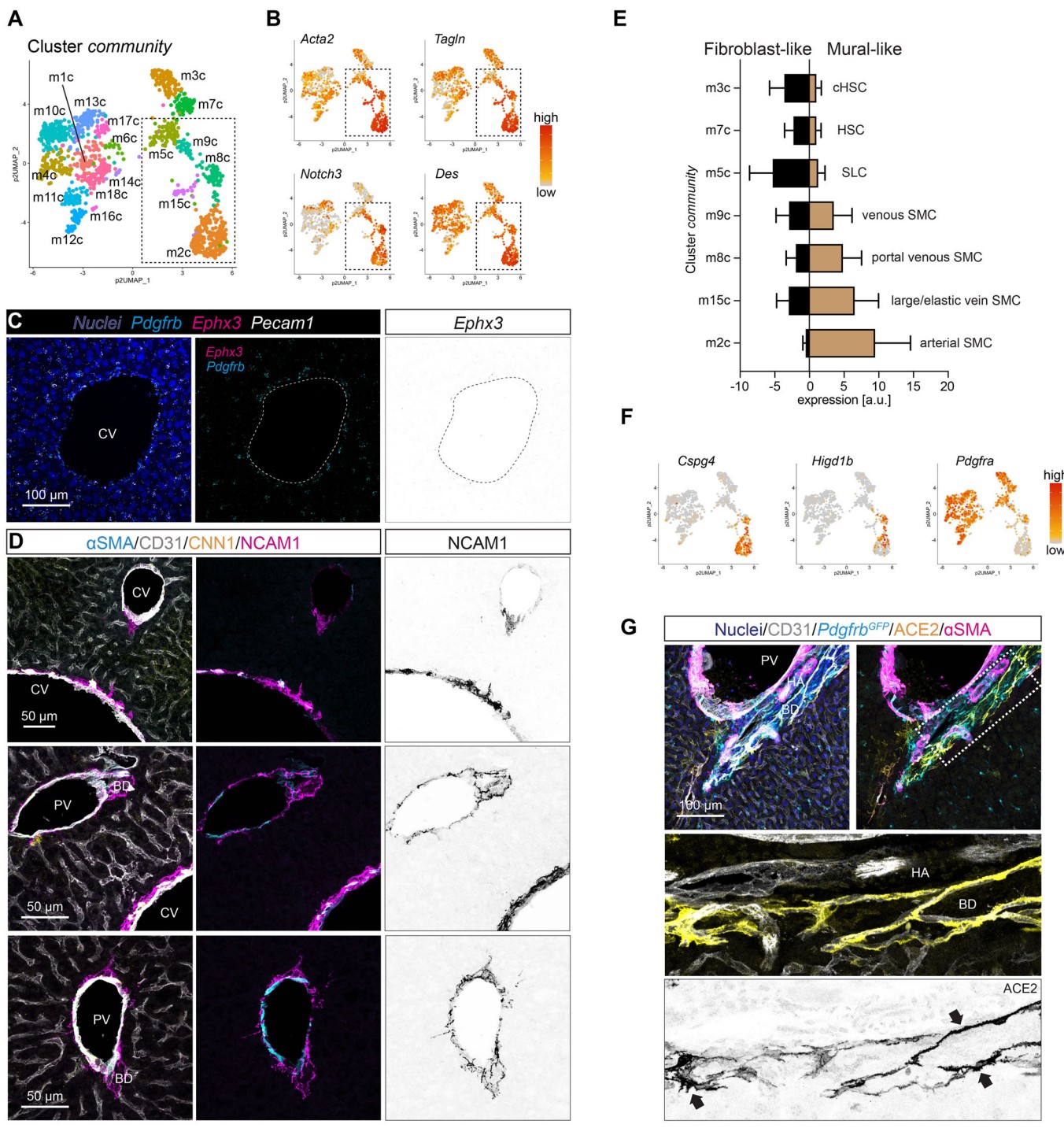

**Figure EV4. Analysis of mural cell subpopulations.**

(A) UMAP visualization of the clustering result of the mesenchymal cell dataset using the community setting for high resolution clustering, with the cell clouds containing vascular mural cell types indicated in the boxed area. (B) UMAP visualization of the expression levels of vascular SMC marker genes with the cell clouds containing vascular mural cells highlighted by the boxed area. (C) ISH for *Pdgfrb*, *Ephx3*, and *Pecam1* on a liver tissue section showing a region of a central vein. (D) IF for αSMA, CD31, CNN1, and NCAM1 on a liver tissue section, showing a region of a central vein (upper panel), portal and central vein (middle panel), or portal vein (lower panel). (E) Calculation of fibroblast or mural cell gene-score for the 90-gene list reported before (Muhl et al, 2020), for community clusters containing HSC and vascular mural cell types (stacked barplot showing mean values, for visualization fibroblast-gene expression values are inverted [multiplication by -1], error bars show s.d., *n* refers to cells: m3c = 113, m7c = 72, m5c = 87, m9c = 39, m8c = 72, m15c = 33, m2c = 236). (F) UMAP visualization of the expression levels of the pericyte marker genes (*Cspg4* and *Higd1b*), and *Pdgfra* in the mesenchymal cell dataset. (G) IF for CD31, ACE2, and αSMA on a liver tissue section from a *Pdgfrb*^GFP reporter mouse. Arrows indicate pericytes at the peribiliary vasculature. PV portal vein, BD bile duct, HA hepatic artery, CV central vein. Scale bars are indicated in the respective image panels.

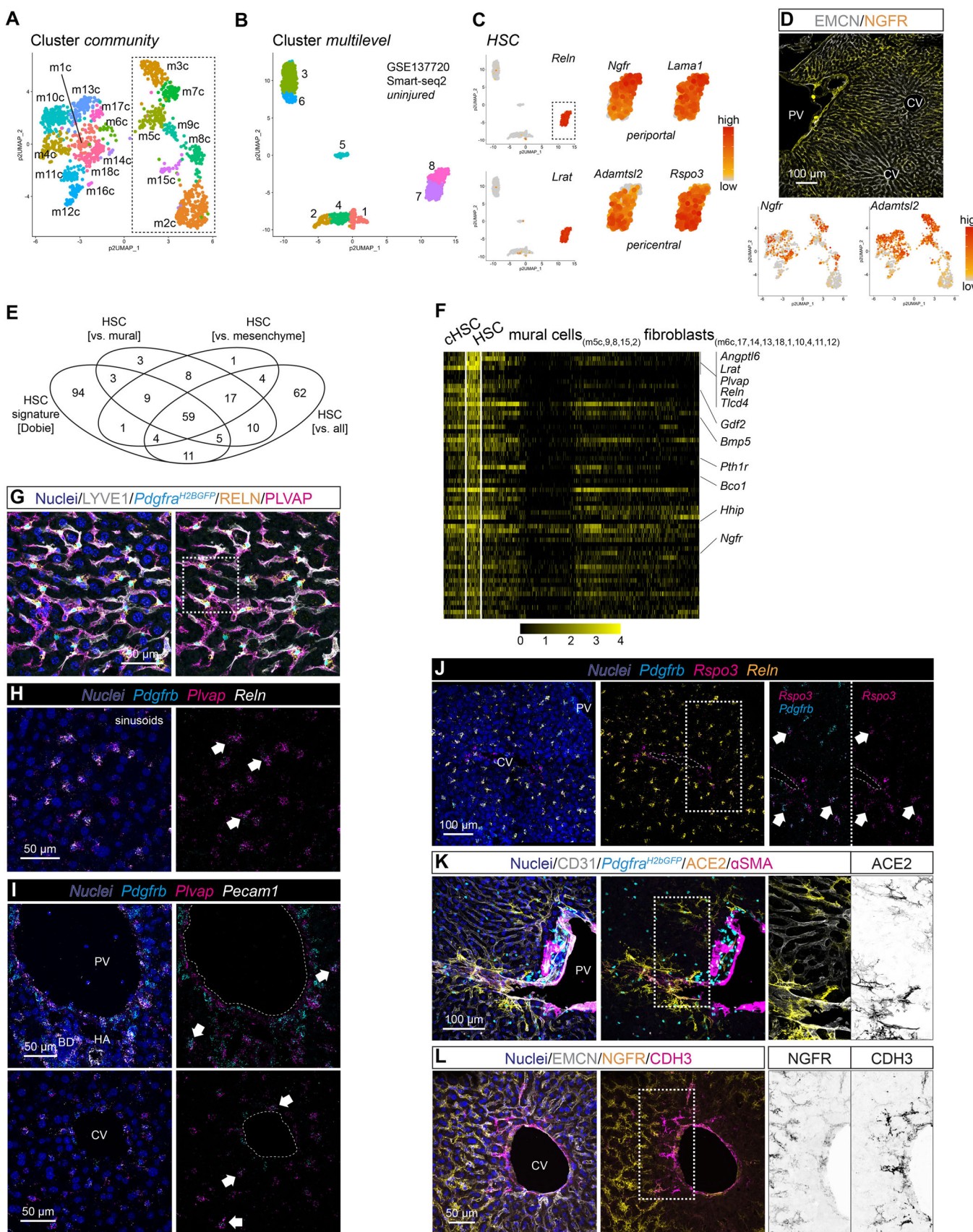

**Figure EV5. Analysis of HSC signature genes and marker gene expression.**

(A) UMAP visualization of the clustering result of the mesenchymal cell dataset using the community high resolution setting with HSC and vascular mural cell clusters indicated by the boxed area. (B) UMAP visualization of the clustering results of the GSE137720 (Smart-seq2) dataset using cells from uninjured samples. (C) UMAP visualization of the expression level of HSC marker genes (*Reln* and *Lrat*) and HSC zonation markers (*Ngfr, Lama1, Adamtsl2*, and *Rspo3*) magnified from the indicated area of the GSE137720 dataset. (D) IF for EMCN and NGFR on a liver tissue section (upper panel) and UMAP visualization of the expression levels of *Ngfr* and *Adamtsl2* in the mesenchymal cell dataset (lower panel). (E) Venn diagram showing the overlap of HSC-enriched genes calculated from the GSE137720 dataset using uninjured cells (Dobie), the mesenchymal cell subset (vs. mural, vs. mesenchymal), and the complete dataset (vs. all) (see also Dataset EV1). Note that 59 genes are commonly detected as enriched in HSC (cluster #m7c) in all four comparisons. (F) Heat map showing the expression levels of the 59 HSC enriched genes in the mesenchymal cell subset. (G) IF for LYVE1, RELN, and PLVAP on a liver tissue section from a *Pdgfra*^H2bGFP reporter mouse. The indicated boxed area is shown magnified in Fig. 7F. (H) ISH for *Pdgfrb, Plvap*, and *Reln*, on a liver tissue section focusing on the sinusoidal region. Arrows indicate *Pdgfrb Plvap* double-positive HSC. (I) ISH for *Pdgfrb, Plvap*, and *Pecam1* on a liver tissue section focusing on the portal tract (upper panel) or central vein (lower panel) region. Arrows indicate *Pdgfb Plvap* double-positive HSC. (J) ISH for *Pdgfrb, Rspo3*, and *Reln* on a liver tissue section. Arrows indicate *Pdgfrb Rspo3* double-positive HSC close to the central vein. (K) IF for CD31, ACE2, and αSMA on a liver tissue section from a *Pdgfra*^H2bGFP reporter mouse focusing on the portal tract. (L) IF for EMCN, NGFR, and CDH3 on a liver tissue section focusing on the central vein. PV portal vein, BD bile duct, HA hepatic artery, CV central vein. Scale bars are indicated in the respective image panels.

