## [Peer Review File · EMBO Reports]

A comprehensive molecular atlas of the mesenchymal cell types in the mouse liver.

Riikka Pietila, Guillem Genové, Giuseppe Mocchi, Yuyang Miao, Jianping Liu, Stefanos Leptidis, Francesca Del Gaudio, Martin Uhrbom, Elisa Vázquez-Liébanas, Sonja Gustafsson, Byambajav Buyandelger, Elisabeth Raschperger, Johan Björkegren, Emil Hansson, Konstantin Gaengel, Maarja Mäe, Marie Jeansson, Michael Vanlandewijck, Liqun He, Carina Strell, Xiao-Rong Peng, Urban Lendahl, Christer Betsholtz, and Lars Muhl

Corresponding author(s): Lars Muhl (lars.muhl@ki.se) , Christer Betsholtz (Christer.Betsholtz@ki.se)

Review Timeline:

Submission Date:	12th Feb 25
Editorial Decision:	21st Mar 25
Revision Received:	30th Jun 25
Editorial Decision:	12th Aug 25
Revision Received:	20th Aug 25
Accepted:	25th Aug 25

Editor: Deniz Senyilmaz Tiebe

Transaction Report:

Dear Dr. Muhl,

Thank you for transferring your manuscript to EMBO Reports. My apologies for this unusual delay in getting back to you. Three referees agreed to review your manuscript. So far, we have received two referee reports that are copied below. Given that both referees are in fair agreement that you should be given a chance to revise the manuscript, I would like to ask you to begin revising your study along the lines suggested by the referees.

Please note that this is a preliminary decision made in the interest of time, and that it is subject to change should the third referee offer very strong and convincing reasons for this. As soon as we receive the final report on your manuscript, we will forward it to you as well.

Referees express interest in the presented molecular atlas of the mouse liver. However, they also raise some concerns that need to be addressed to consider publication here.

Given these positive recommendations, we would like to invite you to revise your manuscript with the understanding that the referee concerns (as in their reports) must be fully addressed and their suggestions taken on board. Please address all referee concerns in a complete point-by-point response. Acceptance of the manuscript will depend on a positive outcome of a second round of review. It is EMBO reports policy to allow a single round of major experimental revision only and acceptance or rejection of the manuscript will therefore depend on the completeness of your responses included in the next, final version of the manuscript.

We realize that it is difficult to revise to a specific deadline. In the interest of protecting the conceptual advance provided by the work, we recommend a revision within 3 months. Please discuss the revision progress ahead of this time with me if you require more time to complete the revisions, or if you have questions or comments regarding the revision (also by video chat).

1. A data availability section providing access to data deposited in public databases is missing (where applicable).
2. Your manuscript contains statistics and error bars based on $n=2$. Please use scatter plots in these cases.

You can submit the revision either as a Scientific Report or as a Research Article. For Scientific Reports, the revised manuscript can contain up to 5 main figures and 5 Expanded View figures, and it should not exceed 27000 characters. If the revision leads to a manuscript with more than 5 main figures it will be published as a Research Article. In this case the Results and Discussion section should be separate. If a Scientific Report is submitted, these sections have to be combined. This will help to shorten the manuscript text by eliminating some redundancy that is inevitable when discussing the same experiments twice. In either case, all materials and methods should be included in the main manuscript file.

3) We replaced Supplementary Information with Expanded View (EV) Figures and Tables that are collapsible/expandable online. A maximum of 5 EV Figures can be typeset. EV Figures should be cited as 'Figure EV1, Figure EV2' etc... in the text and their respective legends should be included in the main text after the legends of regular figures.

4) a .docx formatted letter INCLUDING the reviewers' reports and your detailed point-by-point responses to their comments. As

part of the EMBO publication's Transparent Editorial Process, EMBO reports publishes online a Review Process File (RPF) to accompany accepted manuscripts. This File will be published in conjunction with your paper and will include the referee reports, your point-by-point response and all pertinent correspondence relating to the manuscript.

<https://www.embopress.org/page/journal/14693178/authorguide#transparentprocess>

5) a complete author checklist, which you can download from our author guidelines

<https://www.embopress.org/page/journal/14693178/authorguide>. Please insert information in the checklist that is also reflected in the manuscript. The completed author checklist will also be part of the RPF.

6) Please note that all corresponding authors are required to supply an ORCID ID for their name upon submission of a revised manuscript (<<https://orcid.org/>>). Please find instructions on how to link your ORCID ID to your account in our manuscript tracking system in our Author guidelines

<<https://www.embopress.org/page/journal/14693178/authorguide#authorshipguidelines>>

7) Before submitting your revision, primary datasets produced in this study need to be deposited in an appropriate public database (see <https://www.embopress.org/page/journal/14693178/authorguide#datadeposition>). Please remember to provide a reviewer password if the datasets are not yet public. The accession numbers and database should be listed in a formal "Data Availability" section placed after Materials & Method (see also

<https://www.embopress.org/page/journal/14693178/authorguide#datadeposition>). Please note that the Data Availability Section is restricted to new primary data that are part of this study. * Note - All links should resolve to a page where the data can be accessed. *

Additional information on source data and instruction on how to label the files are available:

<https://www.embopress.org/page/journal/14693178/authorguide#sourcedata>

9) Our journal encourages inclusion of *data citations in the reference list* to directly cite datasets that were re-used and obtained from public databases. Data citations in the article text are distinct from normal bibliographical citations and should directly link to the database records from which the data can be accessed. In the main text, data citations are formatted as follows: "Data ref: Smith et al, 2001" or "Data ref: NCBI Sequence Read Archive PRJNA342805, 2017". In the Reference list, data citations must be labeled with "[DATASET]". A data reference must provide the database name, accession number/identifiers and a resolvable link to the landing page from which the data can be accessed at the end of the reference. Further instructions are available at <http://www.embopress.org/page/journal/14693178/authorguide#referencesformat>

10) Regarding data quantification (see Figure Legends:

<https://www.embopress.org/page/journal/14693178/authorguide#figureformat>)

11) The journal requires a statement specifying whether or not authors have competing interests (defined as all potential or

actual interests that could be perceived to influence the presentation or interpretation of an article). In case of competing interests, this must be specified in your disclosure statement. Further information: <https://www.embopress.org/competing-interests>

12) Please also note our reference format:

13) All Materials and Methods need to be described in the main text using our 'Structured Methods' format, which is required for all research articles. According to this format, the Methods section includes a Reagents and Tools Table (listing key reagents, experimental models, software and relevant equipment and including their sources and relevant identifiers) followed by a Methods and Protocols section describing the methods using a step-by-step protocol format. The aim is to facilitate adoption of the methodologies across labs. More information on how to adhere to this format as well as a downloadable template (.docx) for the Reagents and Tools Table can be found in our author guidelines:

I look forward to seeing a revised version of your manuscript when it is ready. Please let me know if you have questions or comments regarding the revision.

Kind regards,

Deniz Senyilmaz Tiebe

Deniz Senyilmaz Tiebe, PhD
Senior Scientific Editor
EMBO Reports

Referee #1:

Manuscript title: A comprehensive molecular atlas of the cell types in the mouse liver.

General: In this manuscript, Pietilä et al. report a detailed single-cell transcriptome atlas of the various parenchymal and mesenchymal cell types of the adult mouse liver. They made use of transgenic reporter mice and antibody-based cell enrichment strategies, and sequenced a total of 3,491 cells using smart-seq. A very nice web portal complements the transcriptome dataset. Moreover, the authors validated key findings using orthogonal approaches, including immunofluorescence and in situ RNA hybridization. Altogether, while the overall cell number is perhaps on the lower end, this is a very comprehensive and useful resource for the community.

Major comments:

1. Among the endothelial cells, the authors found a small cluster likely derived from the peribiliary vasculature (PBV). This has not been described in detail before in published single-cell atlases of the hepatic vasculature, and is highly interesting. Cell-cell communication analyses with for instance cholangiocytes would be a very valuable addition to the paper, to better understand potential signaling/cross-talk between these two cell types.
2. Continuing on the previous comment, in Figure 7 the authors show the presence of PBV markers in specific clusters of the Ramachandran & Brunomo datasets. These datasets harbor healthy as well as cirrhotic livers. It would be very meaningful to know whether the PBV clusters as annotated by the authors are enriched in cirrhotic livers at all. This would be expected, since their markers potentially overlap with capillarized LSECs that are often found in cirrhosis. If so, IF or other means of in situ validation in human/mouse cirrhotic livers would be very interesting, but this is probably beyond the scope of this manuscript.
3. In figure 1B, PECAM1 and ALB seem quite widely expressed in all cell types. Especially for ALB, this could indicate the presence of ambient RNA contamination, which the authors did not account/correct for in their single-cell analysis. It would be very valuable to know what the overall gene expression patterns in the dataset look like after running it through DecontX (for example), to try and account for possible ambient RNA contamination.
4. Annotation of additional genes in the heatmaps in Figure 1F-L would be very helpful, as for some clusters, no genes are shown at all.
5. Cluster e6 groups among other central vein EC clusters, but also expresses many sinusoidal markers. Could this be a doublet, or a possible intermediate cell state?
6. While the figures are very detailed and compelling, there is a lot of information in every one of them, which makes the paper a

little hard to follow at times. Figure organization could thus be improved. A few suggestions:

- a. Moving of all GO enrichment analyses to one supplementary figure, as these are not the key message of the paper. This would create more space, which could be used to increase the size of some of the ISH/IF stainings, and/or to insert zoomed insets for more of the stainings.
- b. The concept of multilevel vs. community clusters is hard to follow. A graphical representation of what the difference between both is would be very helpful.
- c. The UMAP feature plots are very small, in terms of their dot sizes, but also in terms of their overall size. Since the paper is supplemented with a beautiful online portal, they may not even be 100% necessary. Removing at least some of them, and increasing the (dot) size of the others will be very helpful.
- d. As the paper and figures are so in-depth and comprehensive, a final summary figure (e.g., graphical summary), showing the main results and key novel findings, would be extremely helpful.
- e. Figure S1B is very hard to follow. Please adapt or remove.
- f. Some findings in the ISH images are very hard to see (e.g. S2J). Zoomed insets will be very helpful (as mentioned above).
7. The authors used Pagoda2 for analysis of their data, which is created for very large single-cell datasets (>Mio cells). It is thus slightly unexpected/unclear why the authors chose this method, and whether the appropriate adjustments have been applied to make this workflow work for their much smaller dataset. This will require a more detailed explanation and justification.
8. The authors filtered for cells with {less than or equal to} 1500 expressed genes. This seems very stringent, and may have resulted in potential loss of cells in their final analysis?

Referee #2:

Overall, this study presents a novel single-cell map of the mouse liver, and the web browser developed by the authors is a valuable tool for exploring their dataset. However, there are significant limitations, particularly in cell-type coverage and comparisons to existing mouse liver maps. The absence of certain previously identified cell populations raises questions about the use of the term "comprehensive map." Since this work is presented as a resource for cell type definitions and marker lists, greater clarity in defining and presenting these markers is essential. Addressing these points will strengthen the study and provide a more rigorous interpretation of the findings.

Major comments

1. While the resource presents a novel map of mouse liver cells the map is relatively small, containing only 3,491 cells. While this reviewer appreciated the value in the depth of Smart-seq2 data. The authors should elaborate on the value added by this dataset, especially given that Tabula Muris Senis contains nearly as many Smart-seq2 cells. A direct comparison with this resource (<https://www.nature.com/articles/s41586-020-2496-1>) would help clarify its unique contributions. Are there populations enriched in this map over that seen previously? On this theme the authors should address why there was no specific capture for lymphocytes in their design
2. The manuscript lacks a clear presentation of marker genes used for cell annotation. The authors should provide a comprehensive list of markers, specifying whether they are derived from their map (with thresholds and definitions included) or the literature. Some examples where the lack of clarity and detail made the result difficult to interpret.
 - a. In Figure 1F-L, markers used for sub clustering (parenchymal, endothelial, immune, mesenchymal) should be explicitly listed. The legend also needs a clearer description of what is shown. Was the annotation of cell type limited to the few genes listed here?
 - b. Author say "Annotation based on commonly accepted markers (Figure 1H) assigned one cluster (cluster #e2) as portal vein endothelial cells" but the marker in Fig 1H (Rspo3) is not specific to #e2. Possibly the authors meant to refer to another panel? Again presenting lists of markers in a table would be more clear. What are the markers used for "sinusoidal capillary endothelial cells"?
 - c. Statements such as "We identified fibroblasts residing in the peribiliary niche (cluster #m4)" should be supported with evidence. Are these markers novel from this map or established peribiliary niche markers from literature, in which case it should be cited?
 - d. It is difficult to evaluate the specificity of marker expression in Fig5C without being shown in the full mesenchyme map. Additionally cluster DEG statistics should be given for these markers. Exploring in the browser the authors have provided visually it appears there is expression of the listed genes in fibroblasts and HSCs as well. So quantifying the specific to mural cells would help.
 - e. In the public data analysis section there needs to be more quantification of the marker specificity, not just visualizations. For example it is unclear if Cdh3 is specific to any cell population in Fig7F. Quantification would aid interpretation.
3. The authors should provide more detailed methodology in a few other places, beyond cell marker specificity as mentioned.
 - a. In Fig S1A the expression of Alb is not entirely specific to hepatocytes. Did the authors consider the impact of contaminating mRNA. Were an methods used to assess or correct this (ie soupX).
 - b. In Fig2D it is unclear how the authors zoned endothelial cells. Was this a zonation score? They state "Annotation based on commonly accepted markers" what markers and what citations established these markers?
 - c. "Large-caliber portal veins" should be clearly defined. The differential NPNT levels between portal and central veins should be quantified across multiple regions not just representative examples.

- d. Terms such as "scattered" and "robust" should be replaced with quantitative descriptions.
 - e. Fig 7K the number of "top genes" should be stated and thresholds defined.
4. The authors comment on a "distinct hepatocyte population that expressed Lgr5 and located pericentrally" but it appears to be only 4 cells (Fig S2C). Without validating this population conclusions should not be made on so few cells.
 5. The combination of cholangiocytes and hepatocytes in cell-type-specific analysis is questionable. These populations are transcriptionally distinct (e.g., ARG1, CDH1, SPP1, and SOX9 expression). The rationale and markers used for this grouping should be explicitly stated. Distinct cholangiocyte populations might have emerged if clustered separately.
 6. The analysis of the myeloid lineage is rather limited. The statement "The liver contains at least two types of immune cells" is odd, given the extensive literature on liver immunology. Several key scRNA-seq studies on liver immune cell populations have not been cited, several of which defined far more than two liver immune cells.
 - a. Nine immune cell types identified in this study: <https://sciencedirect.com/science/article/pii/S0092867421014811?via%3Dihub>
 - b. Tabula Muris Senis (<https://www.nature.com/articles/s41586-020-2496-1>) annotated nine liver immune cell types.
 - c. This study identified 20 liver immune cell types:
https://journals.lww.com/hep/fulltext/2024/03000/single_cell_immune_profiling_of_mouse_liver_aging.10.aspx
 - d. This study was not cited: <https://www.nature.com/articles/s42003-024-07315-x#Sec14>
 - e. These citations should be added and novelty of the work discussed with these papers in mind.
 7. The authors present IF and ISH to validate finding in their single-cell map but this is never quantified as cell counts in a region. In all cases only one region is show per experiment. At least a few regions should be shown or ideally several regions should be quantified with images as representative examples.
 - a. Related, in Fig 4E there should also be a non liver edge region for comparison ideally with cell counts should be compared between edge and non edge
 8. In the analysis of public data conclusions are made based around integrating two studies with comments made on the non-overlapping populations (ie line 521 Fig 7HI). As the conclusions hinge on the effectiveness of the integration, more details need to be provided on the integration. It is possible that the effect is related to technical differences between the two studies. Additionally, what "pathological challenges" are there between Filliol and Mederacke. This reviewer appreciated that this could be found in the original publications but this is important context for this section to be understood.

Minor comments

1. Typo: "Kuppfer cells" should be corrected to "Kupffer cells" (line 255).
2. Figure S3E: It is unclear what is being presented. Is something missing, or should this be moved to the methods section as a table?
3. UMAP distances should not be interpreted as meaningful (e.g., "distributed to adjacent cell clouds in the UMAP analysis").
4. Figure S5G: There is numeric citation in the figure legend. This should be updated so the "90-gene list" can be understood by readers.

Point-by-point reply

On behalf of all co-authors, we would like to thank the reviewers for their constructive criticism, which has helped us to improve our paper. We are confident that our revised paper has become a strong candidate to meet the criteria for publication in *EMBO Reports*. We appreciate the overall positive response of the reviewers to the paper and the appreciation of the online database as a valuable resource for the research community. Indeed, we have already on different occasions received feedback from fellow researchers that the online database has been of great value for their work.

Some general revisions that have been addressed are, for example, the change of the title of our paper to 'A *comprehensive molecular atlas of the mesenchymal cell types in the mouse liver*' to better describe the paper's focus on vascular and perivascular stromal cell types analyzed and described. Further, we have revised the analysis of the publicly available datasets of hepatic stellate cells and fibroblasts to illustrate the useability and value of our cell type identification in a concise way.

The detailed answers to all comments made by the reviewers can be found in the point-by-point replies below.

Sincerely,
Lars Muhl and Christer Betsholtz

Referee #1:

Manuscript title: A comprehensive molecular atlas of the cell types in the mouse liver.

General: In this manuscript, Pietilä et al. report a detailed single-cell transcriptome atlas of the various parenchymal and mesenchymal cell types of the adult mouse liver. They made use of transgenic reporter mice and antibody-based cell enrichment strategies, and sequenced a total of 3,491 cells using smart-seq. A very nice web portal complements the transcriptome dataset. Moreover, the authors validated key findings using orthogonal approaches, including immunofluorescence and in situ RNA hybridization. Altogether, while the overall cell number is perhaps on the lower end, this is a very comprehensive and useful resource for the community.

Major comments:

1. Among the endothelial cells, the authors found a small cluster likely derived from the peribiliary vasculature (PBV). This has not been described in detail before in published single-cell atlases of the hepatic vasculature, and is highly interesting. Cell-cell communication analyses with for instance cholangiocytes would be a very valuable addition to the paper, to better understand potential signaling/cross-talk between these two cell types.

We appreciate the comment of the reviewer and agree that the identification of the endothelial cells in the PBV represents a new and interesting finding. According to the suggestion by the reviewer we have now included a cell-cell communication analysis using CellChat focusing on cell populations within the peribiliary niche; cholangiocytes, PBV endothelial cells, and peribiliary fibroblasts. The new analysis is included as Figure 4N,O.

2. Continuing on the previous comment, in Figure 7 the authors show the presence of PBV markers in specific clusters of the Ramachandran & Brunomo datasets. These datasets harbor healthy as well as cirrhotic livers. It would be very meaningful to know whether the PBV clusters as annotated by the authors are enriched in cirrhotic livers at all. This would be expected, since their markers potentially overlap with capillarized LSECs that are often found in cirrhosis. If so, IF or other means of in situ validation in human/mouse cirrhotic livers would be very interesting, but this is probably beyond the scope of this manuscript.

The reviewer raises an interesting point. However, while the human PBV populations were obtained from both healthy and cirrhotic samples, their proportions were hugely different between the two studies (as can be seen in Appendix Figure S3B,C) and therefore not suitable for comparison. Thus, we decided to not include any specific analysis of PBV phenotype and occurrence in healthy and cirrhotic human livers. We believe that dedicated studies with accordingly collected samples are necessary for proper in-depth analysis of the PBV also in human tissues, and such an analysis is beyond the scope of the present study.

3. In figure 1B, PECAM1 and ALB seem quite widely expressed in all cell types. Especially for ALB, this could indicate the presence of ambient RNA contamination, which the authors did not account/correct for in their single-cell analysis. It would be very valuable to know what the overall gene expression patterns in the dataset look like after running it through DecontX (for example), to try and account for possible ambient RNA contamination.

The reviewer points at an important aspect that is worthy of discussion in all single-cell RNA-sequencing studies, namely the cross-contamination by neighboring cells or ambient RNA. In our analysis, no 'decontamination' algorithms were applied. In our experience, judgement by biological knowledge is a powerful and sufficient method to identify single-cell transcriptomes that cannot be properly interpreted due to cross-contamination.

Regarding the expression of *Alb* or *Pecam1* in our dataset, our interpretation is that the expression of *Alb* in non-hepatocytes is not caused by simple contamination, since other specific and abundant hepatocyte genes display a different pattern across the dataset. Examples for this are: *Aldob*, *Car3*, *Adh1*, *Mat1a*, *Hmgcs2*, and *Scd1* as shown in the attached Data for reviewers 1B. If the expression of hepatocyte genes in non-hepatocyte cell populations would be due to cross-contamination by ambient RNA or cell fragments, one would assume a similar expression pattern for many of the abundant hepatocyte transcripts, however, in our dataset, this doesn't seem to be the case. The same is true concerning the expression of *Pecam1*, which is expressed for example by immune cells, although at a lower level compared to endothelial cells. Other abundantly expressed endothelial cell transcripts, such as *Kdr*, *Ptprb*, and *Tie1* display a different expression pattern across the dataset. An example of endothelial cell contaminated transcript can be seen in cluster i9c (Data for reviewers 1C) as also discussed in our paper (lines: 283-299).

The initial choice for coloring the dot plot (Figure 1B) may have been suboptimal, implicating a higher expression (dark gray color) compared to e.g., the barplot visualization. We have therefore changed to color scheme of Figure 1B to light grey to red.

4. Annotation of additional genes in the heatmaps in Figure 1F-L would be very helpful, as for some clusters, no genes are shown at all.

We thank the reviewer for pointing us at this shortcoming. More genes have been added to the heatmaps shown in Figure 1F,H,J,L. Additionally, all heatmaps for which not all genes are visible within the Figures (incl. EV and Appendix) can be accessed with full annotation as Supplementary data files via a link provided on the online database (<https://muhldatahub.org/Publications/LiverScRNAseq/database.html>).

5. Cluster e6 groups among other central vein EC clusters, but also expresses many sinusoidal markers. Could this be a doublet, or a possible intermediate cell state?

We agree with the reviewer that cells collected in cluster e6 likely represent an intermediate cell type with hybrid-expression of central vein and sinusoidal markers. The cells in cluster e6 express the highest level of *Kit* that we suggest is located close to the central vein, representing the last sinusoidal endothelial cells before entering the central vein. This is detailed in Figure 2L,M and Figure EV2F). This is also taken up in the discussion section (lines 644-651).

6. While the figures are very detailed and compelling, there is a lot of information in every one of them, which makes the paper a little hard to follow at times. Figure organization could thus be improved. A few suggestions:

a. Moving of all GO enrichment analyses to one supplementary figure, as these are not the key message of the paper. This would create more space, which could be used to increase the size of some of the ISH/IF stainings, and/or to insert zoomed insets for more of the stainings.

We thank the reviewer for this constructive feedback. For the reformatting of the Figures to adhere to the EMBO Reports layout and Figure-levels we have for example moved the GO analyses to the Appendix Figures and updated several of the IF and ISH figure-panels to provide better visibility.

b. The concept of multilevel vs. community clusters is hard to follow. A graphical representation of what the difference between both is would be very helpful.

The full UMAP visualizations for the clustering result using the multilevel or community setting are given in the main Figure or corresponding Expanded View/Appendix Figure. The background of the two different resolutions for the clustering have been explained in more detail in the corresponding methods sections (lines 857-862).

c. The UMAP feature plots are very small, in terms of their dot sizes, but also in terms of their overall size. Since the paper is supplemented with a beautiful online portal, they may not even be 100% necessary. Removing at least some of them, and increasing the (dot) size of the others will be very helpful.

We have changed the layout (increased point size and changed the color scheme) of the UMAP plots for better visibility and have reduced the overall number of feature plots for better readability of the figures.

d. As the paper and figures are so in-depth and comprehensive, a final summary figure (e.g., graphical summary), showing the main results and key novel findings, would be extremely helpful. We thank the reviewer for the appreciation of the data

and the suggestion to summarize the findings in the Synopsis/graphical abstract. A graphical abstract/Synopsis is now included with the revised paper.

e. Figure S1B is very hard to follow. Please adapt or remove.

Figure S1B has been moved to the new Appendix Figure S1B-D, and more detailed explanation has been added in the Figure and in the figure legend. The figure details the underlying clustering from the four different datasets which is the basis for the presentation in the barplot visualization. We believe this illustration has value to understand and interpret the data when presented as barplots. We therefore kept the figure.

f. Some findings in the ISH images are very hard to see (e.g. S2J). Zoomed insets will be very helpful (as mentioned above). We thank the reviewer for pointing out this weakness. We have changed several IF/ISH figure panels to add better visibility. Previous Figure S2J has been modified to show the single channel images as inverted (greyscale) color scheme (new Figure EV2D).

7. The authors used Pagoda2 for analysis of their data, which is created for very large single-cell datasets (>Mio cells). It is thus slightly unexpected/unclear why the authors chose this method, and whether the appropriate adjustments have been applied to make this workflow work for their much smaller dataset. This will require a more detailed explanation and justification.

We have for many years applied the pagoda2 R-software package for our scRNA-seq studies. In our experience the clustering results from pagoda2 can recapitulate different cell (sub)types well, especially for Smart-seq2 data, when compared to e.g., Seurat. We believe that early versions of pagoda2 were not yet intended for the analysis of very large datasets but developed generally for scRNA-seq analysis.

For reference see:

(<https://htmlpreview.github.io/?https://raw.githubusercontent.com/kharchenkolab/pagoda2/main/doc/pagoda2.walkthrough.html>)

8. The authors filtered for cells with {less than or equal to} 1500 expressed genes. This seems very stringent, and may have resulted in potential loss of cells in their final analysis?

We have applied the threshold of 1500 detected genes as minimum of expressed genes per cell also in our previous studies containing Smart-seq2 data. Compared to 10X Genomics-derived scRNA-seq data, the Smart-seq2 pipeline offers higher depth per cell, and therefore we are confident that our chosen thresholds are adequate. For comparison the reviewers can find a summary of the detected genes per cell in each cluster of the complete dataset presented as Violin plot (Data for reviewers 1A). The plot illustrates that for none of the clusters the average expression of genes is close to the lower limit (1500 genes).

Referee #2:

Overall, this study presents a novel single-cell map of the mouse liver, and the web browser developed by the authors is a valuable tool for exploring their dataset. However, there are significant limitations, particularly in cell-type coverage and

comparisons to existing mouse liver maps. The absence of certain previously identified cell populations raises questions about the use of the term "comprehensive map." Since this work is presented as a resource for cell type definitions and marker lists, greater clarity in defining and presenting these markers is essential. Addressing these points will strengthen the study and provide a more rigorous interpretation of the findings.

Major comments

1. While the resource presents a novel map of mouse liver cells the map is relatively small, containing only 3,491 cells. While this reviewer appreciated the value in the depth of Smart-seq2 data. The authors should elaborate on the value added by this dataset, especially given that Tabula Muris Senis contains nearly as many Smart-seq2 cells. A direct comparison with this resource (<https://www.nature.com/articles/s41586-020-2496-1>) would help clarify its unique contributions. Are there populations enriched in this map over that seen previously? On this theme the authors should address why there was no specific capture for lymphocytes in their design

We thank the reviewer for the reference to the Tabula Senis dataset. In contrast to our data, the Tabula Senis liver Smart-seq2 dataset does not contain hepatic stellate cells, mural cells, or fibroblasts, all cell types that are described with high detail in our study. Further, in our study we present the identification of endothelial cells from the PBV.

Therefore, we are confident that our study adds substantial value with regard to the description and identification of different hepatic cell types and their subpopulations. Since the Tabula Senis dataset is neither focused on hepatic cell types but cross-organ analysis of cells from different ages, we rather refrain from a direct comparison to our study. The enrichment strategies for our dataset are described in detail in the methods sections (*Preparation of single-cell solutions from mouse liver tissue* and *Fluorescent activated cell sorting (FACS)*). Lymphocytes have not been in the focus of our study and therefore, apart from targeted enrichment of CD68 positive cells, no further specific enrichment approaches were done.

For reference, in the tabula senis Smart-seq2 dataset: in total 2860 cells from the liver, thereof are 1162 hepatocytes, 617 LSEC, 262 KC, and the rest are other immune cells.

2. The manuscript lacks a clear presentation of marker genes used for cell annotation. The authors should provide a comprehensive list of markers, specifying whether they are derived from their map (with thresholds and definitions included) or the literature. Some examples where the lack of clarity and detail made the result difficult to interpret.

a. In Figure 1F-L, markers used for sub clustering (parenchymal, endothelial, immune, mesenchymal) should be explicitly listed. The legend also needs a clearer description of what is shown. Was the annotation of cell type limited to the few genes listed here?

Identification of genes with cluster-enriched expression (comparing cells in one cluster to all other cells in the dataset) was performed using the FindAllMarkers-function of the Seurat R-software package using standard thresholds as specified in the methods sections. Differential gene expression analysis (comparing two groups

of cells) was performed with the FindMarkers()-function of the Seurat R-package, the respective thresholds and applied parameters are detailed in the methods section and the accompanying supplementary tables with results of DEG and GO analysis are now provided with the paper as Table EV4 and EV5, respectively.

In general, the annotation of clusters/cells was done based on the cumulative evidence from different analyses presented in the paper, including all genes identified in the heatmaps and other differential gene expression analysis, as well as IF and ISH results, also retrospectively.

To offer better comprehension of the data we have added more genes to the heatmaps in Figure 1F,H,J,L, and made all heatmaps from which not all genes are visible in the figures available with full annotation as Supplemental data files accessible through a link on the online database page (<https://muhldatahub.org/Publications/LiverScRNAseq/database.html>).

b. Author say "Annotation based on commonly accepted markers (Figure 1H) assigned one cluster (cluster #e2) as portal vein endothelial cells" but the marker in Fig 1H (*Rspo3*) is not specific to #e2. Possibly the authors meant to refer to another panel? Again presenting lists of markers in a table would be more clear. What are the markers used for "sinusoidal capillary endothelial cells"?

The reviewer is right, that *Rspo3* is not a marker for cells in cluster e2, but instead a marker for cells in cluster e1 as indicated in the heatmap in Figure 1H. *Rspo3* is a marker for central vein endothelial cells. For the portal vein endothelial cells, *Gja5* that was shown in the original heatmap was the marker gene which is now replaced by *Ly6a* and *Cd34* (Figure 1H). For clarification we now indicate more genes in the heatmaps, and all heatmaps can be accessed with full annotation as Supplementary data files. See also response above.

c. Statements such as "We identified fibroblasts residing in the peribiliary niche (cluster #m4)" should be supported with evidence. Are these markers novel from this map or established peribiliary niche markers from literature, in which case it should be cited?

We thank the reviewer for pointing out this oversight. We have added additional marker genes (Violin plot in Figure 4H) and the corresponding references to the respective results section (line 369). Further, the location of cells in cluster m4 to the peribiliary niche has been validated by staining for specifically expressed genes in this cluster that to our knowledge have not been discussed before, for example *Wif1* (previous Figure 4L / new Figure 4J and Figure EV3G), *Gfra2* (previous Figure S4F / new Figure 4K), and *Pi16* (new Figure EV3G). Based on these observations, we are confident to suggest that cells collected in cluster m4 originate from the peribiliary niche.

d. It is difficult to evaluate the specificity of marker expression in Fig5C without being shown in the full mesenchyme map. Additionally cluster DEG statistics should be given for these markers. Exploring in the browser the authors have provided visually it appears there is expression of the listed genes in fibroblasts and HSCs as well. So quantifying the specific to mural cells would help.

The expression of the markers in the complete mesenchymal cell dataset UMAP was shown in previous Figure S5B and is now shown in Figure EV4B. The displayed markers were selected on generally accepted and previously shown enriched

expression of these genes in vascular mural cells (see e.g. Muhl et al. Dev. Cell 2022), and therefore no statistical methods were applied, although *Acta2* and *Notch3* are included in the heatmap in Figure 1L as markers for cluster m2 or m7, respectively. We appreciate that the expression of the displayed genes is not exclusive for mural cells, and neither do we proclaim this. Rather, the present work and our previous studies have established the similarities and heterogeneity of fibroblast and mural cells (pericytes and smooth muscle cells) in different organs, and the requirement to employ gene lists, rather than single genes for cell type definition.

e. In the public data analysis section there needs to be more quantification of the marker specificity, not just visualizations. For example it is unclear if *Cdh3* is specific to any cell population in Fig7F. Quantification would aid interpretation.

We thank the reviewer for the suggestion to modify the analysis concerning the public datasets. Accordingly, we have re-worked the analysis and believe that we have found a better way to convey our findings from the previously published datasets (new Figure 8, Appendix S6). In brief, we have simplified the re-analyzed data by concentrating on cell populations captured with hepatic stellate cell (HSC)-targeted approaches from the GSE172492 from Mederacke et al. and GSE137720 from Dobie et al. (10X) datasets. According to the reviewer's suggestion, we have now included more quantitative analysis of cells exhibiting an expression profile corresponding to the by us identified capsular (cHSC) or general HSC populations (Figure 7D-F). We are confident that the new analysis and its presentation conveys our interpretation that the cHSC population expands in disease (here CCl₄ treatment). Furthermore, this result suggests that a more refined interpretation of disease-regulated gene expression patterns is possible with the complemented cell type annotation at baseline. Nevertheless, future approaches that capture all cell types from the baseline/control state and the disease state within the same experimental setup are required for optimal interpretation of the results and the identification of disease-relevant gene expression patterns. Studies for which we provide important background information with our revised paper.

3. The authors should provide more detailed methodology in a few other places, beyond cell marker specificity as mentioned.

a. In Fig S1A the expression of *Alb* is not entirely specific to hepatocytes. Did the authors consider the impact of contaminating mRNA. Were any methods used to assess or correct this (ie soupX).

The reviewer has a good point; from the presented Figure 1B it seems as if the expression of *Alb* (and *Pecam1* for endothelial cells, see also related comment from reviewer 1 above) is not restricted to hepatocytes, with a possibility of cross-contamination. We haven't applied any algorithm-based correction methods to avoid loss of data but instead rely on our experience with the interpretation of scRNA-seq data and detection of possible contaminated transcriptomes. Regarding the expression of *Alb*, our interpretation is that other, non-hepatocyte cell types may express *Alb*, as indicated by the comparison with other abundant hepatocyte transcripts. See Data for reviewers 1B,C and response #3 to reviewer 1 above. Since hepatocytes are not in the focus of our paper we have not followed up on the expression of *Alb* in non-hepatocyte populations.

b. In Fig2D it is unclear how the authors zoned endothelial cells. Was this a zonation score? They state "Annotation based on commonly accepted markers" what markers and what citations established these markers?

We thank the reviewer for pointing out this inconsistency. We have added more specific description of the first used marker genes, as shown in Figure 1H (line 184-186). The genes shown in Figure 2D are additionally derived from several DEG analysis to detect overrepresented genes in the respective clusters (e.g., for e5c, e2c, combined [e3, e5, e6, e7, and e8], e1, and e4). Liver sinusoidal endothelial cell zonation has been described in detail in previous studies, and we don't claim any novel insights into this aspect, however, to place the discovery of peribiliary vascular endothelial cells into context we describe the general zonation pattern of hepatic endothelial cells. Further, we believe that the description and discussion of general attributes of cells in our dataset (although already characterized elsewhere) aids the interpretation and usefulness of our data and online database as a resource.

c. "Large-caliber portal veins" should be clearly defined. The differential NPNT levels between portal and central veins should be quantified across multiple regions not just representative examples.

We thank the reviewer for this critical insight. As suggested, we have performed quantitative measurements to score the amount of α SMA and NPNT coverage around portal and central veins. The results, together with new IF images are presented in the new Figure 5F,G, and the protocol for the quantification has been added to the methods section (*Image quantification*). Details about the applied statistical methods are described in the corresponding Figure legend and the Statistics and reproducibility section.

d. Terms such as "scattered" and "robust" should be replaced with quantitative descriptions.

We have edited the sentences including scattered or robust.

e. Fig 7K the number of "top genes" should be stated and thresholds defined.

Figure 7K has been updated (new Appendix Figure S6F) due to the re-analysis and different approach to analyze the data from published datasets.

The numbers for genes refer to the HSC signature genes presented in Figure 8D and Appendix Figure S6E. The respective Figure legends have been updated accordingly, and we provide more information about the origin of the genes presented in the Venn diagram (see also Table EV2). The result tables for the DEG analysis are provided in Table EV4.

4. The authors comment on a "distinct hepatocyte population that expressed Lgr5 and located pericentrally" but it appears to be only 4 cells (Fig S2C). Without validating this population conclusions should not be made on so few cells.

We agree with the reviewer that too few cells were captured to make conclusions about the Lgr5+ hepatocyte subtype. Hepatocytes are not in the focus of our study, and we have therefore decided to remove the Lgr5-related data from the revised paper, and have edited the text accordingly (line 165-167).

5. The combination of cholangiocytes and hepatocytes in cell-type-specific analysis is questionable. These populations are transcriptionally distinct (e.g., ARG1, CDH1, SPP1, and SOX9 expression). The rationale and markers used for this grouping

should be explicitly stated. Distinct cholangiocyte populations might have emerged if clustered separately.

The separation into four different sub-datasets for in-depth analysis was done based on gross cell type/class annotation, such as epithelial (parenchymal) cells for hepatocytes and cholangiocytes (e.g. expression of *Krt8/Krt18* in both cell types). Neither of these cell types was in the focus of the study, although both are important to provide context for the comparison to the stromal mesenchymal cell types. For clarification, we have changed the title of the manuscript to 'A *comprehensive molecular atlas of the mesenchymal cell types of the mouse liver*' for clarification. The high-resolution clustering using the pagoda2 community setting did not identify subclusters within the cholangiocyte cloud, which in our experience means that the collected cells don't exhibit sufficient transcriptional difference to be identified as different clusters, therefore our interpretation of the data is that only one subtype of cholangiocytes has been captured, as discussed in the results section (line 168-174).

6. The analysis of the myeloid lineage is rather limited. The statement "The liver contains at least two types of immune cells" is odd, given the extensive literature on liver immunology. Several key scRNA-seq studies on liver immune cell populations have not been cited, several of which defined far more than two liver immune cells.

We thank the reviewer to pointing us to this inaccurate wording. Immune cells were not the focus of our study but are important to add context. Therefore, the analysis of immune cells was restricted to the identification of basic subtypes (B-cells, monocytic cells) and the major population of Kupffer cells (528 out of 651 immune cells), which is discussed in relation to the other captured immune cell populations. The section describing our immune cell data has been revised to account for the known liver immune cell landscape more accurately (line 257-267). Additionally, different immune cell populations become more prevalent and relevant during hepatic diseases but since our focus lies on the characterization of hepatic cell types during homeostasis, we concentrated on Kupffer cells, the major immune cell population at steady state.

a. Nine immune cell types identified in this study:
<https://sciedirect.com/science/article/pii/S0092867421014811?via%3Dihub>

This paper is cited.

b. Tabula Muris Senis (<https://www.nature.com/articles/s41586-020-2496-1>) annotated nine liver immune cell types.

The Tabula Senis study although containing single-cell transcriptomes of the liver is not focused on liver cells. Therefore, we respectfully decided against the citation of the paper.

c. This study identified 20 liver immune cell types:
https://journals.lww.com/hep/fulltext/2024/03000/single_cell_immune_profiling_of_mouse_liver_aging.10.aspx

We thank the reviewer for alerting us about this publication. The paper is now included in the citations.

d. This study was not cited: <https://www.nature.com/articles/s42003-024-07315-x#Sec14>

The paper suggested by the reviewer discusses the effects of myocardial infarction for systemic immune responses, including the liver (in mice and zebrafish). We appreciate the suggestion to cite the paper, however, since the focus of our study is on the healthy homeostatic condition of liver cells, we respectfully decided to not cite the paper.

e. These citations should be added and novelty of the work discussed with these papers in mind.

We thank the reviewer for providing references to important papers concerning the immune landscape of the liver. We don't claim novelty for the immune cell populations, as we also don't do for the hepatocytes. However, it is important to have these cell types in the dataset to compare expression patterns of other described cell types. This is highlighted by the controversial observations about Kupffer cell subpopulations that express certain endothelial cell-specific genes such as *Esam* or *Cdh5*, as discussed in relation to the Kupffer cell population in the results section (line 287-299). The suggested articles by the reviewer report the hepatic immune landscape in the context of different diseases and disease models. Since our study mainly focuses on cells from the healthy liver at homeostatic conditions we respectfully refrain from including some of the suggested citations in our study (see above).

7. The authors present IF and ISH to validate finding in their single-cell map but this is never quantified as cell counts in a region. In all cases only one region is shown per experiment. At least a few regions should be shown or ideally several regions should be quantified with images as representative examples.

a. Related, in Fig 4E there should also be a non liver edge region for comparison ideally with cell counts should be compared between edge and non edge

To improve the impact of our results, we have followed the suggestion by the reviewer and have added quantitative measurements of cell counts for capsular HSC (new Figure 4F) and NPNT+ vascular SMC (new Figure 5G), including new images.

8. In the analysis of public data conclusions are made based around integrating two studies with comments made on the non-overlapping populations (ie line 521 Fig 7H). As the conclusions hinge on the effectiveness of the integration, more details need to be provided on the integration. It is possible that the effect is related to technical differences between the two studies. Additionally, what "pathological challenges" are there between Filliol and Mederacke. This reviewer appreciated that this could be found in the original publications but this is important context for this section to be understood.

We thank the reviewer for pointing out the unclear and over-complicated presentation of the public (disease) datasets, as well as inclusion of possible confounding factors by the use of several integrated datasets. For simplification and to better present our interpretations we re-evaluated our approach and restricted our analysis to two 10X datasets GSE172492 from Mederacke et al. and GSE137720 from Dobie et al., focusing on the captured HSC populations (previous Figure 7E-M and Figure S7C-E / new Figure 8A-G and Appendix S6). We have added gene signature analysis based on our findings from cells in homeostasis and thereby can quantitatively visualize the increase of capsular HSC (like) cells in disease (CCl₄) conditions (new Figure 8D-F).

Minor comments

1. Typo: "Kupffer cells" should be corrected to "Kupffer cells" (line 255).

Thank you, this has been corrected.

2. Figure S3E: It is unclear what is being presented. Is something missing, or should this be moved to the methods section as a table?

This has been removed.

3. UMAP distances should not be interpreted as meaningful (e.g., "distributed to adjacent cell clouds in the UMAP analysis").

We thank the reviewer for pointing this out. The sentences with related expressions have been edited (line 391, 411).

4. Figure S5G: There is numeric citation in the figure legend. This should be updated so the "90-gene list" can be understood by readers.

Thank you, this has been changed.

Referee #3:

In the study, "A comprehensive molecular atlas of the cell types in 1 the mouse liver", Pietilä R. et al. generated a single-cell dataset from the livers of healthy mice.

Despite significant advancements in single-cell and spatial transcriptomics, comprehensive transcriptomic information encompassing all major hepatic cell types at homeostasis remains scarce. The authors propose that such a resource would be beneficial, offering a baseline for comparison with transcriptomic data from pathological conditions. Furthermore, the authors emphasize the necessity of establishing precise molecular profiles for rarer hepatic cell types, such as vascular mural cells, fibroblasts, and cholangiocytes. To achieve this, the authors created a detailed molecular atlas utilizing scRNA sequencing in conjunction with immunofluorescence and in situ RNA hybridization. They characterized the zonation of poorly defined cell subtypes and compared it to existing literature.

This study provides a comprehensive view of liver homeostasis and cellular compartments. The methodology was well executed to address the research aims. However, much of the content in this study might have been reported. However, the results are well presented, cautious regarding the novelty and significance of this study. Several concerns need to be addressed.

Major comments:

1. Several murine and human liver scRNA-seq datasets (healthy and liver diseases) have been published. What is the necessity and novelty of conducting a new scRNA-seq analysis on healthy mouse livers?

The reviewer is right in that there are a multitude of liver scRNA-seq studies and datasets already published. However, to the best of our knowledge, none of these studies describes in detail several cell populations that we identify in our paper, such as the endothelial cells of the peribiliary vasculature (PBV), capsular HSC (cHSC), the different subtypes of vascular mural cells, and the heterogeneity of fibroblasts. We therefore believe that our study provides sufficient novelty and interest to the field.

2. Why did authors sort the cell types before scRNA sequencing detection but still mix them for bioinformatic analysis? Are 3491 liver cells adequate to generate this sophisticated subtype characterization or represent most liver cell types?

In our study, we aimed to characterize mesenchymal and vascular cells of the liver, especially vascular mural cells (in context to HSC) and mesenchymal cell types that are often underrepresented in published liver single-cell datasets. Therefore, we decided to enrich for these cell types by means of reporter gene expression and antibody panning, a strategy we have successfully applied in our previous studies. For the Smart-seq2 protocol cells are required to be deposited in 384-well plates, which is usually done using FACS. For unbiased identification of cell clusters, we have pooled all captured cells (transcriptomes) after enrichment, which we believe is the best practice for objective cell subtype identification. We explicitly state this in the methods sections, that antibody or reporter labeling has no influence on cell annotation (line 786-789). The annotation is instead based on the individual transcriptomes, clustering, and IF/ISH validation. Furthermore, we acknowledge and show at tissue level that some of the anatomically relevant cell types are missing in our data, such as endothelial cells originating from hepatic arteries.

3. The authors show several interesting cell subtypes according to the mathematical scRNA-seq data processing pipeline. It will be necessary to discuss how your cell subtype identification could be influenced by the unsupervised cell clustering settings (e.g., dimension reduction factors)? Authors will need to justify the reason for choosing the current setting.

The reviewer raises an important point inherent to all single-cell dataset analyses methods. We have applied standard parameters for mathematical classification and clustering, as described in the methods section (e.g. *Sequence data processing* section). Further, we highlight the limitations and raise the awareness about the limitations of these mathematical approaches in the 'Limitations to the study' section. For biological validation of our in silico identified cell subtypes we have employed IF and ISH for (novel and previously reported) cell subtypes to validate; first their occurrence in liver tissue and second their anatomic localization.

4. It has been known that scRNA-seq acquires better non-parenchymal quality but is likely poor in hepatocyte quality and numbers. To enrich cell amount and improve subtype resolution, a combined sc and snRNA-seq analysis is recommended. Authors may mention it in the discussion or limitation sections.

Hepatocytes have not been the focus of our study and therefore we respectfully refrain from discussing the differences between single-cell and single-nuclei RNA-seq of parenchymal cells. Although, the transcriptomes that we have obtained from single hepatocytes appear to exhibit good quality.

A following comment: As the predominant cell type in the liver, the hepatocyte-related analyses and discussions were insufficient in this study. Hence, more findings on hepatocyte subtypes need to be highlighted rather than only confirming previous knowledge in the literature. For instance, it may generate novel points by elucidating interactions between zoned hepatocytes and surrounding non-hepatocytes.

As such, authors can consider some methods or approaches, for one instance, using CellChat, CellphoneDB and other equivalent tools to map cell-cell interactions (among all cell subtypes). This may generate an in-depth understanding of

microenvironmental influences or importance with the novel cell categories; for another instance, using cell trajectory prediction tools to assess evolutionary relevance among diverse cell subpopulations.

Otherwise, I would suggest authors narrow down the aims by focusing on non-parenchymal cells and potentially reshape the study.

We thank the reviewer for alerting us about this issue. Since hepatocytes have not been in the focus of our study, we have not collected enough cells to allow for an in-depth analysis that would provide significant novelty above the already existing information about hepatocyte heterogeneity. For our study, the hepatocytes serve as a comparative cell type to better determine the specific expression patterns of the other cell types in the liver. For example, to be able to identify potential contamination of co-expression of genes. To avoid misunderstandings, we have changed the title of the manuscript to '*A comprehensive molecular atlas of the mesenchymal cell types of the mouse liver*'.

5. Authors utilized immunofluorescence IHC and ISH methods to validate spatial distribution and relationships multiple cell types. However, the current data presentation appears weak. Authors may consider providing cell quantities and spatial correlations in larger histological scales. Otherwise, authors can gather published stRNA-seq datasets as references and verify significant findings.

We thank the reviewer for this suggestion. In the revised paper, we have updated several of the IF/ISH images for better representation and have added quantitative measurements of two identified cell types: capsular HSC (cHSC) and NPNT-positive vascular SMC that are presented in Figure 4F and Figure 5G, respectively.

In general, for all ISH or IF analyses, tissue sections from at least two mice were analyzed with similar results, as stated in the 'Statistics and reproducibility' section (lines 1017-1020).

6. In the last part (mainly Figure 7), authors gathered several published scRNA-seq datasets to verify the gene signatures reported in the literature or analyzed in their own datasets. The whole result part might only make loose links descriptively, which are not enough to provide solid evidence to understand functions or pathological contributions of these cell subtypes in liver injury or diseases. Authors may consider specific in vitro experiments to validate cell phenotypes and functions. For instance, authors may assess functional and transcriptomic alterations on mouse primary HSCs and endothelial cells (under injury interventions).

We thank the reviewer for highlighting the unclear presentation within the previous Figure 7. The analysis of published datasets from hepatic disease models was revised and simplified to present our interpretations in a clearer, more accessible way, including quantitative analysis and illustrations (Figure 8D-F).

We appreciate the suggestions from the reviewer to perform in vitro assays to validate our conclusions. However, we believe that such experiments would require a high amount of optimization for accurate interpretations, since phenotypes of isolated cells change rapidly in culture conditions. Instead, dedicated in vivo experiments using lineage tracing approaches would be better suited to investigate the relationships of e.g., the different HSC populations in homeostasis and disease as inferred by our analysis. Although highly relevant, we feel that such experiments fall outside the scope of our present study.

7. I appreciate that the authors humbly indicated multiple limitations of the study. However, some of the limitations did weaken the conclusions of this study. Authors may still need to work on finding solutions by either enriching the analysis methods or using more relevant datasets.

We thank the reviewer for the critical assessment of the 'limitations' section. Nevertheless, we are confident that our study provides several important new aspects for liver biology, such as the identification of PBV endothelial cells, capsular HSC, vascular mural cell populations, fibroblast heterogeneity, and we are confident that our interpretations and data as they are presented, together with the online database, will provide researchers with new and important insights about the cellular heterogeneity of the liver mesenchyme. In our point of view, it is important to discuss the limitations of a study to allow readers who are maybe not experts in single-cell transcriptomic analysis an objective interpretation of the presented data. Therefore, we would like to keep the 'Limitation of the study' section as revised.

Specific comments:

1. In Figure 3, authors annotated B cells, myeloid monocytic cells, and Kupffer cells from scRNA-seq data. However, authors displayed spatial and transcriptomic characteristics of these cell types in the healthy mouse livers, which have been reported in many previous studies. To make differences, authors may consider digging deeper into cell communications or illustrating spatial specificity in larger histological scales.

A more detailed characterization of hepatic immune cell populations was not the focus of our present study. The immune cell populations are included to provide context and as reference cell types (like hepatocytes and cholangiocytes) for better interpretation of the gene expression patterns of mesenchymal and vascular cells. See also the related comments to point 6 or reviewer #2 above. To emphasize the narrower focus of our study we have changed the title of the paper to '*A comprehensive molecular atlas of the mesenchymal cell types in the mouse liver*'.

A following technical question: were blood cells removed from the liver before sampling?

- If so, authors may look into other liver-infiltrated immune cell types, including T cells, dendritic cells, neutrophils, and so on, to hopefully reveal some novel points.

- If not, influences of blood cell presence (e.g., myeloid monocytic population) in your analyses need to be discussed.

We thank the reviewer for highlighting this point. Indeed, no perfusion was done before organ and cell harvest, therefore, likely circulating immune cells are present in the dataset. We have now addressed this in the results section (lines 272-274).

2. In Figure 3, the authors revealed a suspicious KC population expressing adhesion molecules (e.g., *Cdh5*). However, the IF/IHC results are not clear and moderately convincing enough to reach conclusions. Authors may allocate colocalization analyses on fluorescent signals or utilize the flow cytometry on isolated liver KCs for better characterization.

We appreciate the criticism of the reviewer concerning the expression of *Cdh5* in Kupffer cells. Indeed, the validation of *Cdh5* expression by Kupffer cells is challenging due to the strong expression of *Cdh5* in sinusoidal endothelial cells and the tight interaction between Kupffer cells and endothelial cells. We believe that our

scRNA-seq analysis (Appendix Figure S4B) is sufficient to suggest expression of *Cdh5* in Kupffer cells at distinctively higher levels compared to other canonical endothelial cell markers (*Tie1*, *Ptprb*), thereby ruling out a possible contamination from endothelial cell transcriptomes. Further, recent publications that are not explicitly discussing the expression of *Cdh5* in Kupffer cells but displaying the expression in their data (Scott et al. 2016, e.g., Fig 1B (doi:10.1038/ncomms10321), Scott et al. 2018,), e.g. Fig. 3H, 4E (doi.org/10.1016/j.immuni.2018.07.004), Guo et al. 2024, e.g. Fig 3B (doi.org/10.1016/j.immuni.2024.08.016), Zhao et al. 2022, e.g. Fig 2D (doi.org/10.1172/JCI1150489), and Zhao et al. 2024, e.g. Fig 2C,I (doi.org/10.7554/eLife.95811) independently validate the expression of *Cdh5* in Kupffer cells. These examples from previous studies are consistent and include bulk as well as single-cell RNA-sequencing data. Certainly, the expression of *Cdh5* in Kupffer cells is an interesting aspect, but due to the complexity and the amount of required analysis, we feel that the inclusion of more data related to the expression of *Cdh5* in Kupffer cells lies outside of the scope of the present study. Instead, we have initiated a follow-up study to investigate *Cdh5* expression in Kupffer cells in more detail.

3. In Figure 4L, the authors need to provide more evidence to characterize peribiliary fibroblasts, which is a critical finding. Cell-cell interaction analyses from scRNA-seq data or in-depth quantification (e.g., spatial correlation, cell distancing) from IF/IHC are suggested.

In addition, taking a global view of the liver cell clusters (UMAP in Figure 1D-K), the m4 cluster seems quite distancing from cholangiocytes, potentially indicating low transcriptomic similarity and weak cellular interactions. Interpretation and discussions are needed.

I would be rather cautious about concluding such a cell subtype, unless authors can provide more characteristic evidence from spatial exploration.

We thank the reviewer for expressing concerns about the identification of peribiliary fibroblasts. To clarify, we have identified specific transcripts in cells of cluster m4 and provide the ISH data that shows presence of these transcripts (*Wif1*, *Gfra2*, *Pi16*) in the peribiliary niche. Earlier studies have reported specific mesenchymal/fibroblast populations of the portal tract that locate close to the bile duct (peribiliary). Reported transcripts in these previous studies match with our annotation (Figure 4H-J, Figure EV3G-I) (line 366-376). Another recent study investigated specifically the stromal cell types of the extrahepatic bile duct using scRNA-seq (Singh et al. 2024 DOI:10.1097/HC9.000000000000368). We have utilized their data and can show that one of the mesenchymal cell populations of the Singh et al. study expresses the same marker genes as the peribiliary fibroblasts in our study (Data for reviewers 2). With the spatial continuum of the extrahepatic and intrahepatic peribiliary mesenchyme, this further strengthens our annotation of cluster m4 as peribiliary fibroblasts and that they can be distinguished from other portal fibroblast populations.

As stated in the limitations to the study sections (line 720-730), we advocate for follow-up studies to investigate certain specific cell types with, for example, refined lineage tracing models. Such studies may employ data from our present paper for the development of relevant lineage tracing models to decode the portal mesenchyme in high detail.

The spatial distance of fibroblast (peribiliary fibroblasts, cluster m4) and cholangiocyte transcriptomes in the UMAP display of the combined analysis cannot be translated into spatial relationship between the corresponding cells in tissues. Since fibroblasts and cholangiocytes represent different cell classes (mesenchymal vs. epithelial), highly different transcriptomes are expected which consequently places them apart in the UMAP display. Even if cell-cell communication takes place between peribiliary fibroblasts and cholangiocytes, most cell-cell interaction is not homotypic, but complementary interaction of different proteins reflecting distinct gene expression patterns. We have included a cell-cell communication analysis of the peribiliary cell types (new Figure 4N,O) that highlights extracellular matrix interactions between cholangiocytes and peribiliary fibroblasts.

4. Line 527-529: 'Comparing the marker genes of myHSC and cyHSC with those of cHSC and general HSC revealed substantial overlap between these subpopulations (Figure 7J,K, Table 2), suggesting that myHSC may originate from cHSC'.

Are there any established methods or algorithms behind it? Considering they are all identified as HSCs, having around 25% transcriptomic similarity profile may not be sufficient to make conclusions on cell origination.

We thank the reviewer for highlighting the unclear presentation and description of the data in previous Figure 7. For clarification and better understanding, we have re-analyzed the data from the public dataset and focused on only two datasets of HSC (GSE172492 from Mederacke et al. and GSE137720 Dobie et al.) presented in the new Figure 8A-G and Appendix Figure S6A-G. We have employed standard methods to visualize expression of selected genes and gene signatures. No statistical methods have been applied. The application of statistical methods on single-cell data is usually not informative if pseudo-bulk is not applicable, however, studies need to be designed for such approaches and since this is a reanalysis of published datasets, we opted to omit statistical analysis. Nevertheless, we believe that the new way of presenting our analysis illustrates clearly the increase in cells matching our capsular HSC (cHSC) gene signature in samples from disease conditions (new Figure 8D-G), suggesting a relevance of cHSC in hepatic disease.

The reviewer is correct that the provided evidence may not be sufficient to make conclusions about cellular origin, and this has neither been our intention, rather we would like to raise awareness about the inconclusive cellular origin of HSC populations in disease states, and our data suggests a possibility for capsular HSC (cHSC) as—at least in part—the origin of activated (myofibroblastic) HSC populations in disease. Of course, dedicated *in vivo* experiments using lineage tracing approaches are needed to make well-founded conclusions about the cellular origin(s) of HSC in disease. This aspect is highlighted in the discussion section (lines 604-619).

5. In Figure 7Q, to support the predicted fibroblast activation, certain types of cell trajectory methods are suggested (e.g., RNA velocity methods, Monocle).

We thank the reviewer for this suggestion. We have included a new analysis using the Monocle3 R-software package to infer trajectories to the fibroblast populations from disease models. These analyses are now displayed in new Figure 8H-K and Appendix Figure S7.

6. It will be important to compare cell subtypes by gender differences since the authors collected cells from both male and female mice.

The reviewer mentions an important and timely aspect of sex-dependent analyses. However, our dataset, although containing cells from female and male mice, was not designed to investigate sex-specific differences and hence does not provide the required balanced cell numbers from both sexes for all cell types. Therefore, we decided to not include any (sub-optimal) analysis of sex differences into the present study. We have now included UMAP visualizations and tables where distribution of collected cells from female and male mice is shown for all cell populations (Appendix Figure S1A). Also, the tissue samples used for validation by IF and ISH were from both sexes and showing comparable results. Nevertheless, follow-up studies focusing on certain cell types should address the possible sex-specific phenotypes of distinct cell types. These studies can now be planned and conducted based on the information provided in our present study. For transparency, we have added a statement in the limitations of the study sections (lines 730-723).

Data for reviewers 1

A Violin plot showing the detected number of genes in each cell, grouped by identified clusters within the complete dataset. **B** Barplots showing the expression level of genes highly expressed by hepatocytes. **C** Barplots showing the expression level of genes highly expressed by endothelial cells. The barplot visualization allows for comparison of gene expression patterns across the dataset. Note the different expression patterns of the hepatocyte and endothelial cell specific genes in the other clusters and cell types, which suggests no cross-contamination of transcriptomes with hepatocyte or endothelial cell genes, except cluster i9c (marked by arrow). **D** Dot plot visualization of the same genes presented in B, C and additional hepatocyte and endothelial cell marker genes.

Data for reviewers 2

All cells of extrahepatic bile ducts from three wt samples in the GSE223099 dataset (Singh et al. 2024) were first analyzed using the Seurat R-software package. Identified mesenchymal cells were selected for subsequent in-depth analysis. **A** Heat map showing the top20 genes representative for each identified cluster from the selected mesenchymal cells. **B** UMAP visualization of the selected mesenchymal cell dataset color coded for the identified clusters. **C** UMAP visualization of the expression levels of genes identified in our manuscript to exhibit enriched expression in peribiliary fibroblasts. Note the expression in a small subpopulation of cluster 0, indicated by the arrow. This suggests a relationship of intrahepatic peribiliary fibroblasts identified in our manuscript with extrahepatic peribiliary fibroblast populations (from the GSE223090 dataset) and strengthens our conclusions about the peribiliary fibroblasts identified in our study.

Dear Lars,

Thank you for submitting your revised manuscript. It has now been seen by two of the original referees.

As you will see, referees find that the study is significantly improved during revision and recommend publication. However, the editorial points below need to be addressed before I can accept the manuscript.

- Please remove the Author Contributions section from the manuscript text.
- We note the following regarding in-text callouts:
 - o A Table 1 cited, but not provided
 - o "Supplementary information" and "Supplementary data files" are incorrect callouts and should be updated.
- We note that the Tables EV1-5 are better suited for the Dataset format. We note that Table EV6 should be incorporated into the Reagents & Tools table. The table nomenclature should be updated as Dataset EV1-5, respectively, and this should be corrected in all places (source file names, legends, titles in the manuscript tracking system, manuscript text callouts and in the Reagents & Tools table).
- In the Appendix file, the author list and affiliations are not needed. However, we need a table of contents, where each item and their page numbers are listed on the title page.
- We note the following regarding the Data Availability section:
 - o Please deposit the data at <https://muhldatahub.org/Publications/LiverScRNAseq/database.html> in a public database (please see <https://www.embopress.org/page/journal/14693178/authorguide#datadeposition>)
 - o The Data Availability section is reserved for the primary datasets generated in the study. Therefore, existing datasets that were re-analyzed in the study (GSE172492, GSE168933, GSE136103, and GSE137720) need to be removed from this section and instead need to be referenced in the relevant locations in the manuscript text by following our dataset citation guidelines (please see <https://www.embopress.org/page/journal/14693178/authorguide#referencesformat>).
 - o Please make the dataset GSE297062 publicly available and remove the reviewers' access link/token from the Data Availability section.
 - o Please remove the following sentence: Further information about reagents and resources are available from the corresponding author, Lars Muhl (Lars.Muhl@ki.se) upon reasonable request.
- Nomenclature of individual EV Figure files needs to be corrected as Figure EV1, etc. (instead of Expanded View Figure EV1).
- Our production/data editors have asked you to clarify several points in the figure legends - Figure Legends (main + EV):
 - o Please note that the exact p values are not provided in the legends of figures 4F, 5G.
 - o Please note that the box plots need to be defined in terms of bounds of box and whiskers, and percentile in the legends of figure 5G.
 - o Please note that the box plots need to be defined in terms of minima, maxima, centre, bounds of box and whiskers, and percentile in the legend of figure EV4 E.
 - o Please note that information related to n is missing in the legends of figures 2D, 3C, 4H, 5D, H; 8I, K; EV1 F, EV2 G, EV4 E.
- Please provide another image for the synopsis. This image should provide a rapid overview of the question addressed in the study but still needs to be kept fairly modest since the image size cannot exceed 550 (width) x 300-600 (height) pixels. I note that you already submitted a synopsis image, however it is too busy and it will not be legible when formatted according to our website display requirements.

Thank you again for giving us to consider your manuscript for EMBO Reports, I look forward to your minor revision.

Kind regards,

Deniz

--

Deniz Senyilmaz Tiebe, PhD
Senior Scientific Editor
EMBO Reports

Referee #1:

The revised manuscript is a substantial improvement to the already very nice first version, and the authors have properly addressed all comments and concerns. I have no further comments, and recommend this manuscript for publication in its current form.

Referee #2:

The reviewer appreciates the additional analysis and quantification provided by the authors. The revised manuscript presents a clearer focus on the mesenchymal populations, as reflected in the updated title. These changes have improved the clarity of the findings however as the authors chose not to expand their analysis of hepatocyte, cholangiocyte, and immune populations, as suggested by two reviewers, it has become apparent that the overall scope of the manuscript is somewhat narrow.

The authors have addressed all minor editorial requests.

Dr. Lars Muhl
Karolinska Institute
Department of Medicine, Huddinge
Huddinge SE-141 57
Sweden

Dear Lars,

Thank you for submitting your revised manuscript. I have now looked at everything and all is fine. Therefore, I am very pleased to accept your manuscript for publication in EMBO Reports.

Congratulations on a nice work!

Kind regards,

Deniz

--

Deniz Senyilmaz Tiebe, PhD
Senior Scientific Editor
EMBO Reports
